# Coupling cellular drug-target engagement to downstream pharmacology with CeTEAM

Nicholas C. K. Valerie [1] ✉, Kumar Sanjiv[2], Oliver Mortusewicz [2], Si Min Zhang [2], Seher Alam[1], Maria J. Pires[1], Hannah Stigsdotter[2], Azita Rasti[2], Marie-France Langelier [3], Daniel Rehling[4], Adam Throup [2], Oryn Purewal-Sidhu[2], Matthieu Desroses[2], Jacob Onireti [1], Prasad Wakchaure[2], Ingrid Almlöf[2], Johan Boström[1], Luka Bevc[2], Giorgia Benzi[2], Pål Stenmark [4], John M. Pascal [3], Thomas Helleday[2], Brent D. G. Page[2,5,6] & Mikael Altun [1,6]

Cellular target engagement technologies enable quantification of intracellular drug binding; however, simultaneous assessment of drug-associated phenotypes has proven challenging. Here, we present cellular target engagement by accumulation of mutant as a platform that can concomitantly evaluate drug-target interactions and phenotypic responses using conditionally stabilized drug biosensors. We observe that drug-responsive proteotypes are prevalent among reported mutants of known drug targets. Compatible mutants appear to follow structural and biophysical logic that permits intra-protein and paralogous expansion of the biosensor pool. We then apply our method to uncouple target engagement from divergent cellular activities of MutT homolog 1 (MTH1) inhibitors, dissect Nudix hydrolase 15 (NUDT15)-associated thiopurine metabolism with the R139C pharmacogenetic variant, and profile the dynamics of poly(ADP-ribose) polymerase 1/2 (PARP1/2) binding and DNA trapping by PARP inhibitors (PARPi). Further, PARP1-derived biosensors facilitated high-throughput screening for PARP1 binders, as well as multimodal ex vivo analysis and non-invasive tracking of PARPi binding in live animals. This approach can facilitate holistic assessment of drug-target engagement by bridging drug binding events and their biological consequences.

Establishing drug-target engagement in cells is a pillar of drug discovery critical for reducing attrition in the development of new medicines[1–4]. Techniques, such as the cellular thermal shift assay (CETSA)[5], have enabled an advanced understanding of biophysical drug-target interactions and complement traditional proteome profiling approaches[6] to unravel biological or therapeutic effects. Nonetheless, current techniques provide incomplete characterizations of drug pharmacology because they are unable to seamlessly integrate

downstream cellular responses. Commonly, this is because the cellular environment must be perturbed to detect drug binding, which can complicate the interpretation of relevant biology in the unperturbed state[7,8]. Therefore, orthogonal approaches to understand drug pharmacology more holistically are warranted.

One avenue to circumvent this issue involves drug-dependent modulation of protein stability and abundance in cells, which is governed by protein translation and destruction (proteolysis). Binding of

[1]Science for Life Laboratory, Division of Clinical Physiology, Department of Laboratory Medicine, Karolinska Institutet, Karolinska University Hospital, Huddinge SE-141 52, Sweden. [2]Science for Life Laboratory, Department of Oncology-Pathology, Karolinska Institutet, Solna SE-171 65, Sweden. [3]Département de Biochimie et Médecine Moléculaire, Faculté de Médecine, Université de Montréal, Montréal, QC H3C 3J7, Canada. [4]Department of Biochemistry and Biophysics, Stockholm University, Stockholm SE-106 91, Sweden. [5]Faculty of Pharmaceutical Sciences, University of British Columbia, Vancouver V6T 1Z3, Canada. [6]These authors contributed equally: Brent D. G. Page, Mikael Altun. ✉e-mail: nicholas.valerie@ki.se

small molecules to a protein typically confers increased stability towards denaturation and proteolysis by preventing protein unfolding[9], but this is difficult to observe in an unperturbed cellular environment. Protein turnover can also be increased by facilitating the availability to the cellular proteolysis machinery[10]. The protein quality control system regulates general proteostasis, as well as the timely destruction of misfolded proteins[11]. Conditional molecular biosensors[12] and destabilizing domains[13] can exploit these regulatory pathways by rapidly accumulating in response to ligand binding, which decreases turnover rate of the protein via the ubiquitin-proteasome and/or autophagy systems by stabilizing a partially folded intermediate[14]. These approaches have employed engineered, destabilized missense variants and degron tags, which act as accelerants for protein turnover, to shed light on metabolic dynamics[12] and protein function[15] under physiological settings.

Here, we show that known destabilizing missense mutants of notable preclinical and clinical drug targets (MTH1, NUDT15, PARP1, DHFR, OGG1, PARP2) can function as stability-dependent biosensors that enable comprehensive interrogation of drug-target interactions. These drug biosensors are conditionally stabilized by the presence of a binding ligand, resulting in their increased abundance, and can either be engineered into cells or may already be present as naturally occurring mutations. We find that amenable mutants align with biophysical measures of stability, may be imputed from available structural information, and further be applied to relevant paralogs to expand the pool of drug biosensing mutants – as exemplified by alanine scanning of leucine residues in the PARP1 helical domain (HD) and transference of responsive PARP1 destabilization to an analogous residue on PARP2.

This simple readout of drug binding can then be readily combined with downstream pharmacological events in a single experimental interface, which we have termed cellular target engagement by accumulation of mutant (CeTEAM). We demonstrate this proof-of-concept by first uncoupling target binding from divergent cellular activities of MTH1 inhibitors utilizing a V5-G48E missense mutant. Next, we repurpose the NUDT15 R139C variant, an established prognostic factor of thiopurine sensitivity, as an HA-tagged biosensor to detect thiopurine species modulating NUDT15 activity. Finally, we profile the live-cell dynamics of PARP1 binding and DNA trapping by PARP inhibitors (PARPi) with a synthetic mutant of PARP1, L713F-GFP. Luciferase-coupled PARP1 biosensors also enable screening for PARP1 binders at scale and successfully identified >90% of cell-active PARP1i, as well as other chemical modulators of PARP1 stability. PARP1-derived drug biosensors also permit multimodal ex vivo analysis of drug-target engagement and non-invasive detection of drug binding in live animals, thereby demonstrating the translatability to in vivo applications. We envision that CeTEAM will be a powerful tool that enables holistic characterization of cellular drug-target engagement by linking drug binding to phenotypic events.

## Results

### Known missense mutants function as stability-dependent drug biosensors

Structural destabilization of proteins is a common outcome of missense mutations observed across the proteome[16–18]. Based on this principle, there have been many successful examples of engineered biosensors that are conditionally stabilized by cognate binding ligands, thereby leading to their rapid accumulation in cellular environments[12–14]. We asked if this phenomenon was common to naturally occurring and synthetically generated missense variants of relevant drug targets. When we introduced known destabilized variants of human MTH1 (G48E)[19], NUDT15 (R139C)[20], or PARP1 (L713F)[21] into cells under a doxycycline-inducible promoter, their low abundance was rescued by proteasome inhibitors and facilitated by specific fusion tag detection, confirming their rapid turnover (Supplementary

Fig. 1, Supplementary Discussion). Interestingly, this stabilization could be recapitulated by bona fide inhibitors in a time- and dose-dependent manner unrelated to their expression, while known inactive molecules could not (i.e., iniparib; Fig. 1a–c, Supplementary Fig. 2). Notably, NUDT15 R139C was also stabilized following exposure to the nucleoside analog drug, thioguanine, a prodrug metabolized to a known NUDT15 substrate in cells (Fig. 1b)[20]. We observed similar effects with exogenously expressed variants of cancer targets DHFR[14] (P67L; Supplementary Fig. 3) and OGG1[22] (R229Q; Supplementary Fig. 4a and b). OGG1 R229Q is also present as a biallelic mutation in KG-1 leukemia cells[22] and was stabilized after addition of reported OGG1 inhibitors, similarly to exogenous mutant (Supplementary Fig. 4c–e)[23,24]. Collectively, this implied that mutant protein accumulation was driven by ligand-induced stabilization, not feedback regulation. We then reasoned that this phenomenon could be generally adapted to monitor cellular drug-target engagement and related phenotypes in a single assay, which we call cellular target engagement by accumulation of mutant (CeTEAM; Fig. 1d). Such an approach could deconvolute drug binding events and resultant phenotypic changes to provide novel insights to drug mechanism-of-action.

### Definition and expansion of amenable PARP1/2 mutants

First, we wanted to establish if CeTEAM-compatible mutations could be rationally identified, as opposed to discovered serendipitously. The PARP1 HD domain consists of an α-helical bundle with several leucine residues directed towards the hydrophobic core (Fig. 2a)[21]. Destabilization of the HD domain is a critical allosteric change for PARP1 enzymatic function and contributes to DNA retention by PARP inhibitors (PARPi)[25,26]. Earlier work demonstrated that mutations of HD leucine residues to alanine generally destabilized PARP1 to thermal denaturation, including L713[21]. We then generated the same L698A, L701A, L698A/L701A, L765A, and L768A mutants, expressed them in cells with a C-terminal eGFP tag, and added the PARPi, veliparib, to determine the effect on variant stabilization and accumulation compared to WT and L713F (Fig. 2b and c). Benchmarking to L713F, L765A, L768A, and L698A/L701A had comparable, dose-dependent accumulation after veliparib treatment, while L698A and L701A were like WT PARP1.

We then asked if an HD mutant's amenability to CeTEAM correlated with previously reported thermal stability changes for full-length PARP1[21]. L765A and L768A mutants had larger thermal shifts (like L713A), while L698A and L701A were more like WT (Fig. 2d). Thus, leucine residues proximal to the hydrophobic core have a greater contribution to stability than those situated on the periphery (L698A and L701A; Fig. 2a). Cross-examination of the datasets showed that these differences translated well to the biosensing ability of a given HD mutant (Fig. 2d). This was particularly clear for the L698A/L701A double mutant, where the combination afforded an additive effect on CeTEAM suitability, akin to the destabilization seen in vitro (Fig. 2c, d). Thus, suitable CeTEAM mutants can be logically identified from available biophysical and structural information. Likewise, we then asked if the other functional CeTEAM HD mutants would have similar biosensing $EC_{50}$ values as L713F, as they destabilize the same region. To compare, we performed live-cell microscopy of GFP intensity following a dose-response with veliparib and 3-aminobenzamide (3-AB), an earlier and less potent PARPi, to compare the apparent stabilization $EC_{50}$ values (Fig. 2e, Supplementary Fig. 5a). Nuclei were identified using the cell-permeable DNA stain, Hoechst 33342. Indeed, drug treatment resulted in sigmoidal saturation curves for each mutant that superimposed well for both PARPi tested, albeit with differences in signal intensity and dynamic range. Remarkably, biosensor accumulation was discernable at both low-nanomolar (veliparib) and near-millimolar (3-AB) range. Therefore, mutations in a similar protein region appear to yield similar biosensing ability.

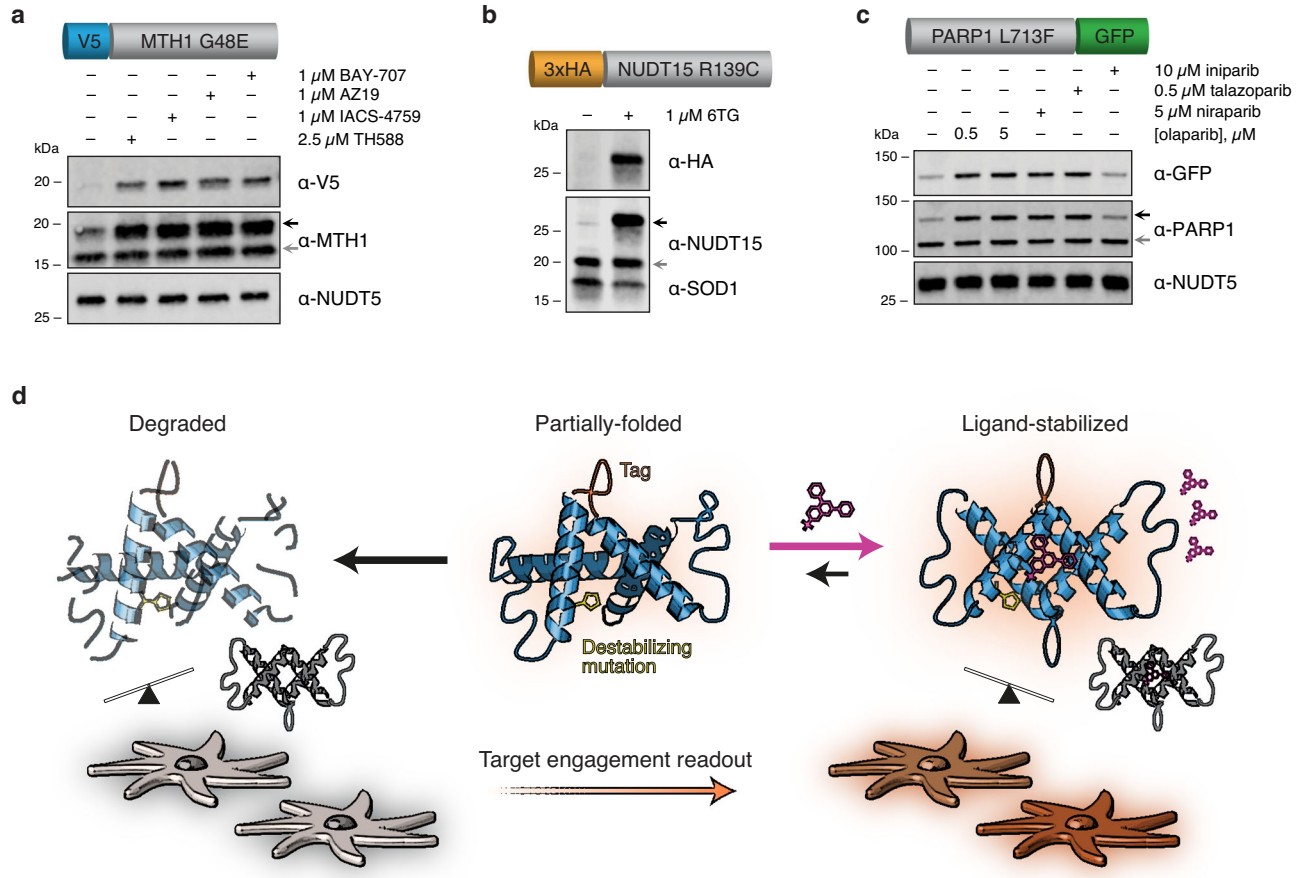

**Fig. 1 | CeTEAM is predicated on stability-dependent biosensors to measure drug binding. a** U-2 OS V5-MTH1 G48E cells were treated with the indicated MTH1 inhibitors for 24 hours. **b** HCT116 3-6 3xHA-NUDT15 R139C cells were incubated with the indicated molecules for 72 hours. **c** U-2 OS PARP1 L713F-GFP cells were treated with PARP inhibitors for 24 hours. Biosensors were pre-induced with doxycycline, and all blots are representative from two independent experiments. **d** A schematic description of CeTEAM. Stability-dependent drug biosensors (blue) containing a destabilizing mutation (yellow) accumulate in the presence of binding ligand (pink) and detection can be facilitated by protein fusion tags (orange) to measure drug-target engagement. The presence of endogenous target protein (gray) and physiological conditions enable phenotypic multiplexing and discerning of on- from off-target effects of test compounds.

Following exploration of intra-protein CeTEAM suitability with PARP1, we then asked if destabilization can be transferred to analogous residues in related protein family members. PARP1 and PARP2 share high structural homology within the catalytic domain, which makes the development of selective PARPi challenging. This includes the HD domain, where L713 in PARP1 corresponds to L269 in PARP2 (Fig. 2f) and similarly destabilizes it[27]. We generated a PARP2 drug biosensor in cells comprising a L269A mutation and a C-terminal eGFP tag, then determined its stabilization amenability by PARPi (Fig. 2g, h, Supplementary Fig. 5b). Like PARP1 L713F, there was robust stabilization by bona fide PARPi but not with iniparib, a now debunked PARPi[5,28], which was more apparent by an extended dose-response with live-cell fluorescence microscopy comparing veliparib, 3-AB, and iniparib (Fig. 2i). The biosensing dynamic range of PARP2 L269A and PARP1 L713F was comparable, likely reflecting both the similar affinity of most PARPi for PARP1/2[29] and ligand-stabilizing potential (Fig. 2e, i). Thus, it is feasible to expand the CeTEAM repertoire by transferring destabilizing mutations to close paralogs.

### Exploring divergent activities of MTH1 inhibitors with a G48E biosensor

As target engagement can readily be discerned under physiological conditions, we then proceeded to explore pharmacological insights afforded by this approach within the context of the respective target proteins. MTH1 (NUDT1) is a sanitizer of the oxidized nucleotide pool that initially garnered immense interest as an oncology drug target[30-32]. Subsequent investigations have highlighted potent and structurally diverse MTH1 inhibitors (MTH1i) that engage MTH1 in cells but cannot reproduce the anti-cancer activity of earlier molecules (Fig. 3a)[33-35]. To reconcile these differences, we utilized an unstable MTH1 G48E variant that binds to and is stabilized by MTH1i similarly to WT[19]. We confirmed that TH588, AZ19, IACS-4759, and BAY-707 all induced accumulation of a V5-MTH1 G48E drug biosensor in cells and, to a lesser extent, WT protein after 24 hours by western blot (Fig. 1a, Fig. 3b). We also simultaneously probed for other markers associated with MTH1i cellular activities – specifically, mitotic progression (phospho-histone H3 [pHH3] Ser10) and DNA damage (γH2A.X) have been linked to TH588 mechanism-of-action[36,37]. Notably, only TH588 induced DNA damage and mitotic arrest despite similar stabilization of V5-G48E by all tested molecules (Fig. 3b).

For further insight, we performed flow cytometry with clonal V5-G48E cells to simultaneously track V5 and pHH3 Ser10 signals, as well as DNA content by Hoechst 33342 staining, after treatment with TH588 or AZ19 (an MTH1i with no reported cytotoxic activity[35], Fig. 3c). Clonal V5-G48E cells were utilized to maximize robustness and uniformity when correlating to MTH1i phenotypic events (Supplementary Fig. 6a).

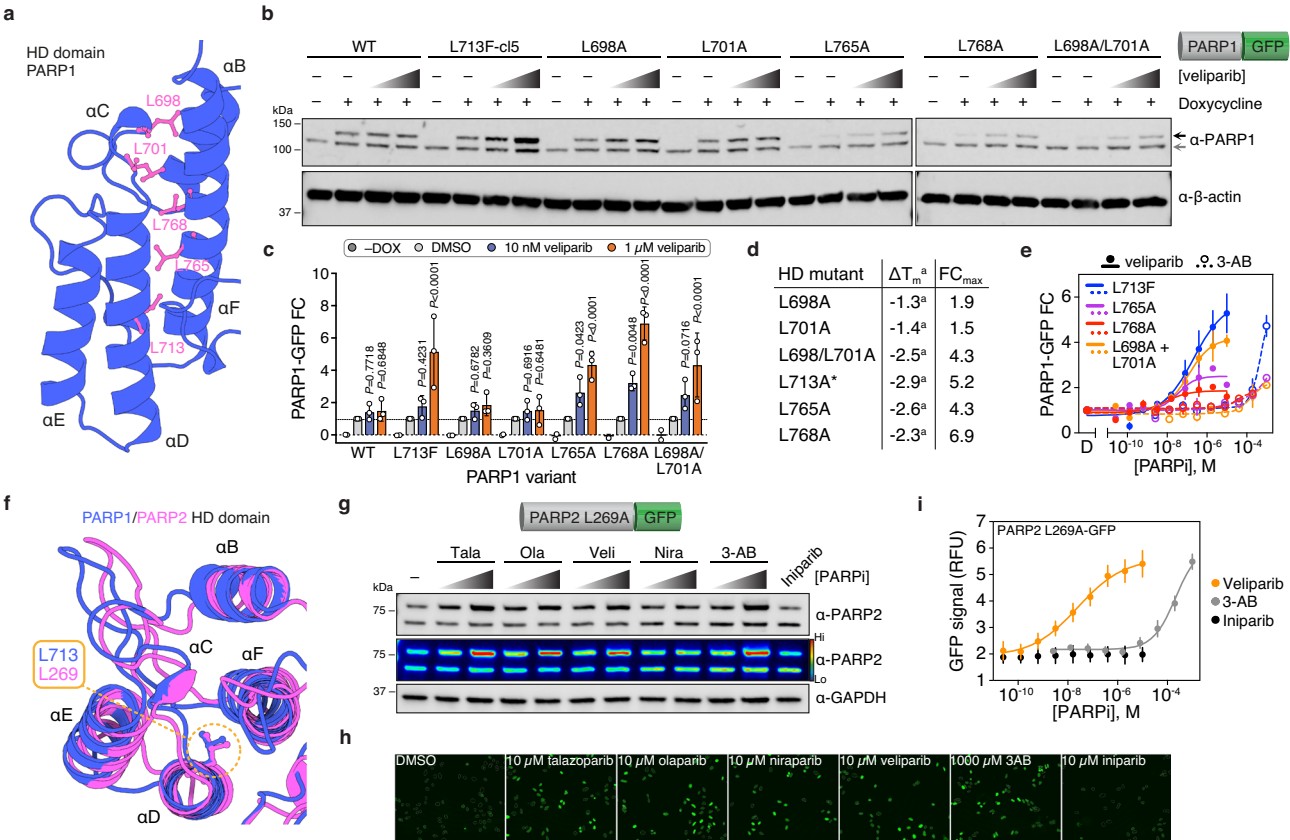

**Fig. 2 | Delineation and expansion of CeTEAM-permissive mutations with PARP1/2. a** Leucine residues of interest (magenta) within the PARP1 HD domain (PDBID: 7KK2, made with Protein Imager[93]). **b** A representative western blot of inducible C-terminal GFP-tagged PARP1 variants in U-2 OS cells incubated with DMSO, 10 nM, or 1 μM veliparib for 24 hours. Black arrow – PARP1-GFP; gray arrow – endogenous PARP1. **c** Quantitation of PARP1-GFP variant fold change relative to DMSO control (light gray) following 10 nM (blue) or 1 μM veliparib (orange; related to **a**). GFP abundance normalized to β-actin and no (–) DOX controls are shown for each variant. Means of $n = 3 \pm$ SD. Means of $n = 2$ shown for –DOX controls. P values shown for two-way ANOVA with multiple comparisons to each DMSO control (Dunnett's test; F [DFn, DFd]: $F_{Interaction}$ [12, 42] = 4.899, $F_{Row\ Factor}$ [6, 42] = 10.15, $F_{Column\ Factor}$ [2, 48] = 52.01). **d** Comparison of HD mutant thermal stability changes from Langelier et al. to fold change (over DMSO control) after 1 μM veliparib treatment (from **c**). **a** – denotes reported values from Langelier et al., * – thermal

stability change reported for L713A. **e** Live-cell fluorescence fold change of functional PARP1-GFP CeTEAM variants with veliparib (solid) or 3-AB (open/dashed) dose response after 24 hours. Means shown $\pm$ SEM (for $n_{L713F-v}$: 4, $n_{L713F-3AB}$: 3); means and range for all others ($n = 2$). **f** Overlay of PARP1 (blue, PDBID: 7KK2) and PARP2 (magenta, PDBID: 3KCZ) HD domains with L713/L269 denoted (made with Protein Imager[93]). **g** A representative western blot (from $n = 2$) demonstrating stabilization of constitutive PARP2 L269A-GFP in U-2 OS cells by various PARPi after 24 hours (3-AB – 100 μM and 1 mM, iniparib – 20 μM, all others – 10 nM and 1 μM). A psuedocolor density depiction in RFU is also shown. **h** Example GFP fluorescence micrographs of PARP2 L269A-GFP after 24-hr PARPi treatment. Nuclei are demarcated by outlines and scale bar = 100 μm. **i** Live-cell PARP2 L269A-GFP fluorescence following 24-hr dose-response with either veliparib (orange), 3-AB (gray), or iniparib (black). Means of $n = 5 \pm$ SEM. FC fold change, RFU relative fluorescence units.

Before flow cytometry analysis, we sought to define the ranges of V5-G48E stabilization by both MTH1i to enable meaningful comparisons of downstream pharmacology. We then performed a dose-response with both molecules to chart biosensor saturation by western blot – where AZ19 yielded a slightly better stabilization $EC_{50}$ (2 nM vs 15 nM for TH588; Fig. 3d, Supplementary Fig. 6b). These values were similar to those obtained previously by CETSA[35]. V5-MTH1 WT and endogenous MTH1 also had slight but discernable increases in abundance at higher drug doses (Supplementary Fig. 6c, d). We then interpolated points on both binding curves that were grouped into three V5-G48E occupancy designations: pre-saturation (partial target occupancy), saturation (maximum target occupancy), and literature (supersaturated target occupancy [i.e., a cell-based assay concentration often used in the scientific literature[31,35]]). By using apparent occupancy instead of MTH1i concentration, we can make direct comparisons between a molecule's phenotypic effects in relation to target binding. Applying the occupancy-designated MTH1i concentrations, we recapitulated the same saturation trend by flow cytometry (Fig. 3e). However, TH588 exclusively showed an enrichment in pHH3 Ser10 and G2-

phase cells consistent with mitotic delay at supersaturated occupancy (10 μM) in both G48E (Fig. 3e, f) and WT cells (Supplementary Fig. 6d), while at the saturation point (150 nM) these phenotypes were absent. In other words, we see that TH588-dependent gross phenotypic perturbations occur well beyond MTH1 saturation.

We then wanted to understand the interrelatedness of MTH1i target binding and the observed mitotic arrest seen in TH588-treated cells. To this end, pHH3 Ser10 and Hoechst intensities were visualized in the context of high V5 signal, arbitrarily classified as greater or equal to the top 2% of the DMSO control intensity, in cells treated with 10 μM MTH1i (Supplementary Fig. 6e). While V5-enriched cells in the AZ19 samples reflected the cell cycle distribution of their respective general populations, TH588-stabilized V5-G48E cells were overwhelmingly enriched in mitosis (1.30% vs 0.70% and 34.8% vs 10.8% pHH3 Ser10+/4 N DNA, respectively), despite AZ19 treatment yielding twice as many V5 high cells (Supplementary Fig. 6f and g). Thus, mitotic delay phenotypes are overrepresented in cells with high TH588 exposure but not AZ19 – in line with previous observations regarding TH588-mediated toxicity[35–37]. One plausible explanation for this

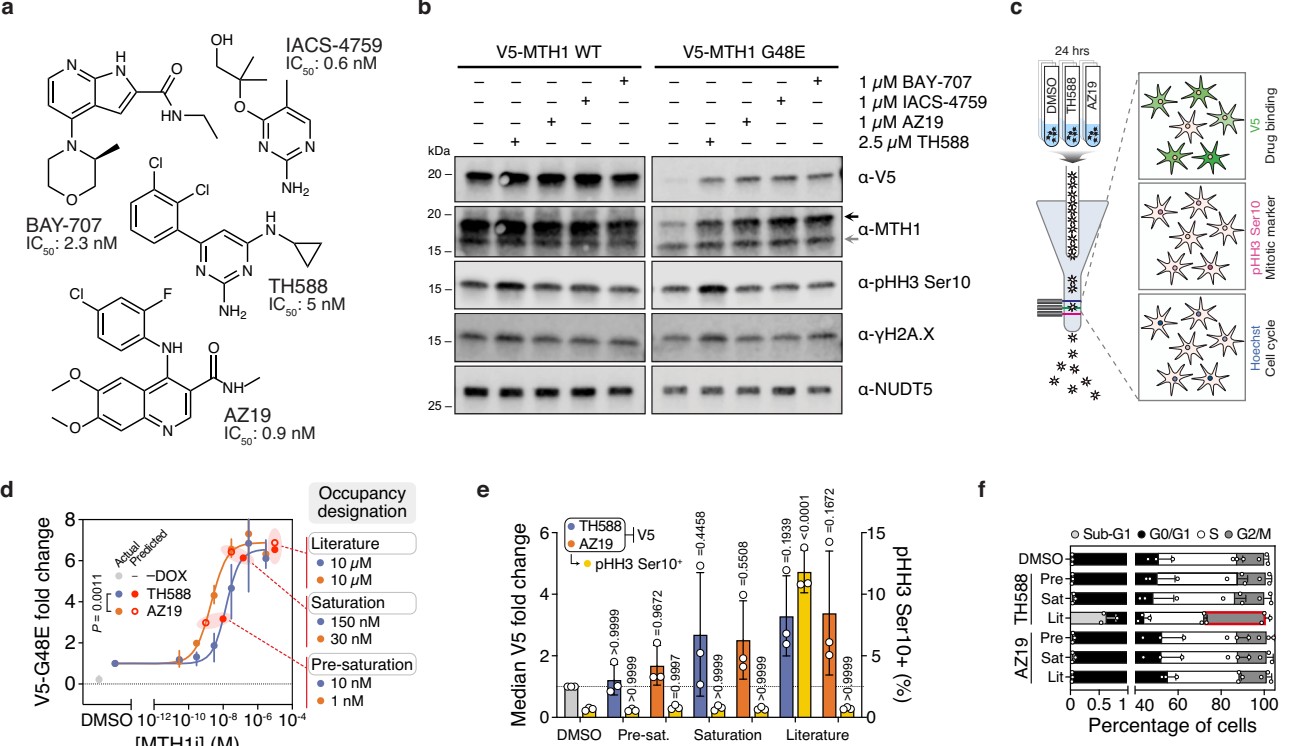

**Fig. 3 | Parsing differential pharmacology of equipotent MTH1 inhibitors with a V5-G48E drug biosensor. a** MTH1 inhibitors tested and their reported biochemical IC$_{50}$ values[31,33–35]. **b** A representative blot ($n = 2$) of induced V5-MTH1 WT or V5-MTH1 G48E in U-2 OS cells following incubated with DMSO, 2.5 μM TH588, 1 μM AZ19, 1 μM IACS-4759, or 1 μM BAY-707 for 24 hours. Black arrow – V5 MTH1; gray arrow – endogenous, WT MTH1. **c** A schematic depicting multiparametric CeTEAM analysis by flow cytometry of TH588 and AZ19 pharmacology by V5 (drug binding, AF-488), pHH3 Ser10 (mitotic marker, AF-647), and Hoechst (cell cycle) readouts using V5-MTH1 G48E clone 6 cells. **d** V5-G48E saturation profiles of TH588 (blue) and AZ19 (orange) after 24 hours by western blot. Mean of $n = 3 \pm$ range. Comparison made by two-sided extra sum-of-squares F Test (F [DFn, DFd] = 12.67 [1, 36]). Interpolated data points representing occupancy designations are shown in red and defined adjacent to data plots. **e** Median V5-G48E fold change (left axis; TH588 – blue, AZ19 – orange) and percent pHH3 Ser10$^+$ cells (right axis; yellow) of described occupancy designations by flow cytometry (mean of $n = 3 \pm$ SD). Reference line at V5 fold change = 1. P values shown for one-way ANOVA with multiple comparisons to the DMSO control (Dunnett's test; F$_{HH3}$ [DFn, DFd] = 122.3 [6, 14], F$_{V5}$ [DFn, DFd] = 1.625 [6, 14]). **f** Proportion of cells in sub-G1 (light gray), G0/G1 (black), S (white), and G2/M (dark gray) phases by Hoechst intensity. Mean of $n = 3 \pm$ SD. Red highlight – $P < 0.05$. P$_{Pre,TH588,SubG1} > 0.9999$, P$_{Pre,TH588,G0/G1} = 0.9998$, P$_{Pre,TH588,S} > 0.9999$, P$_{Pre,TH588,G2/M} = 0.9997$, P$_{Sat,TH588,SubG1} > 0.9999$, P$_{Sat,TH588,G0/G1} = 0.9251$, P$_{Sat,TH588,S} = 0.9978$, P$_{Sat,TH588,G2/M} = 0.9963$, P$_{Lit,TH588,SubG1} > 0.9999$, P$_{Lit,TH588,G0/G1} = 0.1700$, P$_{Lit,TH588,S} = 0.0934$, P$_{Lit,TH588,G2/M} = 0.0001$, P$_{Pre,AZ19,SubG1} > 0.9999$, P$_{Pre,AZ19,G0/G1} = 0.9992$, P$_{Pre,AZ19,S} = 0.9948$, P$_{Pre,AZ19,G2/M} = 0.9950$, P$_{Sat,AZ19,SubG1} > 0.9999$, P$_{Sat,AZ19,G0/G1} = 0.9997$, P$_{Sat,AZ19,S} = 0.9997$, P$_{Sat,AZ19,G2/M} = 0.9995$, P$_{Lit,AZ19,SubG1} > 0.9999$, P$_{Lit,AZ19,G0/G1} = 0.7657$, P$_{Lit,AZ19,S} = 0.8923$, P$_{Lit,AZ19,G2/M} > 0.9999$ by ordinary two-way ANOVA with multiple comparisons to the DMSO control (Dunnett's test; F [DFn, DFd]: F$_{Interaction}$ [18, 56] = 3.103, F$_{Row Factor}$ [6, 56] = 0.01093, F$_{Column Factor}$ [3, 56] = 552.0).

phenomenon is the activation of the mitotic surveillance pathway, a USP28- and p53-mediated G1 checkpoint preventing cell cycle reentry after prolonged mitosis[37]. Interestingly, when we compared 10 μM TH588 with a slightly lower, cell-active dose (2.5 μM), we see that cells can still exit mitosis and arrest in the next G1-phase at lower doses but fail to do so at higher concentrations, in support of the mitotic surveillance checkpoint hypothesis (Supplementary Fig. 6h, i). Collectively, our data argues that TH588 cytotoxicity is independent of MTH1 binding, and CeTEAM can effectively parse divergent biological activities among ligands exhibiting comparable intracellular target binding.

**Leveraging the NUDT15 R139C variant to detect thiopurines in cellulo**

We then turned our attention to the NUDIX hydrolase, NUDT15, which is implicated as a determinant of chemotherapeutic drug efficacy. Specifically, NUDT15 deactivates the nucleoside analog drug, thiopurine, by hydrolyzing the active triphosphates and limiting DNA damage-induced toxicity[20,38,39]. Several destabilizing variants of NUDT15 have been associated with clinical thiopurine intolerance, including R139C[38]. The R139C variant has a rapid turnover in cells but

still binds substrates and NUDT15 inhibitors (NUDT15i) similarly to wild-type protein in vitro[20,40]. When we expressed R139C in cells as an HA-fusion, its protein abundance was low but robustly accumulated within 24 hours of 6-thioguanine (6TG) exposure and was accompanied by expected DNA damage (γH2A.X) at the 72-hour mark, as 6TG-mediated genotoxicity manifests after multiple rounds of DNA replication (Fig. 1b, Supplementary Fig. 1c, Fig. 4a)[20]. We could also confirm that NUDT15 activity can be blocked by multiple thiopurine metabolites to varying degrees – most notably by diphosphate species but not methylated counterparts (Fig. 4b), in agreement with the successful development of thiopurine-mimetic NUDT15i[41]. Thus, thiopurines can bind, stabilize, and drive the intracellular accumulation of NUDT15 R139C.

To explore this phenomenon systematically in cells, we further derivatized the reported NUDT15i, NSC56456 (TH7410)[41], into an inactive analog (TH8228; Supplementary Fig. 7), as well as a potential 6TG prodrug (TH8234; Fig. 4c, Supplementary Fig. 8). We resolved a 1.8 Å co-crystal structure and see that NSC56456 binds the NUDT15 active site similarly to 6-thio-GMP (Supplementary Fig. 9a–d, Supplementary Table 1). Thus, we anticipated that the TH8228 methylsulfanyl moiety should discourage binding due to steric clashing within an

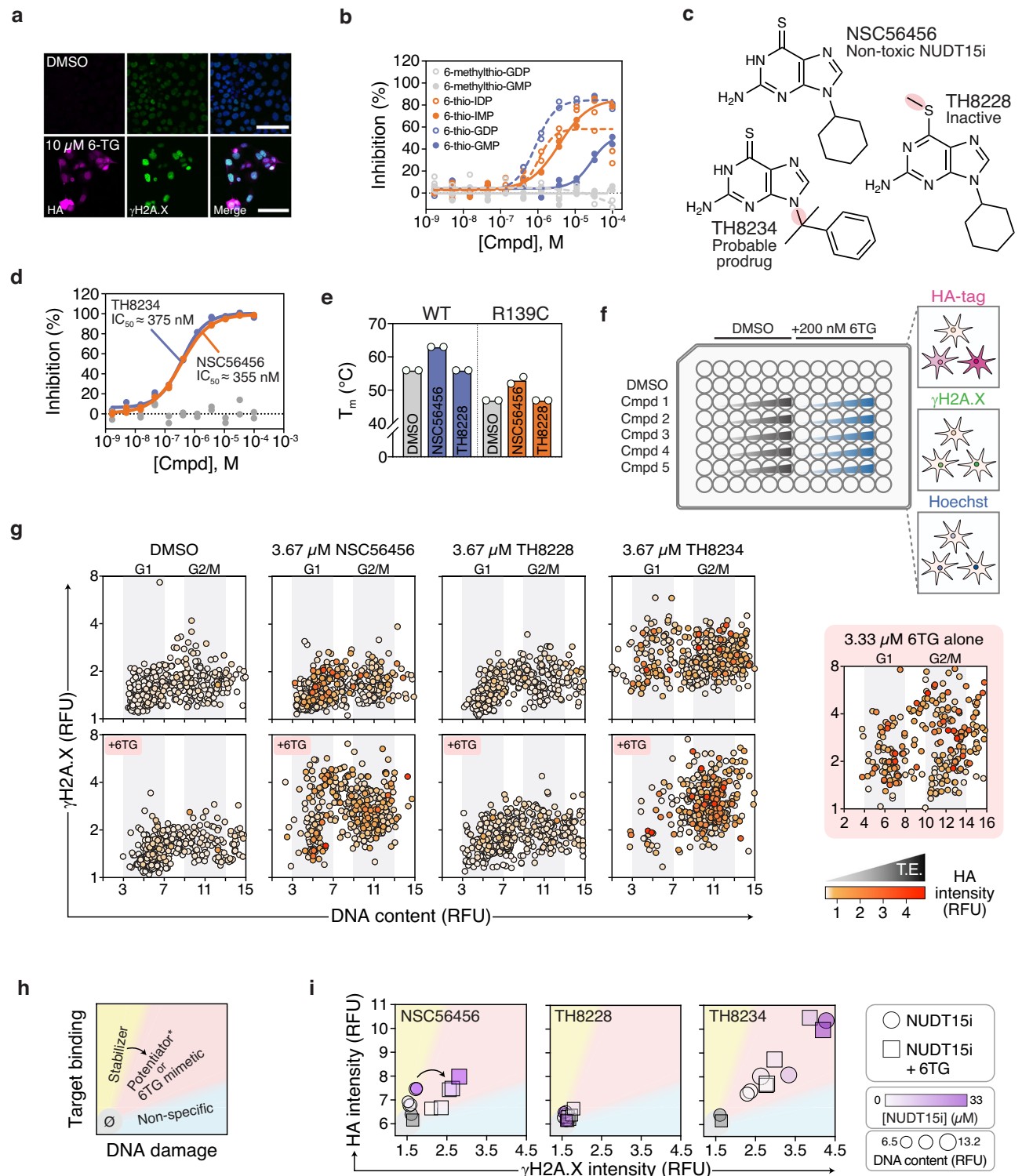

adjacent hydrophobic pocket, similar to methylated metabolites tested earlier (Supplementary Fig. 9b). Indeed, while NSC56456 and TH8234 similarly inhibited NUDT15 (IC$_{50}$ = 383 and 375 nM, respectively), TH8228 was completely inactive (Fig. 4d). NSC56456 also stabilized wild-type NUDT15 by DSF assay, as before[41], whereas TH8228 did not (Fig. 4e, Supplementary Fig. 9e, f). Likewise, we also confirmed that NUDT15 R139C had a significantly lower melting temperature than WT (T$_m$; 47 versus 56 °C, respectively[20]) but was still comparably stabilized by NSC56456 (ΔT$_m$ = 7 and ΔT$_m$ = 6 °C, respectively) but not by TH8228. These data reinforce that NSC56456 and TH8234 bind and

inhibit both NUDT15 proteoforms, while TH8228 is an appropriate negative control.

Triaging of bona fide inhibitors can be complicated by assuming on-target binding/inhibition equates to intended phenotypic responses and vice versa. We hypothesized that TH8234 may convert to free 6TG in cellulo via N-dealkylation of the α,α-dimethylbenzyl group[42], which could lead to undesirable toxicity. We first confirmed that these compounds could dose-dependently stabilize intracellular HA-R139C, while TH8234 also gave a small, but statistically insignificant increase in WT NUDT15 abundance (Supplementary Fig. 10a, b). Previous work

**Fig. 4 | Leveraging the NUDT15 pharmacogenetic variant, R139C, to decipher thiopurine pharmacology in cellulo. a** Representative microscopy images (from $n = 2$ independent experiments) of doxycycline-induced HCT116 3-6 3xHA-NUDT15 R139C cells treated with DMSO or 10 μM 6TG for 72 hours and stained with indicated markers. Hoechst staining is shown in the merged image. Scale bar=200 μm. **b** NUDT15 inhibition by thiopurine metabolites ($n = 2$ with lines of best fit). **c** Structures of NSC56456, TH8234, and TH8228 with moieties of interest highlighted in red. **d** NUDT15 inhibition by TH8228 (gray), NSC56456 (batch ID: BV122529; orange), and TH8234 (blue). $n = 2$ with lines of best fit. **e** Melting temperatures of NUDT15 WT (blue) and R139C (orange) with 50 μM NUDT15i by DSF assay compared to DMSO (gray). Means of $n = 2$. **f** A schematic depicting a high-content microscopy assay for simultaneous detection of target engagement (HA) and phenotypes (DNA damage response – γH2A.X, cell cycle – Hoechst) of potential NUDT15 inhibitors -/+ low-dose 6TG. **g, h** Representative per-cell three-dimensional analysis of γH2A.X (y-axis), Hoechst (x-axis), and HA intensities (white-orange-red gradient) following treatment with DMSO, 3.67 μM NSC56456, 3.67 μM TH8228, or 3.67 μM TH8234 ± 200 nM 6TG and compared to 3.33 μM 6TG alone. $n = 500$ cells per condition, except $n_{6TG} = 399$. **i** Binning of NUDT15i into non-responder/∅ (gray), stabilizer (yellow), potentiator or 6TG mimetic (red; NUDT15 binding-related 6TG potentiation), and non-specific (blue; NUDT15 binding-independent DNA damage) based on HA-R139C intensity and DNA damage induction. Stabilizers may reclassify to potentiators in the presence of 6TG. **j** Per-drug analysis of median HA (y-axis), γH2A.X (x-axis), and Hoechst intensities (symbol size) for NSC56456, TH8228, and TH8234 at multiple concentrations (white-magenta gradient) either alone (circles) or combined with 6TG (squares) and compared to DMSO (gray). RFU relative fluorescence units.

has demonstrated that genetic or pharmacological ablation of NUDT15 activity sensitizes cells to thiopurines by approximately 10-fold[20,38,39]. We, therefore, established a high-content immunofluorescence microscopy CeTEAM workflow to profile these compounds alone and in combination with low-dose 6TG using HA-tagged NUDT15 R139C as a NUDT15i reporter (Fig. 4f). To follow 6TG-dependent genotoxicity in parallel, we prolonged the assay and measured γH2A.X and DNA content (cell cycle) readouts at 72 hours[20]. First, we confirmed that 200 nM 6TG alone neither increased HA-R139C and γH2A.X signals nor grossly affected the cell cycle (Supplementary Fig. 10c–e). By initial observation, NSC56456 both dose-dependently stabilized NUDT15 R139C and enhanced thioguanine-mediated DNA damage (Supplementary Fig. 10f–i). TH8228 affected neither R139C nor the 6TG-dependent DNA damage response. TH8234, meanwhile, stabilized R1–39C and induced a 6TG-like response without added 6TG.

To better understand NUDT15 binding and potentiation of 6TG-mediated toxicity, we mapped multiparametric CeTEAM data at the single cell level comparing the NUDT15i at 3.67 μM. As before, clonal selection enhanced the uniformity of biosensor responses (Supplementary Fig. 11a–d). This revealed that HA-R139C stabilization generally occurs independently of cell cycle phase, but as expected, a majority of γH2A.X-positive cells were stalled in G2-phase after high-dose 6TG (Fig. 4g). However, both markers were absent at the lower 200 nM dose. Combining low-dose 6TG with NUDT15i confirmed NSC56456-dependent potentiation of 6TG toxicity and TH8228 inactivity. TH8234, meanwhile, elicited a 6TG-like response without supplemental 6TG, comparable to an equivalent concentration of 6TG alone. 6TG supplementation potentiated TH8234 phenotypes, further arguing it transitions to a thioguanine-like metabolite (Fig. 4g, Supplementary Fig. 10h, i), although this was not confirmed empirically.

We reasoned that we could also triage NUDT15i based on HA-R139C stabilization and markers of 6TG potentiation. Using these readouts, we can categorize potential cell-active NUDT15i into five groups: non-responder, stabilizer, potentiator, 6TG-mimetic, or non-specific (Fig. 4h). Here, R139C-stabilized potentiation of 6TG toxicity indicates an actionable NUDT15i and R139C-impartial DNA damage would be considered NUDT15-independent activity. In this context, stabilizers can transition to potentiators upon addition of 6TG but may not always do so (e.g., if utilizing a ligand that purely affects target protein stability without inhibiting enzymatic activity). Likewise, NUDT15i acting as 6TG-mimetics would both stabilize HA-R139C and yield G2-phase DNA damage independently of supplemental thioguanine. Applying this logic, NSC56456 is clearly a stabilizer that also potentiates 6TG, TH8228 is a non-responder, and TH8234 behaves like a thiopurine pro-drug (Fig. 4i). Notably, TH8234 consistently stabilized HA-R139C better than NSC56456 in cells (akin to stabilization seen with 6TG treatment) despite equipotent biochemical $IC_{50}$ values. Collectively, the results underscore that thiopurine mimetics are putative chemical starting points for developing new NUDT15 probes[41]. More importantly, CeTEAM revealed that multiple thiopurine species can

effectively bind and stabilize the R139C pharmacogenetic variant in cells, which has some notable implications. First, while active site binding effectively corrects protein folding[40] and restores R139C abundance, its activity is still blocked, suggesting that the thiopurine sensitivity seen in these patients is also significantly driven by NUDT15 inhibition. Second, R139C activity can conceivably be restored by allosteric pharmacological chaperones, and a CeTEAM screening platform can facilitate their discovery.

## Biological validation of PARP1 L713F as a PARPi biosensor

As PARP inhibitors epitomize successful targeted therapies, we also further investigated the L713F variant for CeTEAM-based assays. PARP1 L713F is a synthetic gain-of-function mutant that increases flexibility between the HD and ART domains, conferring both instability and DNA-independent PARylation activity[21,43]. It was therefore relevant to ask if the L713F mutant is a capable surrogate to WT PARP1 for PARPi pharmacology. In line with their similar inhibition by PARPi[44], purified PARP1 WT and L713F catalytic domains were comparably stabilized by clinical-grade PARP inhibitors (PARPi) – with L713F having a decreased baseline melting temperature (41.9 versus 47.4 °C) – thereby supporting its amenability to CeTEAM (Supplementary Fig. 12a–c, Fig. 1c).

As part of its function, PARP1 is recruited to sites of DNA damage and orchestrates the DNA damage response (DDR) via catalysis of polyADP-ribose (PAR)[45]. We then asked if PARP1 L713F could bridge the biophysical detection of PARPi target engagement to DDR-related PARP1 biology. As previously reported[46], we saw that PARP1 L713F-GFP is still recruited to damaged DNA in response to laser microirradiation, albeit with slightly attenuated kinetics – similar to a previous report (Supplementary Fig. 12d, e)[46]. Fluorescence recovery after photobleaching (FRAP) experiments also suggested that PARP1 L713F has slightly less transient mobility than WT, presumably due to higher affinity for DNA (Supplementary Fig. 12f–j)[47]. We then determined how downstream DDR markers (PAR and γH2A.X) were affected following microirradiation in PARP1 WT or L713F cells pre-treated with PARPi for 1 or 24 hours (Supplementary Fig. 13a). L713F-GFP biosensor levels were significantly elevated after only one hour of PARPi (Supplementary Fig. 13b, c). In both untreated WT and L713F PARP1 cells, PAR signal elevated initially then returned to baseline following DNA damage, while γH2A.X temporally increased (Supplementary Fig. 13d, e). Pre-treatment with PARPi suppressed PAR formation with either PARP1 variant, whereas γH2A.X dynamics were unchanged. Two-dimensional analysis of GFP and PAR intensity in single cells revealed that drug-induced stabilization of L713F-GFP highly correlated with proximal markers of PARPi target engagement (PAR suppression; Supplementary Fig. 13f). Notably, basal PAR and γH2A.X were elevated in L713F cells compared to WT, as described previously[46], but obvious signs of toxicity were not observed (Supplementary Fig. 13g, Supplementary Discussion). These results suggest that PARP1 L713F-GFP faithfully reflects PARPi binding in a cellular context.

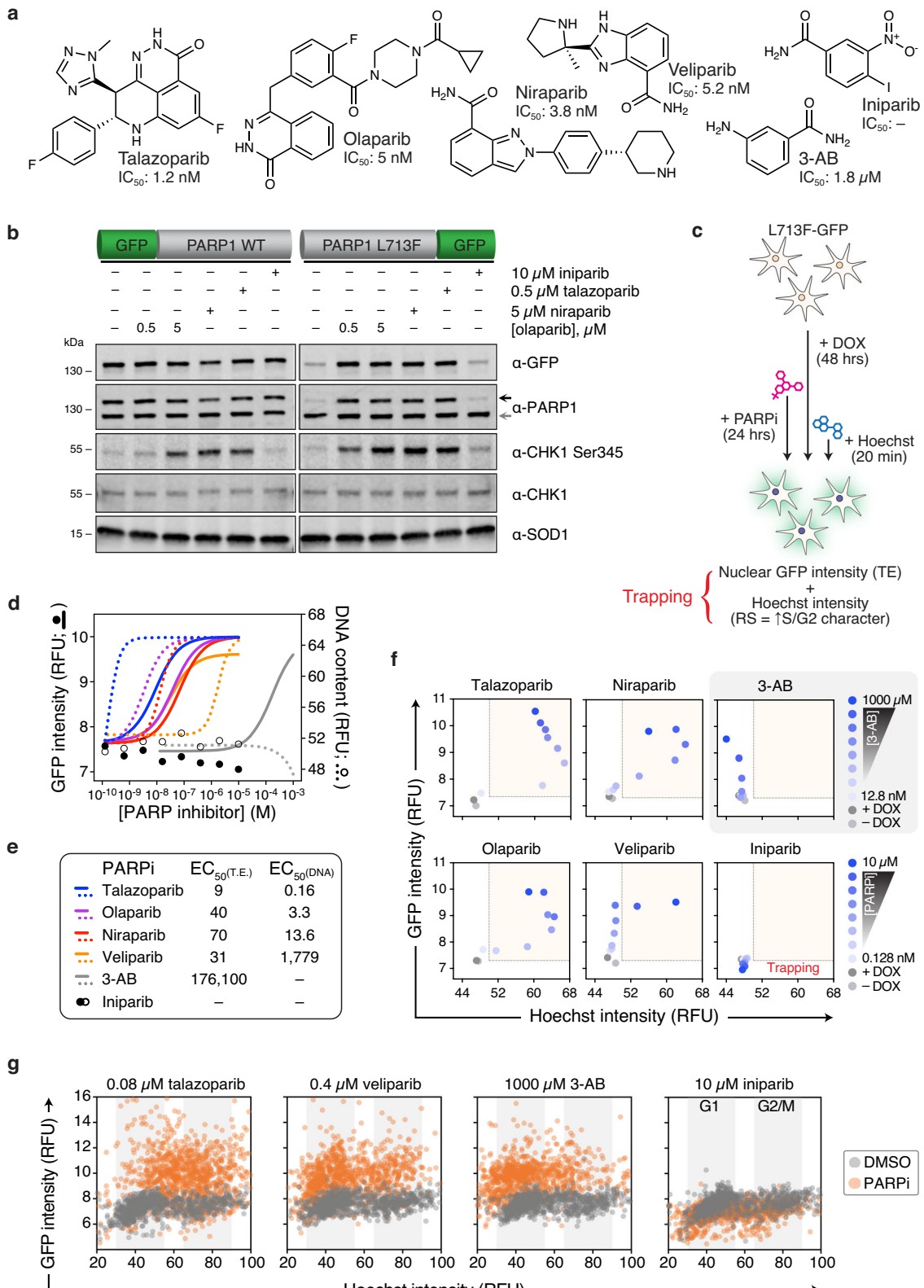

### Dynamic profiling of PARPi binding and DNA trapping in live cells

An underlying factor dictating clinical responses to some PARPi is the inherent ability to trap PARP onto DNA and induce replication stress in cancer cells, which can be independent of inhibition capacity[25,48–50]. However, the associated toxicity of trapping may be undesirable for other indications, such as neurodegenerative diseases, where

enzymatic blockade would suffice[51]. Known trapping PARPi caused significant replication stress after 24 hours, as evidenced by increased phospho-CHK1 and γH2A.X with PARPi treatment alone (Fig. 5a and b, Supplementary Fig. 13h-k), reflecting the decreased mobility of PARP1 elicited by some inhibitors[25]. Nonetheless, detailed characterization of trapping dynamics in relation to PARP1 binding has not been eluci-dated previously.

**Fig. 5 | Profiling of PARPi engagement and trapping in live cells with PARP1 L713F-GFP. a** Chemical structures and PARP1 inhibitory potencies of PARPi studied (SelleckChem and[94]). **b** A representative blot ($n = 2$) of induced GFP-PARP1 WT and L713F-GFP in U-2 OS cells treated with PARPi for 24 hours. Black arrow – GFP-tagged PARP1; gray arrow – endogenous PARP1. **c** Experimental schematic for live cell tracking of PARP1 target engagement (GFP) and PARPi-induced replication stress (cell cycle, Hoechst) by high-content microscopy. Trapping depends on PARP1 engagement and replication stress. **d** Curve fitting of median GFP (solid) and Hoechst (DNA content; open/dashed) intensities in live, PARP1 L713F-GFP clone 5 cells incubated with talazoparib (blue), olaparib (purple), niraparib (red), veliparib (orange), 3-AB (gray), or iniparib (black) for 24 hours following DOX induction. Means from $n = 2$. **e** Summary of observed L713F-GFP stabilization and median DNA content $EC_{50}$ values for tested PARPi (in nM). **f** Concentration-dependent dynamics of PARP1 target engagement and DNA trapping in live cells after PARPi. Median GFP (y-axis) and Hoechst (x-axis) intensities are shown. Representative of $n = 2$, replotted from **d**. Light gray circle – –DOX control; dark gray circle – +DOX control; blue gradient circles – PARPi concentration gradient (3-AB – 12.8 nM to 1 mM; all other PARPi – 0.128 nM to 10 μM), red areas – PARP trapping phenotype. **g** Representative single cell, 2D plots comparing GFP intensity (y-axis) and Hoechst intensity (x-axis) following DMSO (gray) and PARPi treatment (orange; replotted from **d**). Inferred G1 and G2/M cell cycle phases demarcated by gray columns. Overview data in f representative of $n = 500$ cells per group; individual cell plots in **g** are $n = 1000$ cells per group. RFU – relative fluorescence units.

To survey this in a larger cohort of PARPi, we first selected suitable clones, then employed CeTEAM with multiplexed live-cell fluorescent microscopy to concurrently track dose-dependent L713F-GFP accumulation and S/G2-phase shifts in DNA content with Hoechst 33342, which has previously been shown to be a capable surrogate for PARPi-induced replication stress (Fig. 5c, Supplementary Fig. 14a–d)[52]. In this sense, the designation of trapping would require both PARP1 engagement and a replication stress phenotype. All clinical PARPi effectively stabilized L713F-GFP but had differential trapping ability that mirrored previously reported inhibitory and trapping rankings (talazoparib > olaparib ≈ niraparib >> veliparib ∼ 3-AB/iniparib; Fig. 5d, e, and Supplementary Fig. 14e–i)[48,53]. Dose-dependent stabilization of L713F-GFP followed a sigmoidal saturation profile that permitted ranking of PARPi based on their observed stabilization $EC_{50}$s. Similar PARPi saturation profiles were also seen with the paralogous PARP2 L269A-GFP biosensor (Supplementary Fig. 14j). A two-dimensional analysis of these data revealed that trapping phenotypes were apparent only after detectable PARP1 binding, although the discrepancy between target engagement and trapping $EC_{50}$ values varied among inhibitors tested (Fig. 5f). Notably, talazoparib was the most potent binder and trapper followed by olaparib and niraparib. Expectedly, iniparib failed to induce biosensor accumulation[5,28]. While veliparib was equipotent to other clinical inhibitors for PARP1 binding, it was vastly inferior at trapping, as S/G2 shifts only materialized in the micromolar range. As with PARP2, 3-AB elicited accumulation in the near-millimolar range and also induced a G1-phase accumulation, as previously reported[54]. Accordingly, single cell evaluation underscored that cell populations treated with trapping PARPi are clearly and uniformly distinguished from control cells (Fig. 5g). This trapping trend was confirmed by S-phase enrichment of other DDR markers related to replication stress, phospho-CHK1 (pCHK1) Ser345 (Supplementary Fig. 15a–e) or γH2A.X (Supplementary Fig. 15f–j). Thus, CeTEAM may be an effective tool for triaging PARPi based on their PARP trapping potential.

## A PARP1 perturbagen screen enabled by a dual luciferase biosensor system

We found that PARP1 L713F is also amenable to nanoluciferase (nLuc) fusions (Supplementary Fig. 16a, b), which enabled robust and sensitive detection of PARPi binding with lysed (Supplementary Fig. 16c–e) or intact cells (Supplementary Fig. 16f-i). We then complemented the nLuc biosensor with an akaLuc reference, a red-shifted variant of firefly luciferase[55], to normalize PARP1 binding signals (Fig. 6a, Supplementary Fig. 17a and b). akaLuc activity was lost following detergent-mediated lysis (Supplementary Fig. 17c, d), so both luminescence readings were performed with live cells using a spiral averaging feature to account for uneven cell distribution in the wells. Despite clear spectral separation between nLuc and akaLuc (Supplementary Fig. 17e), we determined that sequential detection enabled optimal normalization of PARP1 target engagement and eliminated signal interference (Supplementary Fig. 17f–m). Harnessing the high

sensitivity of bioluminescence, PARP1 L713F-nLuc stabilization was detectable in as little as one hour after veliparib addition, while the dynamic range of the assay was maximal around 24 hours (Fig. 6b). Intriguingly, the apparent potency of treatment, expressed as observed $EC_{50}$ values, did not vary with either time (Fig. 6b) or biosensor abundance (Fig. 6c; Supplementary Fig. 17n), suggesting that this system is a robust methodology for rapidly evaluating cellular target engagement.

One appealing prospect of a luminescence-based CeTEAM detection platform is the possibility to seamlessly scale assays for high-throughput analyses. The PARP1 L713F-nLuc biosensor consistently scored favorable screening-related parameters in initial tests with multiple cell lines and expression systems (Supplementary Fig. 18a)[56]. To demonstrate this in practice, we performed a screen of ~1200 drug-like molecules at 10 μM with the L713F dual luminescence system (Fig. 6d, Supplementary Fig. 18b, Supplementary Data 1). The library consisted of clinical and preclinical small molecules within the Med-ChemExpress (MCE) Epigenetics and Selleck Nordic Oncology sets, including 57 compounds designated as PARP family inhibitors, of which 45 target PARP1 (Supplementary Data 1). On average, the dual luminescence set-up yielded a Z' of 0.29 and S/B of 5.6 (Supplementary Fig. 18c). The akaLuc readout proved critical to triage viability outliers (arbitrarily defined as >4 SDs from controls per plate), which may otherwise skew nLuc/akaLuc ratio linearity (Supplementary Fig. 18d). Aberrant upregulation of both readouts was common in the epigenetics-targeting library (plates 1-10), while general toxicity was more apparent in the oncology set (plates 10-15, Supplementary Fig. 18e, Supplementary Data 1). akaLuc triaging resulted in 840 compounds for analysis from the original 1187 that were reported as L713F-nLuc/akaLuc fold change over DMSO controls (Fig. 6d). Hits from the screen were defined as ≥2 standard deviations from the mean of all samples following $\log_2$ transformation, which improved normality of the dataset, to yield a total of 53 hits (Fig. 6e, f, Supplementary Fig. 18f). Most hits were positive (stabilizers; 47 compounds), although there were also a handful that decreased L713F-nLuc abundance (6 compounds). Unsurprisingly, many PARPi were positive hits (58% within library; Fig. 6g). The hit proportion increased when limiting the analysis to annotated PARP1i (73%) and further still when triaging to PARP1i with $IC_{50} < 1$ μM (92%). Follow-up of PARPi hits yielded a 100% confirmation at the original screening concentration (10 μM; Fig. 6h). Of the three PARP1i that did not qualify as hits, two are early generation inhibitors (DR2313 [$IC_{50}$: 200 nM][57] and PJ34 [$IC_{50}$: 110 nM[58], narrowly missed the akaLuc cutoff but the hydrochloride variant met this criterion]), and the other, EB-47, is a potent NAD-mimetic inhibitor of PARP1 (45 nM)[59]. Subsequent testing of EB-47 by dose-response experiments yielded no stabilization of L713F-nLuc despite previous evidence that it binds L713F (Supplementary Fig. 18g), perhaps reflecting poor cell permeability of this PARPi[47,60].

Additionally, several non-PARPi yielded a significant stabilization of L713F-nLuc, but only a handful were confirmed by follow-up analyses (29% of hits; Fig. 6f, h). Of the four compounds confirmed, two

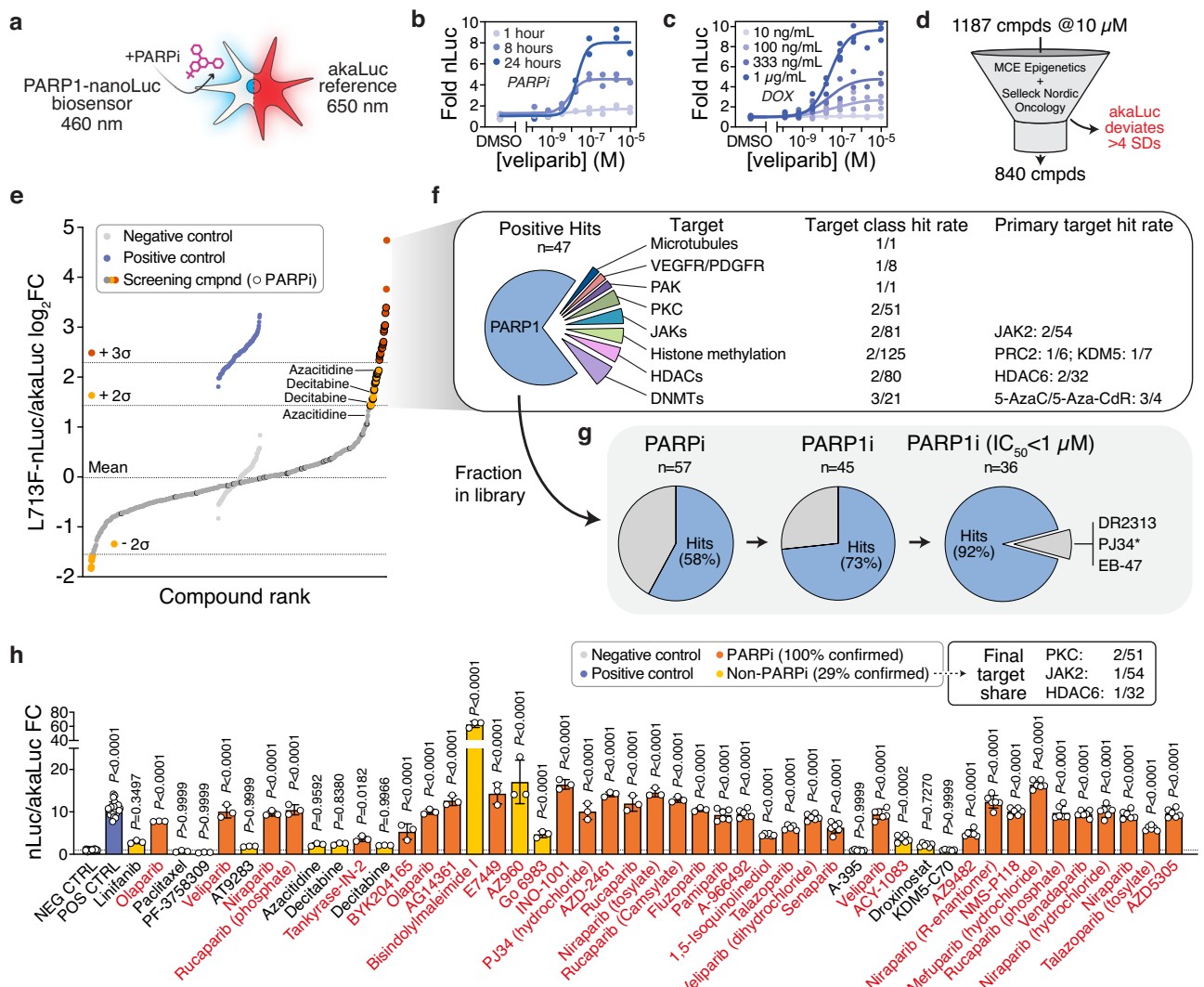

**Fig. 6 | A PARP1 biophysical perturbagen screen enabled by a L713F-nLuc dual luminescence assay format. a** The PARP1 L713F-nLuc biosensor (em: 460 nm) was paired with akaLuc (em: 650 nm) to enable sequential dual luciferase analyses. **b** Time-resolved detection of PARP1 L713F-nLuc stabilization following veliparib treatment and normalized to akaLuc signal. $n = 2$ with line of best fit shown. **c** Dose-dependent veliparib stabilization of different PARP1 L713F-nLuc abundances (DOX gradient) after 24 hours and normalized to akaLuc. $n = 2$ with line of best fit shown. **d** The MedChemExpress Epigenetics and Selleck Nordic Oncology libraries were screened (10 μM, 24 hours) with the L713F-nLuc/akaLuc system. Compounds was excluded if akaLuc intensity differed > 4 SDs from controls, leaving 840 compounds for further analysis. **e** Ranked, log2-transformed L713F-nLuc/akaLuc ratios from 840 screening compounds (dark gray). Negative (light gray, DMSO) and positive controls (blue, 10 μM veliparib) are shown for reference. Hits were defined as at least 2 (orange) or 3 standard deviations (σ, red) from the screening library mean. Annotated PARPi are indicated with black borders and trapping DNMT

compounds are labeled. **f** Detailed overview of positive screening hits ($n = 47$). Non-PARPi were triaged by target class, contextualized by hit rate within the general target class, and by anecdotally defined primary target/compound class. **g** Hit rates of PARPi within the screening library by increasing stringency (general PARPi → PARP1i → PARP1i [IC50 < 1 μM]) and numbers of qualifying compounds. Hit proportions are shown in blue, while non-hits are gray. * – PJ34 missed the akaLuc cutoff. **h** Hit confirmation of PARPi (orange) and non-PARPi (yellow) positive screening hits. Identical positive (blue) and negative controls (gray) are used from the screen, and means of $n = 24$ (negative, positive control), $n = 3$ (linifanib to fluzoparib), or $n = 6$ (pamiparib to AZD5305) data points are shown ± SD. Names of statistically significant compounds in red, and confirmed non-PARPi are summarized by primary target hit rate (final target share). P values are shown for one-way ANOVA analysis with comparisons to DMSO control (Dunnett's test; $F_{Treatment}$ [DFn, DFd] = 200.9 [48, 202]). FC fold change.

were broad spectrum PKC inhibitors (Bisindolylmaleimide I [Bim I, GF109203X] and Gö 6983), one a JAK2 inhibitor (AZ960), and one an HDAC6 inhibitor (ACY-1083). Notably, each hit was rare among molecules targeting the respective protein class in the screening library (i.e., PKC: 2/51, JAK2: 1/54, HDAC6: 1/32 compounds; Fig. 6h), suggesting that their stabilization of L713F-nLuc is related to specific chemotypes rather than the intended target. Although the DNMT trappers, decitabine and azacytidine, failed to meet confirmation significance thresholds, they were highly enriched in the general screen, while non-trapping DNMTi were not (3/4 and 0/17 instances, respectively; Fig. 6e, f, and h, Supplementary Data 1). This finding is in line

with observations that PARP1 is recruited to sites of DNMT trapping-induced DNA damage to initiate repair of these lesions[61], implying that CeTEAM-based platforms can identify indirect target stability changes. Thus, CeTEAM is a tractable approach for both target validation and identifying biophysical perturbagens within larger chemical screens.

## Multimodal assessment of in vivo PARPi target engagement ex vivo

We then sought to assess CeTEAM in animal models, as favorable in vivo pharmacokinetic/pharmacodynamic (PK/PD) profiles are crucial milestones in preclinical drug discovery for confirming target

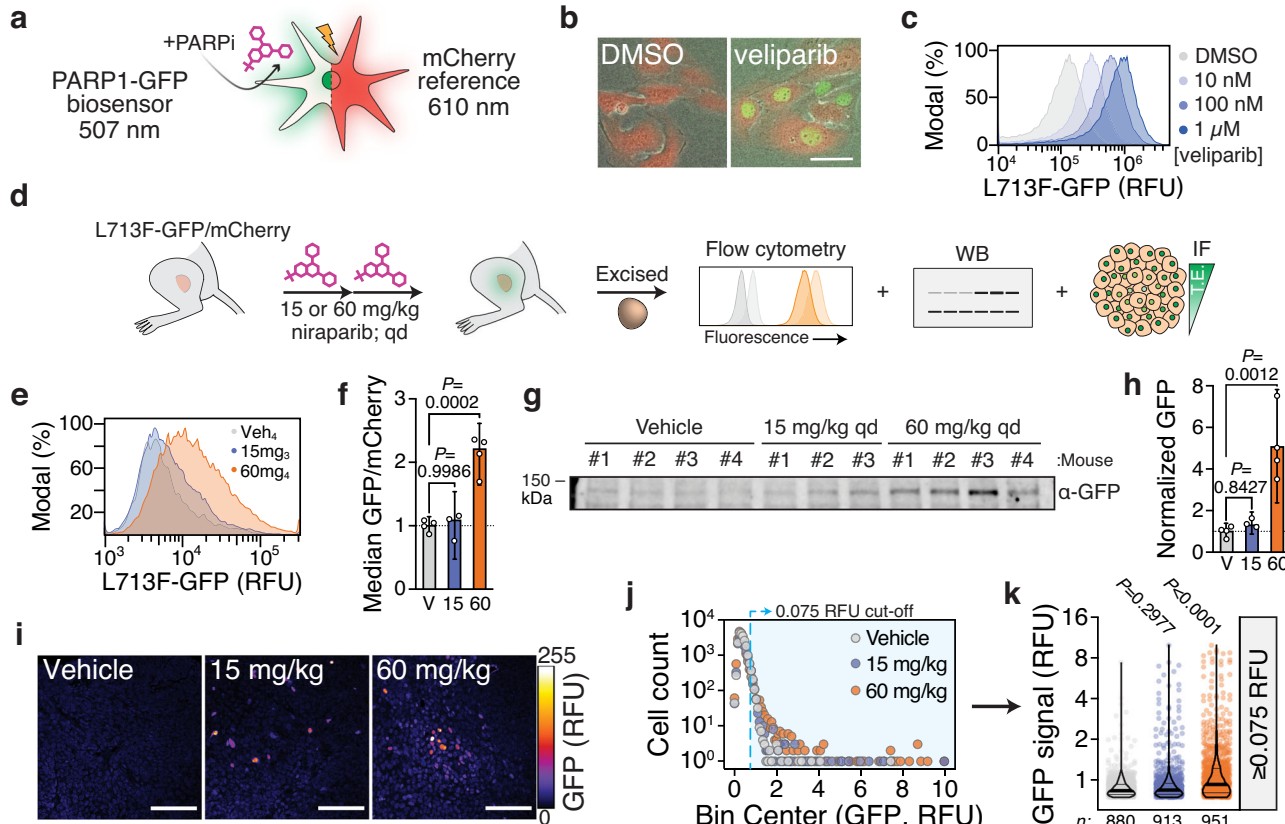

**Fig. 7 | In vivo-optimized CeTEAM PARPi-GFP biosensors for ex vivo detection of drug-target engagement. a** The PARP1 L713F-GFP biosensor was paired with mCherry for normalization. **b** Representative fluorescent micrographs (from $n = 3$) of U-2 OS PARP1 L713F-GFP/mCherry cells treated with DMSO or 1 μM veliparib for 24 hours. Scale bar = 50 μm. **c** mCherry-normalized L713F-GFP signal from 24-hour veliparib by flow cytometry. Modal normalization is shown. Representative of $n = 2$. Gray-blue gradient – veliparib gradient. **d** Graphical overview of in vivo experiments with HCT116 subcutaneous xenografts constitutively expressing PARP1 L713F-GFP and mCherry treated with either niraparib or vehicle control (2x, qd). $n_{Vehicle}$: 4, $n_{15mg/kg}$: 3, $n_{60mg/kg}$: 4 mice per treatment group. **e** Representative flow cytometry histograms of PARP1 L713F-GFP/mCherry tumors. Modal normalization is shown. Veh4: vehicle (mouse #4; gray), 15mg3: 15 mg/kg (mouse #3; blue), and 60mg4: 60 mg/kg (mouse #4; orange). **f** mCherry-normalized L713F-GFP intensity of tumors by flow cytometry. Means with 95% confidence intervals are shown from $n = 3$ (15 mg/kg) or $n = 4$ mice (Vehicle, 60 mg/kg). **g** Gross L713F-GFP signal from

individual tumors by western blot. **h** mCherry-normalized L713F-GFP abundance from western blots in **g** and Supplementary Fig. 19e. Means with 95% confidence intervals are shown from $n = 3$ (15 mg/kg) or $n = 4$ mice (Vehicle, 60 mg/kg). **i** Representative L713F-GFP micrographs from tumor sections with fire LUT pixel density depiction. Scale bars=100 μm. **j** Floating histogram of L713F-GFP intensities across tumors and treatment groups (gray – vehicle, blue – 15 mg/kg, orange – 60 mg/kg). Blue region represents an arbitrary cut-off of GFP intensity ≥ 0.075 RFU. 16,482 total cells per treatment group. **k** Distribution of individual cell L713F-GFP intensities ≥0.075 RFU. Violin plots with median (thick line) and quartiles (thin lines) are overlayed onto individual datapoints. In all cases, P values are shown for one-way ANOVA (Dunnett's test; **f** and **h**; $F_{Treatment\ (f)}$ [DFn, DFd] = 32.64 [2, 8], $F_{Treatment\ (h)}$ [DFn, DFd] =17.384 [2, 8]) or Kruskal-Wallis test (Dunn's test; **k**; Kruskal-Wallis statistic = 160.0) with multiple comparisons to the vehicle control. RFU relative fluorescence units.

engagement and functional effects in desired tissues[4]. To evaluate the PARPi biosensor in vivo, we first paired L713F-GFP with a complementary mCherry reporter that effectively normalized quantification of PARPi target engagement (Fig. 7a–c). This enabled straightforward identification of target cells by microscopy (Fig. 7b) or flow cytometry (Fig. 7c), which is particularly well-suited for heterogeneous in vivo environments.

Mice harboring subcutaneous L713F-GFP/mCherry tumors were then systemically administered niraparib, which has excellent oral bioavailability[62], for two consecutive days prior to tumor excision and evaluation of target engagement (Fig. 7d). Based on earlier pharmacokinetics data[62], we chose a higher niraparib dose of 60 mg/kg to ensure ample exposure at the tumor, but also a lower dose of 15 mg/kg to gauge the response of the biosensor system. Niraparib treatment dose-dependently stabilized L713F-GFP when measured by either live-cell flow cytometry (Fig. 7e, f, Supplementary Fig. 19a–d) or western blot (Fig. 7g, h, Supplementary Fig. 19e) following normalization to mCherry signal. Low dose niraparib negligibly increased GFP signal, while 60 mg/kg elicited a more robust response. Staining of tumor

cross-sections also revealed heterogeneous niraparib detection (Fig. 7i, j, Supplementary Fig. 19f, g) – where dose-dependent differences only became more obvious when restricting the analysis to higher L713F-GFP signal intensities (Fig. 7k). For reference, the flow cytometry analysis indicated ~50% of cells were mCherry+ and up to ~25% of these were GFP+ at 60 mg/kg (Supplementary Fig. 19c, d), suggesting that this discrepancy may be partially due to our staining protocol but also poor or uneven vascularization often seen in HCT116 xenografts[63,64]. Nonetheless, these data suggest that standard measures of drug distribution (e.g., plasma or tumor levels) might paint an incomplete picture of target occupancy. Thus, ex vivo CeTEAM analyses can provide single cell and multimodal insights to drug-target interactions in vivo.

**Non-invasive detection of PARPi engagement in live animals**
An enticing implication of the CeTEAM approach is the possibility to non-invasively track drug-target engagement in living systems. To test the capacity of CeTEAM in live animals, we deployed the dual luminescence PARP1 biosensor system as a tumor xenograft model that to

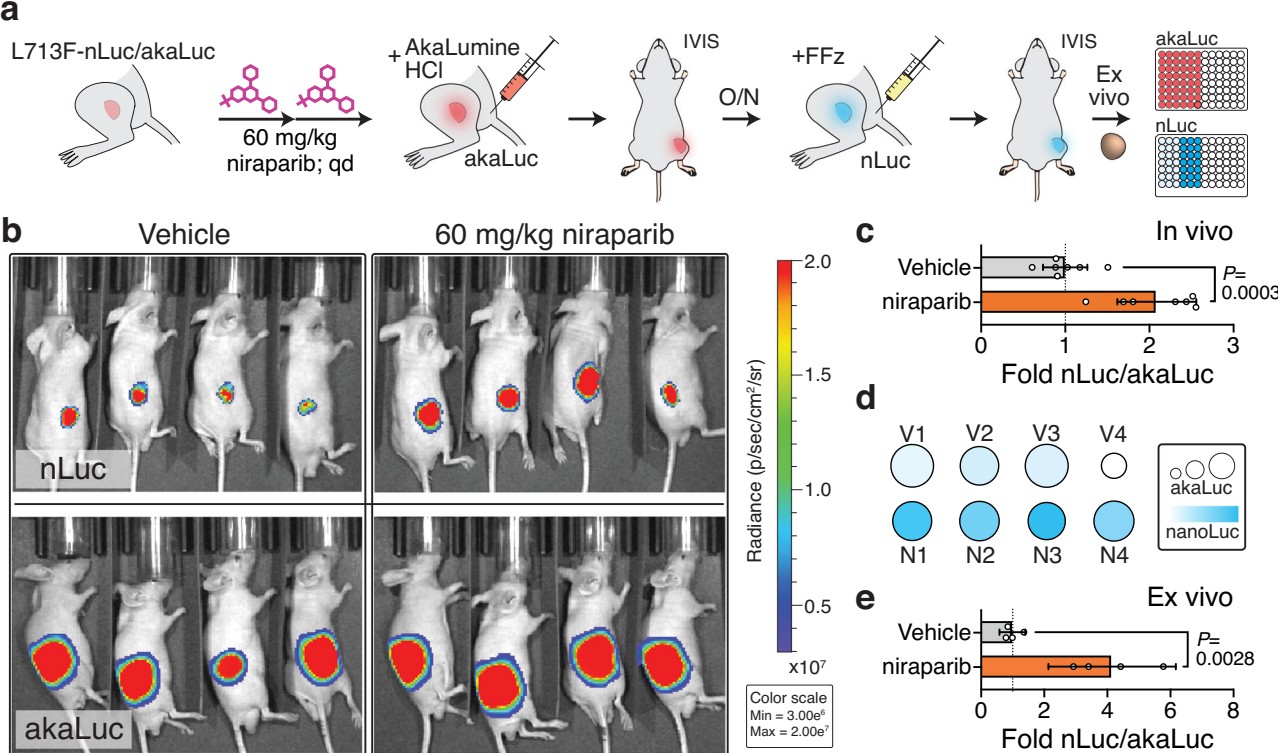

**Fig. 8 | An in vivo-compatible CeTEAM PARPi-nLuc biosensor for multiplexed tracking of drug binding in live animals. a** Graphical overview of in vivo experiments with constitutive expression PARP1 L713F-nLuc/akaLuc subcutaneous HCT116 xenografts treated with either vehicle or 60 mg/kg niraparib. *n* = 7 total mice per group. **b** Representative bioluminescence (radiance) overlays of mice treated as in a following administration of fluorofurimazine (nLuc) or AkaLumine HCl (akaLuc). Radiance intensity in psuedocolor representation. **c** akaLuc-normalized quantification of in vivo L713F-nLuc signals following vehicle (gray) or niraparib (orange) treatment from two experimental arms. **d** Ex vivo L713F-nLuc (blue gradient; 1,149,410 to 17,154,040 RLU) and akaLuc (circle size; 137,106 to 401,261 RLU) luminescence intensity representations of each tumor. **e** Quantitation of ex vivo, akaLuc-normalized L713F-nLuc bioluminescence following vehicle (gray) or niraparib (orange) treatment. For **c** and **e**, means ($n_c$=7; $n_e$ = 4) and 95% confidence intervals are shown, as well as P values from unpaired, two-tailed t tests (t, $df_c$ = 4.969, 12; t, $df_e$ = 4.871, 6).

detect PARPi. Sequential readout of akaLuc then L713F-nanoLuc with an in vivo-optimized nLuc substrate, fluorofurimazine[65], enabled quantitative detection of niraparib target engagement in live animals (Fig. 8a–c). As with the GFP variant, two systemic doses of PARPi were sufficient for clear detection of PARP1 L713F-nLuc binding in subcutaneous HCT116 tumors (Fig. 8a–c, Supplementary Fig. 19h and i) and was consistent across multiple experiments (Supplementary Fig. 19j and k). We were also able to confirm this finding following tumor excision and luminescence detection ex vivo. As before, akaLuc luminescence effectively normalized PARP1 L713F-nLuc signals on a per-tumor basis (Fig. 8d). Ex vivo validation experiments also clearly demonstrated a greater fold-change of PARP1 L713F-nLuc signal in niraparib-treated tumors over controls, presumably due to tissue absorption of blue-shifted nLuc emissions (Fig. 8e)[66]. Despite this drawback, reliable measures of PARPi target engagement in situ are still possible in suboptimal settings with the brightness of an nLuc system. Taken together, the in vivo luminescence data with the PARP biosensor suggest that CeTEAM is a practical, yet powerful, approach to monitor drug binding in living animals, which has exciting applications for longitudinal studies modeling drug resistance mechanisms, among many others.

## Discussion

Confirmation of target binding alone does not provide a complete pharmacological picture of a given test molecule. This is an important aspect of pre-clinical drug discovery that is often overlooked in early phases, despite representing a key branch point dictating future medicinal chemistry efforts. A somewhat surprising revelation from

the literature is that cellular target engagement assays are primarily used as confirmatory assays and not as a tool to discover the best targeted molecules in terms of on-target binding[7]. We envision that CeTEAM has the potential to shift this paradigm to earlier phases of drug discovery by enabling the rapid triaging of test molecules based on combined insights from biophysical and phenotypic components.

The downstream phenotypic events resultant from drug binding are difficult to protract in a single assay due to the perturbations inherent to current cellular target engagement assays. As an example, CETSA has effectively delivered cellular target engagement capabilities to the masses, and its extension to global proteomics enables deciphering of drug mechanism(s)-of-action[5,67]. However, the application of heat to detect binding events introduces confounding factors for interpreting the relationship between drug binding at elevated temperatures and downstream biology occurring at 37 °C (i.e., multiple changing equilibria), which require in-depth deconvolution to accurately decipher[8]. Similarly, approaches such as DARTS[68] or HIPStA[69] require global modifications that are disruptive to cellular biological processes. Therefore, there is high value in assays that can more easily bridge this knowledge gap.

One of the key advantages we envisioned when conceptualizing CeTEAM is that target binding is evaluated under physiological conditions, implying that other readouts, such as normalization labels or downstream pharmacological events, can be directly related to observed binding events up to the level of individual, live cells. This is possible due to a simple measure of biosensor abundance and easily translated from in vitro to in vivo applications to the extent of enabling non-invasive imaging of drug-target engagement in live animals. We

applied this theory utilizing destabilizing missense mutants previously described for MTH1, NUDT15, PARP1, DHFR, OGG1, and PARP2 to profile their cognate inhibitors in a relevant biological context and found high congruence with earlier findings. Following the initial observation that these mutants accumulated in cellular environments only following proteolytic inhibition or exposure to cognate ligands, we confirmed that the purified proteins were stabilized by ligand binding similarly to wild-type counterparts. This led us to the rationale that the proteolytic turnover of the destabilized variants is slowed by the stabilizing effect of ligand binding – presumably by facilitating protein folding and masking degrons that would otherwise be exposed at non-permissible temperatures[14]. Clonal expansion of biosensors significantly improved signal uniformity and robustness to better enable associations with phenotypic outcomes and even establish cause-and-effect, although the heterogeneous, asynchronous nature of cell populations will introduce noise into these measures.

When target biology is well established, CeTEAM repeatedly demonstrated the ability to discern effects related to target binding from those that are not; however, it may not be possible to pinpoint specific off-target binding events. Nevertheless, the assay can be complemented with any number of phenotypic markers to guide experimenters towards an understanding of a given molecule's pharmacology. These could be skewed towards specific pathways related to the chosen target or cherrypicked from multiple pathways for unbiased profiling of downstream phenotypes. Such an approach would be advantageous in many instances where triaging of small molecules must be done at scale (e.g., to understand phenotypic tendencies within inhibitor libraries, to determine novel biology related to a given target protein, etc.). Trends identified from CeTEAM can then be fine-tuned with orthogonal approaches, such as thermal proteome profiling[67], to rapidly identify desirable compounds and more completely understand their mechanism-of-action.

CeTEAM also affords a scalable platform for unbiased screening of biophysical perturbagens. We show this in principle with a drug-like small molecule screen of ~1200 compounds using PARP1 L713F that successfully found >90% of PARP1i with biochemical IC$_{50}$ < 1 μM, although the luminescence readout should also be well-suited for larger primary screens. In addition, there were other hits representing potential off-target binders and indirect stabilizers. The enrichment of DNMT-trapping molecules, decitabine and azacitidine, that give rise to PARP1-dependent recruitment and DNA damage repair, represents a protracted pharmacological outcome imparting PARP1 biophysical interactions. While an earlier time point could enrich for direct binding events and limit toxicity, these factors are tunable to the desired outcome of the screen, as acutely toxic molecules or those aberrantly dysregulating gene expression are readily triaged by akaLuc readout.

Locally disruptive variants with minimal gross structural alterations would be preferred candidates for conditionally stabilized ligand biosensors[14,16,70]. This conceivably reaches a tipping point when the structural changes no longer permit the small molecule to function as a chemical chaperone. While many destabilized mutants are identified serendipitously via disease linkages[16] or by randomized mutagenesis[13], the lack of rational discovery could limit the general adoption of CeTEAM. To this end, we demonstrate that their amenability can be imputed from existing biophysical and structural information and be transferred to close paralogs for expansion of potential drug biosensors. Earlier in vitro work showed that mutation of key leucine residues in the PARP1 HD domain conveyed differing degrees of PARP1 destabilization depending on proximity to the hydrophobic core[21]. When we reproduced these mutants as GFP fusions in cells, we saw that the extent of destabilization for a given mutant generally correlated with its drug biosensing dynamic range. More specifically, a certain threshold of destabilization is likely needed to discernably rescue the abundance with a binding small molecule. This was demonstrated empirically when L698A and L701A alone had the biosensing

equivalence of WT PARP1, however, their combination behaved like the other HD core mutants – in line with their thermal denaturation profiles. Applying this logic, we then successfully conferred the destabilizing effect of the PARP1 L713 mutation to the analogous residue in PARP2 (L269), thereby implying that structural similarity is sufficient to transfer instability to other proteins. While these results suggest that suitable CeTEAM mutations can be defined and applied to structural paralogs to expand the target pool, further data are needed to reinforce this possibility. The advent of modern computational, artificial intelligence, and protein engineering technologies should be helpful in this endeavor and may even extend to proteins without empirically resolved structures[12,18,71,72].

Another important aspect is the potential for functional perturbations by mutagenesis or overexpression, which may have implications for downstream biology and should be investigated empirically. The use of genome editing or cells with endogenously occurring mutations (e.g., OGG1 R229Q, NUDT15 R139C, etc.) could limit artifactual concerns, whereas employing catalytically-inactive biosensors might also be suitable. Similarly, while the use of fusion tags aid in the detection of compound-induced stabilization, they are not strictly necessary for the desired effect and, in some cases, could mask the rapid turnover of amenable mutants. It is also possible for test molecules that interfere with proteolysis or selectively bind to the destabilized target to arise as false positives. Several proteasome inhibitors in our PARP1 L713F-nLuc screen would have made significance thresholds, but their acute toxicities were flagged as akaLuc outliers (Supplementary Data 1). Although the inclusion of reference signals should mitigate this potential issue, counter-screening validation with orthogonal methods is beneficial.

In summary, the current work illustrates the benefits of deploying stability-dependent biosensors for drug discovery efforts – ranging from screening to in vivo quantification of target engagement. CeTEAM enables the direct association of cellular target binding with proximal or distal efficacy markers at cellular or subcellular resolution and high throughput, yielding cause-effect relationships not readily attained in other assays. While biophysical assessment of target engagement is possible within one hour, the study can be protracted to follow downstream effects of test molecules. A clear, overarching theme of these studies was that drug concentrations typically used for cell-based experimentation are much higher than required. For example, clinical PARPi engaged PARP1 in the low nanomolar range with trapping phenotypes manifesting soon thereafter, which contrasts with the micromolar usage in the literature. Phenotypes presenting beyond target saturation are even more likely to be off-target, as was the case with MTH1i, TH588, and is an all-too-common confounding factor in oncology drug discovery. Thus, our approach helps define a pharmacological window related to binding of the desired target. CeTEAM is a pragmatic, complementary approach for accelerating preclinical drug discovery that combines desirable aspects of targeted and phenotypic assays in a highly translatable system.

## Methods
### Cell lines and culturing conditions
U-2 OS osteosarcoma (HTB-96), HEK293T embryonic kidney epithelial (293T; CRL-3216), and KG-1 acute myelogenous leukemia (AML; CCL-246) cells were obtained from the American Type Culture Collection (ATCC, Manassas, VA, USA). HCT116 and HCT116 3-6 colon carcinoma cells were originally obtained from Dr. Bert Vogelstein (Johns Hopkins University). U-2 OS and HEK293T cells were cultured in DMEM high glucose, GlutaMAX medium (Thermo Fisher Scientific), HCT116 and HCT116 3-6 cells were cultured in McCoy's 5a, GlutaMAX medium (Thermo Fisher Scientific), and KG-1 cells were cultured in IMDM (Thermo Fisher Scientific). For in vitro luciferase read-outs, FluoroBrite DMEM or phenol red-free DMEM (Thermo Fisher Scientific) supplemented with GlutaMAX was used. All media were supplemented with

10% heat-inactivated fetal bovine serum (FBS; except for KG-1, which had 20%) and penicillin/streptomycin. Cell cultures were maintained at 37 °C with 5% $CO_2$ in a humidified incubator. Purchased cell lines were authenticated by the ATCC (STR profiling), and no further authentication was performed. The cells were routinely screened for mycoplasma using the MycoAlert kit (Lonza Bioscience) and none were listed as misidentified on ICLAC or known to be cross-contaminated.

## Antibodies and chemicals

anti-HA probe (mouse, clone F-7, cat. #sc7392, lot #L1281), anti-GFP (rabbit, cat. #sc8334, lot #D1907), anti-GFP (mouse, clone B-2, cat. #sc9996, lot #H2018), anti-PARP1 (mouse, clone F-2, cat. #sc8007, lot #D3019), and anti-SOD1 (mouse, clone G-11, cat. #sc17767, lot #G3119) were obtained from Santa Cruz Biotechnology. anti-CHK1 (mouse, clone 2G1D5, cat. #2360S, lot #8), anti-p-CHK1 Ser345 (rabbit polyclonal, cat. #2341S, lot #8), anti-p-CHK1 Ser345 (rabbit, clone 133D3, cat. #2348S, lot #18), anti-p-Histone H2A.X Ser139 (γH2A.X, rabbit, cat. #2577S, lot #12), and anti-vinculin (rabbit, cat. #4650S, lot #5) were obtained from Cell Signaling. anti-NUDT15 (rabbit, cat. #GTX32759, lot #822105550) was purchased from GeneTex. anti-V5 tag (mouse, clone SV5-Pk1, cat. #46-0705, lot #2735895) and anti-V5 tag (mouse, clone E10/V4RR, cat. #MA5-15253, lot #XI358694) was purchased from Invitrogen (now Thermo Fisher Scientific). anti-NUDT5 (rabbit polyclonal) was generated in-house as previously described[73]. anti-MTH1 (NUDT1, rabbit, cat. #NB100-109, lot #F-2) was obtained from Novus Biologicals. anti-p-Histone H3 Ser10 (rabbit, cat. #ab5176, lot #GR3396345-3), anti-β-actin (mouse, clone AC-15, cat. #ab6276, lot #0000182472), anti-α-tubulin (mouse, clone DM1A, cat. #ab7291, lot #GR3341361-15), and anti-OGG1 (rabbit recombinant, clone EPR4664(2), cat. #ab124741) were purchased from Abcam. anti-p-Histone H2A.X Ser139 (γH2A.X, mouse, clone JBW301, cat. #05-636, lot #3313712), and pan-ADP-ribose binding reagent (rabbit Fc tag, cat. #MABE1016, lot #2901597) were obtained from Millipore. anti-DHFR (rabbit, cat. #15194-1-AP, lot #00102546), anti-MTH1 (mouse, clone 2D7G4, cat. #67443-1-Ig, lot #10011993), and anti-PARP2 (rabbit, cat. #55149-1-AP, lot #00073384) were purchased from ProteinTech. Donkey anti-mouse IgG IRDye 680RD (cat. #925-68072, lot #D20803-13) and goat anti-rabbit IgG IRDye 800CW (cat. #925-32211, lot #D21109-25) were purchased from Li-Cor. anti-mCherry (rabbit, cat. # PA5-34974, lot #VB2946310D), donkey anti-mouse IgG Alexa Fluor 488 (cat. #A-21202, lot #1696430), donkey anti-mouse IgG Alexa Fluor 555 (cat. #A-31570, lot #2387458), donkey anti-rabbit IgG Alexa Fluor 568 (cat. #A-10042, lot #1020757), donkey anti-rabbit IgG Alexa Fluor 647 (cat. #A-31573, lot #2420695), goat anti-rabbit IgG Alexa Fluor 488 (cat. #A-11008, lot #913909), and donkey anti-mouse IgG Alexa Fluor 647 (cat. #A-31571, lot #1839633) were purchased from Thermo Fisher Scientific.

Doxycycline hydrochloride (Sigma-Aldrich) was dissolved in MilliQ water (2 mg/mL) and used at 1 μg/mL. MG-132 (Z-Leu-Leu-Leu-al, Sigma-Aldrich) was dissolved in DMSO (10 mM stock) and used at 5 μM. 6-methylthio-GDP, 6-methylthio-GMP, 6-thio-GMP, 6-thio-GDP, 6-thio-IMP, and 6-thio-IDP were purchased from Jena Bioscience and dissolved in MilliQ water to 10 mM. NSC56456 was obtained from the NCI Developmental Therapeutics Program and later re-synthesized[41], while TH8228 and TH8234 were synthesized in-house (see Chemical synthesis and characterization section), but all were dissolved in DMSO. Methotrexate, raltitrexed, TH5487, SU0268, and EB-47 were purchased from MedChemExpress. TH588, AZ19, IACS-4759 and BAY-707 were obtained or synthesized in-house as described previously[31,33–35,74]. Talazoparib, niraparib, olaparib, veliparib, and iniparib (SelleckChem) were dissolved in DMSO. 3-aminobenzamide (3-AB; Sigma-Aldrich) was dissolved in DMSO to a stock of 100 mM. All other inhibitors were dissolved at 10 mM. Furimazine was purchased as part of the Nano-Glo Assay kit (Promega), fluorofurimazine was obtained from Promega as a ready-to-use poloxamer-407 (P-407) desiccate that was reconstituted in sterile PBS[65], and akaLumine HCl

(TokeOni; Sigma Aldrich) was dissolved in MQ water to 40 mM, aliquoted, and stored at -80 °C.

## Chemical synthesis and characterization

### Synthesis of TH008228 (9-cyclohexyl-6-(methylthio)-9H-purin-2-amine).
2-amino-9-cyclohexyl-3H-purine-6-thione (NSC56456, 10 mg, 0.040 mmol) was dissolved in 0.5 M NaOH (1 mL) and stirred for 10 min., after which time MeI (2 μL, 0.040 mmol) was added and stirred for 2 hours at RT. The product was purified by preparative HPLC to give 9-cyclohexyl-6-(methylthio)-9H-purin-2-amine (4.00 mg, 37.9% yield) as a white powder (Supplementary Fig. 7). 1H NMR (600 MHz, DMSO-$d_6$) δ 8.02 (s, 1H), 4.21–4.16 (m, 1H), 2.56 (s, 3H), 1.94 (d, J = 9.0 Hz, 2H), 1.82 (app. t, J = 12.5,4H), 1.68 (d, J = 12.5 Hz, 1H), 1.39-1.34 (m, 2H), 1.25–1.21 (m, 1H); 13 C NMR (150 MHz, DMSO-$d_6$) δ 159.7, 159.3, 150.2, 138.5, 124.3, 52.9, 32.2, 25.2, 24.7, 10.8; LCMS (m/z): $[M+H]^+$ calcd. for C12H17N5S, 263.4; found, 264.2, Rt = 1.451 min., purity at 254 nm >95%.

### Synthesis of TH008234 (2-amino-9-(2-phenylpropan-2-yl)-3H-purine-6(9H)-thione).

Step 1: N-{2-amino-4-chloro-6-[(2-phenylpropan-2-yl)amino]pyrimidin-5-yl}formamide

α,α-dimethylbenzylamine (29.4 mg, 0.217 mmol) was added to a stirred solution of N-(2-amino-4,6-dichloropyrimidin-5-yl)formamide (30 mg, 0.145 mmol) and NEt₃ (2 eq) in ${}^i$PrOH (3 mL) and heated at 85 °C for 18 h. The reaction was cooled to RT, and the crude amino pyrimidine (44 mg, 99.3% yield) was collected by filtration.

Step 2: 6-chloro-9-(2-phenylpropan-2-yl)-9H-purin-2-amine

The crude amino pyrimidine (47 mg, 0.154 mmol) was dissolved in triethyl orthoformate (1.5 mL), heated at 120 °C for 12 hrs, then cooled to RT. HCl (0.05 mL, 12 M) was added and the reaction mixture was stirred for 12 hrs before concentrating under reduced pressure to give the crude 6-chloropurine (44 mg, 99.5% yield).

Step 3: 2-amino-9-(2-phenylpropan-2-yl)-3H-purine-6(9H)-thione

The crude 6-chloropurine (44 mg, 0.153 mmol) was dissolved in EtOH (2 mL), and thiourea (46.6 mg, 0.612 mmol) and formic acid (1 drop) were successively added before heating at 80 °C for 2 hrs. The reaction mixture was purified directly by preparative HPLC (acidic method) to give 2-amino-9-(2-phenylpropan-2-yl)-3H-purine-6(9H)-thione (6 mg, 13.8% yield) as a white solid (Supplementary Fig. 8). 1H NMR analysis revealed a 58:42 ratio between the thioamide and iminothiol tautomeric forms. 1H NMR thioamide tautomer (400 MHz, DMSO-$d_6$) δ 11.94 (s, 1H), 8.17 (s, 1H), 7.33–7.28 (m, 2H), 7.26–7.21 (m, 1H), 7.08–7.05 (m, 2H), 6.60 (s, 2H), 2.03 (s, 6H); 13 C NMR both tautomers (125 MHz, DMSO-$d_6$) δ 174.3, 160.1, 150.1, 147.7, 147.2, 145.0, 139.2, 128.5, 128.3, 128.0, 127.0, 126.6, 126.0, 125.2, 124.8, 124.6, 62.0, 54.7, 30.7, 30.5, 29.3, 29.1; LCMS (m/z): $[M+H]^+$ calcd. for C14H15N5S, 285.4; found, 286.2, Rt = 1.346 min., purity at 254 nm >95%.

**General methods and equipment.** All commercial reagents and solvents were used without further purification. Analytical thin-layer chromatography was performed on silica gel 60 F-254 plates (E. Merck) and visualized under a UV lamp. 1H NMR spectra were recorded on a Bruker DRX-400. Chemical shifts are expressed in parts per million (ppm) and referenced to the residual solvent peak. Analytical HPLC-MS was performed on an Agilent MSD mass spectrometer connected to an Agilent 1100 system with method B1090A: column ACE 3 C8 (50 × 3.0 mm); $H_2O$ ( + 0.1% TFA) and MeCN were used as mobile phases at a flow rate of 1 mL/min, with a gradient time of 3.0 min; Preparative HPLC was performed on a Gilson HPLC system: column ACE 5 C8 (150 × 30 mm); $H_2O$ (containing 0.1% TFA) and MeCN were used as mobile phases at a flow rate of 45 mL/min, with a gradient time of 9 min. For HPLC-MS, detection was made by UV using the 180 − 305 nM range and MS (ESI + ). For preparative HPLC, detection was made by UV at 254 or 220 nM. All intermediates and final compounds were assessed to be >95% pure by HPLC-MS analysis, unless stated otherwise.

## Protein production

Full-length NUDT15 wild-type and R139C were cloned, expressed, and purified as described previously[20]. Wild-type and L713F PARP1 catalytic domains were also expressed and purified as before[21].

## Differential scanning fluorimetry (DSF)

Protein unfolding was detected by differential scanning fluorimetry (DSF[75]). For NUDT15 experiments, 4 μM NUDT15 wild-type or R139C protein were added to 5x SYPRO Orange (ThermoFisher Scientific) in assay buffer (50 mM Tris-HCl, pH 8.0, 100 mM NaCl, 25 mM NaPO₄, 5 mM MgCl₂) in the presence of DMSO (1% final v/v), 50 μM NSC56456, or 50 μM TH8228. A CFX96 Touch Real-Time PCR Detection System (Bio-Rad) was used to increase the temperature from 25 °C to 95 °C in 1 °C/min increments, and fluorescence intensity was measured at each step. Data were acquired and melting temperature ($T_m$) calculated by CFX Maestro™ 1.0 Software (Bio-Rad, version 4.02325.0418) based on minima from the negative first derivative of the melt curve.

For PARP1 experiments, the wild-type catalytic domain (5 μM) or the L713F catalytic domain (5 μM) was incubated with PARPi (250 μM) or DMSO control in 25 mM HEPES, pH 8.0, 150 mM NaCl, 0.1 mM TCEP, 1 mM EDTA and 12.5% DMSO. The experiments were performed as previously described[21] using 5x SYPRO Orange and a Lightcycler 480 (Lightcycler 480 Software, version 1.5.1.62; Roche). The melting temperature ($T_m$) was calculated based on the minima from the negative first derivative of the melt curve using Prism (GraphPad, version 10).

## Enzyme-coupled malachite green assay

The enzyme-coupled malachite green assay for NUDT15 was performed as previously described[39]. Compounds were dispensed by an Echo Acoustic Liquid Handler to generate final concentrations ranging from 1.69 nM to 100 μM. Purified, wild-type NUDT15 (8 nM) and pyrophosphatase (0.2 u/mL) were combined in assay buffer (100 mM Tris-Acetate, pH 8.0, 40 mM NaCl, 10 mM MgAc, 1 mM DTT, and 0.005% Tween-20) and incubated for 10 minutes at room temperature. Negative controls were samples incubated without NUDT15 protein. Then, 100 μM dGTP (NUDT15 substrate) was added prior to a 15-minute incubation. A malachite green working solution (3.2 mM malachite green carbinol hydrochloride [Sigma Aldrich] in 3 M H₂SO₄ complemented with a final concentration of 1.5% ammonium molybdate and 0.17% Tween-20) was then added and followed by an additional 15-minute incubation prior to measuring absorbance at 630 nm on a Hidex Sense microplate reader (Aurentia Solutions, software version 0.5.11.2).

## NUDT15-NSC56456 co-crystallization and structure determination

Full length NUDT15 (15 mg/ml) was prepared in sample buffer containing 20 mM HEPES, pH 7.5, 300 mM NaCl, 10% Glycerol, and 2 mM TCEP in the presence of 5 mM NSC56456 dissolved in DMSO. Sitting drop vapor diffusion was performed at 4 °C and NUDT15 was mixed with reservoir solution (0.1 M Tris pH 8.5, 0.2 M sodium acetate, 30% PEG 4000) in a 1:3 protein/reservoir ratio. Diffraction-quality crystals appeared in the first week, were extracted quickly without additional cryoprotectant, and flash frozen in liquid nitrogen. Data collection was performed at beam line 14.1 at BESSY, Germany, at 100 K and wavelength 0.9184 Å. Data reduction and processing were carried out using DIALS (version 2.0) and AIMLESS (version 0.5.7) from the CCP4 software package[76–80]. The structure was solved by molecular replacement of the template structure file with PDB ID 5BON using Phaser6 (version 2.8.2)[81] followed by iterative building cycles using the Refine program in Phenix (version 1.14)[82]. TLS parameters were determined using the TLSMD webserver[83]. The structure was further validated using PDB_REDO[84] and deposited under PDBID: 7NR6. Statistics are found in Supplementary Table 1.

## Cloning

**General subcloning procedures.** 3xHA-tagged wild-type and R139C NUDT15[20], as well as V5-tagged wild-type and G48E p18 MTH1[19], lentiviral expression constructs were previously established in pINDUCER20 (a gift from Stephen Elledge (Addgene plasmid # 44012)[85]. GFP-PARP1 was transferred into pINDUCER20 via SalI/NotI ligation into pENTR4-N-GFP from pEGFP-C3-PARP1[86]. Similarly, PARP1 WT-GFP was made by subcloning into pENTR1a-C-GFP using SalI/NotI primers with the stop codon removed (non-stop, NS). PARP1 L713F was subcloned from pET28-PARP1 L713F[21] into pENTR1a-C-GFP by flanking SalI/NotI restriction sites and subsequently transferred to pINDUCER20, pCW57.1 (a gift from David Root; Addgene plasmid # 41393), or pLenti CMV Blast DEST (706-1), which was a gift from Eric Campeau & Paul Kaufman (Addgene plasmid # 17451)[87]. To generate pENTR1a-C-nLuc, nanoLuc (gBlock, IDT DNA) was subcloned into pENTR1a by flanking XbaI/XhoI restriction sites. PARP1 L713F was then subcloned into pENTR1a-C-nLuc by flanking SalI/NotI restriction sites prior to transferring into pINDUCER20, pCW57.1, or pLenti CMV blast. mCherry was subcloned from H2B-mCherry (a gift from Robert Benezra; Addgene plasmid # 20972)[88] into pENTR1a by flanking XhoI/XbaI sites prior to transferring to pLenti CMV Blast. Codon-optimized akaLuc[55] was subcloned from pEX-A258-akaLuc (Eurofins) into pENTR4 by flanking NcoI/SalI sites and transferred to pLenti CMV Blast, as before. DHFR WT and P67L were subcloned into pENTR1a-C-V5 by SalI/NotI cleavage-ligation prior to transferring to pINDUCER20. PARP2 WT and L269A were subcloned from pET28 vectors[27] into pENTR1a-C-GFP with flanking SalI/NotI restriction sites prior to transfer to pLenti CMV Blast. All subcloning into entry vectors was validated by automated sequencing, while shuttling into destination vectors was performed with Gateway LR Clonase II (ThermoFisher Scientific) and positive clones were confirmed by colony PCR.

**Site-directed mutagenesis (SDM).** Site-directed mutagenesis of OGG1 R229Q; PARP1 L698A, L701A, L765A, L768A, and L698A/L701A; and PARP2 L269A was performed based on the method reported by Zheng et al.[89] OGG1 R229Q mutagenesis primers were designed with Agilent QuikChange Primer Design. Successful mutagenesis was confirmed by automated sequencing. Mutagenesis primers are provided in Supplementary Data 2.

**Plasmids, primers, and synthetic DNA.** All primers and custom vectors were ordered from Eurofins Genomics. gBlock fragments were ordered from IDT. Other plasmids, unless developed in-house, were purchased from Addgene. The sources and sequences of all nucleic acids used in this study are summarized in Supplementary Data 2.

## Lentivirus production and transduction

Lentiviral production was performed following transfection of third generation lentiviral packing vectors by calcium phosphate precipitation. pINDUCER20, pCW57.1, or pLenti CMV Blast lentiviral constructs were co-transfected with lentiviral packaging vectors (Gag-Pol, Rev, and VSV-G envelope) into subconfluent HEK293T cells. Viral particles were harvested at 48- and 72-hours post-transfection, and target cells were transduced at 1:1 dilution of lentivirus and fresh, complete medium in the presence of polybrene (8 μg/mL). Forty-eight hours post-transduction, target cells were re-plated at low density in the presence of G418/neomycin (Sigma-Aldrich, 400 μg/mL for six days; pINDUCER20), puromycin (Sigma-Aldrich, 1 μg/mL for three days; pCW57.1), or blasticidin (Sigma-Aldrich, 5 μg/mL for four days; pLenti CMV Blast) that was replenished at three-day intervals. HCT116 pCW57.1-PARP1 L713F-nLuc cells were selected with 10 μg/mL puromycin over four days to enrich for high transductants.

## Reverse transcription quantitative PCR (RT-qPCR)

U-2 OS or HCT116 cells were plated at 40,000 cells/well in 12-well plates in the absence or presence of 1 μg/mL DOX. The following day, DMSO control or indicated inhibitor was added to the cells before harvesting with TRIzol (Thermo Fisher Scientific). RNA was purified with the Direct-zol RNA MiniPrep kit (Zymo Research) according to the manufacturer's instructions and quantified by NanoDrop (Thermo Fisher Scientific). cDNA was then generated with the iScript cDNA Synthesis Kit (Bio-Rad) according to the manufacturer's instructions. qPCR was performed with 2.5 ng cDNA per sample and iTaq Universal SYBR Green Supermix (Bio-Rad) using a Bio-Rad CFX96 Real-Time PCR Detection System (CFX Maestro™ 1.0, version 4.0.2325.0418). Relative quantity of target genes was calculated using the ΔΔCt method via normalization to *GAPDH*, *β-actin*, and/or *18s*. All qPCR primers are listed in Supplementary Data 2.

## Western blotting

Cells were plated and treated as described. At the termination of the experiment, cells were harvested and lysed directly in 1x Laemmli buffer. Following heating for 5 minutes at 95 °C, the samples were homogenized by sonication and either directly loaded for electrophoresis or frozen at -80 °C for later use. Protein samples were separated on 4-20% gradient Mini-PROTEAN gels (Bio-Rad) or TruPAGE gels (Sigma) prior to transferring onto 0.2 μm nitrocellulose with a Trans-Blot Turbo Transfer System (Bio-Rad). After blocking with LI-COR Blocking Buffer (TBS; LI-COR) for 1 hour at room temperature, primary antibodies were added at the following concentrations in 1:1 LI-COR Blocking Buffer and TBS + 0.05% Tween-20 at 4 °C overnight: anti-HA probe (mouse monoclonal, 1:500), anti-GFP (rabbit polyclonal or mouse monoclonal, 1:500), anti-mCherry (rabbit polyclonal, 1:2000), anti-PARP1 (mouse monoclonal, 1:500), anti-SOD1 (mouse monoclonal, 1:500), anti-CHK1 (mouse monoclonal, 1:1000), anti-p-CHK1 Ser345 (rabbit polyclonal, 1:1000), anti-NUDT15 (rabbit polyclonal, 1:1000), anti-V5 tag (mouse monoclonal, 1:500), anti-NUDT5 (rabbit polyclonal, 1:1000), anti-MTH1 (rabbit polyclonal, 1:500 or mouse monoclonal, 1:2000), anti-p-Histone H3 Ser10 (rabbit polyclonal, 1:1000), anti-γH2A.X (mouse monoclonal, 1:1000 or rabbit polyclonal, 1:500), anti-vinculin (rabbit polyclonal, 1:1000), anti-OGG1 (rabbit, 1:1000), anti-α-tubulin (mouse, 1:5000), anti-PARP2 (rabbit, 1:2000). LI-COR secondary antibodies were diluted in 1:1 LI-COR Blocking Buffer (TBS) and TBS + 0.05% Tween-20 at (1:10,000) or HRP-conjugated goat anti-rabbit secondary antibody (1:5000 in 5% milk/TBS) prior to incubating at room temperature for 1 hour. Blots were imaged with a LI-COR Odyssey Fc and analyzed using Image Studio (LI-COR, version 5.2). All uncropped blots with crop marks denoted are available in the Source Data file.

## Qualitative live cell fluorescence microscopy

To initially test GFP PARP1 biosensors in live cells, U-2 OS pINDUCER20-PARP1 L713F-GFP or HCT116 pLenti CMV Blast-PARP1 L713F-GFP/pLenti CMV Blast-mCherry cells were plated in the absence or presence of 1 μg/mL DOX. Where indicated, PARPi were added the following day for 24 to 48 hours. 5 μM MG-132, 100 nM bafilomycin A1, or a combination of both was added for 6 hours prior to imaging with either a ZOE Fluorescent Cell Imager (Bio-Rad, version 2.257) or EVOS FL Cell Imaging System (Thermo Fisher Scientific, version 1.4) at 10, 20, or 40x magnification.

## High-content fluorescence microscopy

For NUDT15 R139C experiments in HCT116 3-6 cells, 750 cells were plated in black, clear bottom 96-well plates (BD Falcon) on day 0 in the absence or presence of 1 μg/mL DOX in a volume of 80 μL complete medium. The following day, the appropriate concentrations of inhibitors were added in 10 μL complete medium (3% DMSO v/v). After a 3-hour pre-incubation, either 6TG or equivalent volumes of DMSO

were added to the cells in 10 μL complete medium for an additional 72 hours (final DMSO 0.3% v/v). Cells were then fixed with 4% paraformaldehyde (PFA) in PBS for 15 minutes, permeabilized with 0.3% Triton-X100 in PBS for 10 minutes, and then blocked with 3% bovine serum albumin (BSA) in PBS for 1 hour. Anti-HA (1:500) and anti-γH2A.X (rabbit, 1:1000) antibodies were incubated overnight at 4 °C in 3% BSA/PBS. The next day, the cells were washed three times (PBS, PBS + 0.05% Tween-20, and then PBS again) before incubation with Alexa Fluor 647 donkey anti-mouse (1:1000) and Alexa Fluor 488 goat anti-rabbit (1:1000) secondary antibodies for 1 hour at room temperature. Following another round of washes, the cells were counterstained with Hoechst 33342 (1 μg/mL in PBS) for 10 minutes prior to imaging.

For PARP1 and PARP2 experiments with live U-2 OS cells, 1000 cells were plated in black, clear bottom 96-well plates (BD Falcon) on day 0 in the absence or presence of 1 μg/mL DOX in a volume of 90 μL complete medium. The following day inhibitors were added to their indicated final concentrations in 10 μL of complete medium (final DMSO 0.1% or 1% [3-AB] v/v). After 24 hours, cell-permeable Hoechst 33342 was added to a final concentration of 1 μg/mL for 20 minutes prior to imaging. In instances where cells were fixed, the same set-up and fixation protocol was used as above. Primary antibodies were anti-GFP (mouse monoclonal, 1:300) combined with anti-p-CHK1 Ser345 (rabbit monoclonal, 1:300) or anti-GFP (rabbit polyclonal, 1:300) with anti-γH2A.X (mouse monoclonal, 1:1000). Anti-mouse Alexa Fluor 488 and anti-rabbit Alexa Fluor 647 (GFP mouse and p-CHK1 rabbit) or anti-rabbit Alexa Fluor 488 and anti-mouse Alexa Fluor 647 (GFP rabbit and γH2A.X mouse) were all used at 1:1000 dilutions.

Imaging was performed on an ImageXpress Micro high-content microscope (Molecular Devices, version 5) or CELLCYTE X (CYTENA, CELLCYTE Studio version 2.7.4) at 10x magnification (20x for fixed cells stained with anti-γH2A.X). For live-cell imaging, the microscope temperature, humidity, and CO₂ environment controller module were used to maintain cell ambient conditions (ImageXpress), or, for the CELLCYTE X, the microscope was contained within the humidified incubator. Image analysis was then performed with CellProfiler software (Broad Institute, version 3.1.0) and data plotted with GraphPad Prism (version 10).

## Confocal microscopy, FRAP, and microirradiation

Photobleaching and microirradiation microscopy experiments were carried out with a Zeiss LSM780 confocal laser scanning microscope, equipped with a UV-transmitting Plan-Apochromat 40x/1.30 Oil DIC M27 objective as previously described[90]. U-2 OS cells stably expressing GFP-PARP1 WT or PARP1 L713F-GFP were incubated in phenol-red free media containing DMSO, 5 μM olaparib or 0.5 μM talazoparib for indicated time periods. Cells were transferred to the microscope and eGFP was excited with a 488 nm Ar laser. The microscope was equipped with a heated environmental chamber set to 37 °C.

Microirradiation was carried out using either the FRAP module of the ZEISS ZEN software (version 2.1) or the tile scanning mode. Cells were pre-sensitized before microirradiation by incubation in medium containing 10 μg/mL Hoechst 33342 for 10 min. For inducing DNA damage with the FRAP module, a 10-pixel diameter spot within the nucleus was irradiated with a 405 nm diode laser set to 100% power. Before and after microirradiation, confocal image series of one mid z-section were recorded at 2 second intervals (typically 6 pre-irradiation and 120 post-irradiation frames). For evaluation of the recruitment kinetics, fluorescence intensities at the irradiated region were corrected for background and for total nuclear loss of fluorescence over the time course and normalized to the pre-irradiation value. For the quantitative evaluation of microirradiation experiments, data of at least 20 nuclei from two independent experiments were averaged and the mean curve and the standard error of the mean

calculated and displayed using Microsoft Excel 2010 and GraphPad Prism (version 10).

For DNA damage induction followed by immunofluorescence staining, cells were seeded on μ-Grid (35 mm with Grid, ibidi) dishes and sensitized with Hoechst 33342 (10 μg/mL, 10 minutes). Laser microirradiation was performed using the tile scan mode (3×3 tiles, image size 128×128, scan speed 177.32 μs, every 7th line scanned, 405 nm laser set to 70%), as previously described[91]. After indicated time periods, cells were fixed in 4% paraformaldehyde and permeabilized with 0.5% Triton X-100. Unspecific binding was blocked by incubation in PBS/4% BSA before staining with respective primary antibodies (pan-ADP-ribose binding reagent, 1:1000; γH2A.X, 1:1000). Primary antibodies were detected using secondary antibodies (diluted 1:500-1:1000 in PBS/4% BSA) conjugated to Alexa Fluor 568 or 647 (Thermo Fisher Scientific). Cells were counterstained with DAPI and kept in PBS until images were taken with the LSM780 microscope.

For FRAP analysis, half of the nucleus was marked using the regions tool of the ZEN software (ZEISS, version 2.1) and photobleached with the 488 nm laser set to maximum power at 100% transmission using 5 iterations at scan speed 8 (5 μs). Before and after bleaching, confocal image series were recorded with the following settings: 500 ms time intervals (20 prebleach and 200 postbleach frames), frame size 256×256 pixels, 170 nm pixel size, bidirectional scanning and a pinhole setting of 2.52 airy units. Mean fluorescence intensities of the bleached region were corrected for background and for total nuclear loss of fluorescence over time. For the quantitative evaluation of photobleaching experiments, data of at least 45 nuclei from five independent experiments were averaged and the mean curve, the standard error of the mean (s.e.m.), halftime of recovery (t½) and mobile fraction (Mf) calculated and displayed using Microsoft Excel 2010 and GraphPad Prism (version 10).

## Flow cytometry

**MTH1 G48E studies.** For MTH1 studies, 400,000 U-2 OS V5-MTH1 G48E clone #6 cells were plated in T25 flasks in the presence of 1 μg/mL DOX on day 0. The following morning (day 1), the cells were then treated with DMSO (0.01% v/v final concentration) or the indicated concentration of MTH1i for 24 hours. On the morning of day 2, the cells were harvested by trypsinization and pooling of culture medium, as well as PBS washes (to ensure collection of dead and mitotic cells). Following a wash with PBS, the cells were fixed with 4% PFA in PBS for 15 minutes, washed once with 1% BSA/PBS, then permeabilized with saponin buffer (0.1% saponin in 1% BSA/PBS) on ice for 30 minutes. The cells were then stained with anti-V5 (mouse, clone E10/V4RR, 1:300) and anti-p-HH3 Ser10 (rabbit polyclonal, 1:500) antibodies diluted in saponin buffer overnight at 4 °C. Next, the cells were washed twice with saponin buffer prior to incubation with donkey anti-rabbit Alexa Fluor 647 and donkey anti-mouse Alexa Fluor 488 antibodies (1:1000 in saponin buffer) for 30 minutes at 37 °C. Following two additional washes with saponin buffer, the cells were incubated with 0.1 mg/mL RNase A (Thermo Fisher Scientific) and 10 μg/mL Hoechst 33342 for 15 minutes at room temperature in 1% BSA/PBS. Control V5-MTH1 G48E cells were also used for singlet antibody controls (one for V5 and one for p-HH3 Ser10). The cells were then analyzed by flow cytometry on a BD Fortessa flow cytometer (Bectin Dickenson) using BD FACSDiva software (version 8.0.1) for acquisition (p-HH3 Ser 10-Alexa Fluor 647: R 670_30-A, V5-Alexa Fluor 488: B 530_30-A, Hoechst: V 450_50-A). Analysis (including cell cycle by the Watson Pragmatic method) and final gating of cell populations was performed with FlowJo software (Bectin Dickenson, version 10.7.1; see Supplementary Fig. 20a). FlowJo was also used to export raw, per-event values for subsequent plotting and analysis in GraphPad Prism (version 10).

**PARP1 L713F studies.** PARP1 experiments were performed with live U-2 OS pINDUCER20-PARP1 L713F-GFP #5/pLenti CMV Blast-mCherry or HCT116 pLenti CMV Blast-PARP1 L713F-GFP/pLenti CMV Blast-mCherry cells. Briefly, 200,000 cells were plated in T25 flasks (in the presence of 1 μg/mL DOX for U-2 OS cells). The following day, varying concentrations of veliparib/niraparib or an equivalent volume of DMSO was added to the cells prior to harvesting 24 hours later. Trypsinized cells were quenched with complete medium, pelleted at 400 x g for 5 minutes, washed by resuspending in sterile PBS/10% FBS, centrifuged again, and then transferred to 5 mL flow cytometry tubes via a 40 μm strainer cap (BD Falcon) in 500 μL sterile PBS/5% FBS. The samples were then analyzed on a BD Accuri C6 flow cytometer (Bectin Dickenson) with BD Accuri C6 software (version 1.0.264.21; GFP – PARP1 L713F stabilization, mCherry – fluorescence normalization). Unstained parental U-2 OS or U-2 OS pINDUCER20-L713F cells without DOX were used to establish final gating and compensation parameters with FlowJo (Bectin Dickenson, version 10.7.1; Supplementary Fig. 20b,c).

## In vitro luciferase assays

**nLuc assays.** For nLuc luciferase assays with U-2 OS PARP1 pINDUCER20-PARP1 L713F-nLuc cells, 1,000 cells per well were plated in white 96-well plates (Greiner) in complete medium and in the absence or presence of 1 μg/mL DOX. The following day, the medium was changed to FluoroBrite DMEM phenol red-free medium (Thermo Fisher Scientific) in the presence of DMSO or PARP inhibitor (fresh DOX was also added where necessary), and the cells were incubated for another 24 hours. nanoLuc signal was assessed using the Nano-Glo Luciferase Assay System (Promega) according to the manufacturer's instructions and the signal was quantified on a Hidex Sense plate reader (software version 0.5.11.2) with 1-second reads and an open filter setting. Readings were made every 3 minutes for up to 30 minutes following the addition of the lysis reagent.

For sensitivity measurements, U-2 OS cells were plated at serial dilutions ranging from 12 to 1,500 cells in white 96-well plates (Greiner) in complete medium and the presence of 1 μg/mL DOX. The medium was changed to FluoroBrite DMEM + DOX the following day and either DMSO or 2 μM veliparib were added to the cells for an additional 24 hours, before assaying by Nano-Glo kit with a Hidex Sense plate reader (software version 0.5.11.2) and an open filter. Signal-to-background (S/B = mean signal/mean background – $\mu_S/\mu_B$) was also calculated[56].

For dose response experiments with HCT116 pCW57.1-PARP1 L713F-nLuc cells, 4000 cells were plated in white 96-well plates (Greiner) in the absence or presence of 1 μg/mL DOX. The following day the medium was changed to complete FluoroBrite DMEM containing talazoparib (max conc. 10 μM), 3-AB (max conc. 1 mM), or iniparib (max conc. 20 μM) for 24 hours. After 24 hours, the medium was changed to furimazine-containing FluoroBrite (1:200 dilution from Nano-Glo kit) and immediately read with a CLARIOstar microplate reader (BMG LABTECH; software version 5.40 R2) with an open filter followed by a nLuc filter (470 ± 40 nm), both with spiral averaging. Data were exported using CLARIOstar MARS (version 3.31, BMG LABTECH).

**Combined nLuc and akaLuc assays.** Luminescence spectral profiling for both nLuc and akaLuc was determined by pre-treatment of HCT116 pCW57.1-PARP1 L713F-nLuc/pLenti-CMV-blast-akaLuc cells with 1 μg/mL DOX for 24 hours and 1 μM veliparib for an additional 24 hours. nLuc spectral profiling was performed immediately after the addition of a 1:200 dilution of Nano-Glo substrate in FluoroBrite DMEM, while the akaLuc spectrum was determined with 200 μM akaLumine HCl in FluoroBrite DMEM. Spectral scanning was determined on a CLARIOstar microplate reader on the spiral averaging setting.

To determine akaLuc tolerance to detergent-mediated lytic conditions, HCT116 pCW57.1-PARP1 L713F-nLuc HI/pLenti-CMV-blast-aka-Luc or U-2 OS pINDUCER20-PARP1 L713F-nLuc/pLenti-CMV-blast-

akaLuc cells were prepared for detection with 200 μM akaLumine HCl diluted in Nano-Glo lysis reagent (according to the lysis conditions described by the manufacturer) or 100 or 200 μM akaLumine HCl diluted in complete FluoroBrite DMEM medium before luminescence detection on an open filter and spiral averaging feature.

For testing akaLuc luminescence and cross-reactivity with nLuc signal, HCT116 pCW57.1-PARP1 L713F-nLuc/pLenti-CMV-blast-akaLuc cells were plated at 4000 cells per well unless otherwise stated and subjected to several different conditions. While testing akaLuc alone, luminescence signals were first established by comparison to parental HCT116 cells. The day after plating, the medium was changed to complete FluoroBrite DMEM with 200 μM akaLumine HCl and immediately measured on a CLARIOStar microplate reader with an open filter and spiral averaging feature.

For experiments testing simultaneous detection of nLuc and akaLuc signals, the cells were plated in complete medium supplemented with 1 μg/mL DOX. The following day, the cells were supplemented with 2 μM veliparib for an additional 24 hours. Just prior to imaging, the medium was replaced with complete FluoroBrite DMEM medium supplemented with 200 μM akaLumine HCl and furimazine (1:200 dilution) and immediately detected sequentially on an nLuc filter (470 ± 40 nm) and an akaLuc filter setting (650 ± 40 nm) using the spiral averaging feature.

For sequential reading of akaLuc signal followed by nLuc signal, HCT116 pCW57.1-PARP1 L713F-nLuc/pLenti-CMV-blast-akaLuc cells were plated in complete medium in the absence or presence of 1 μg/mL DOX (or not at all for HCT116 pLenti-CMV-blast-PARP1 L713F-nLuc/ pLenti-CMV-blast-akaLuc cells used for animal experiments) prior to incubation with indicated concentrations of PARPi for 24 hours. On the day of imaging, the medium was first changed to complete FluoroBrite DMEM with 200 μM akaLumine HCl and immediately imaged with an open filter and spiral averaging feature. Following a wash with FluoroBrite DMEM, the medium was changed again to FluoroBrite DMEM with a 1:200 dilution of furimazine and immediately imaged with an nLuc filter (470 ± 40 nm) and spiral averaging feature. nLuc signal was then normalized to akaLuc signal to obtain relative signal values and then normalized to DMSO-treated (in some cases, +DOX) controls to give fold change.

**Time- and target concentration-dependence of L713F-nLuc stabilization.** Time-dependent stabilization of PARP1 L713F-nLuc by PARPi was determined with HCT116 pCW57.1-PARP1 L713F-nLuc/pLenti CMV Blast-akaLuc cells plated at 4000 per well in the absence or presence of 1 μg/mL DOX. The following day, veliparib was added at different concentrations for 1, 8, or 24 hours prior to sequential akaLuc and nLuc detection as before. Fold nLuc signal was calculated by normalization of nLuc intensity to akaLuc intensity and relative to DMSO + DOX treatment. The influence of PARP1 L713F-nLuc target abundance on stabilization by PARPi was determined as above, but cells were pre-treated with 10, 100, 333, or 1 μg/mL DOX overnight followed by 24-hr treatment with veliparib at different concentrations. Luciferase activity signals were again detected sequentially and nLuc intensity was normalized to akaLuc intensity and set relative to DMSO + DOX readings.

**Small molecule screen for PARP1 biophysical perturbagens**
Complete screening details are summarized in Supplementary Table 2.

**Composition, storage, and plating of screening library.** The 1187 compound screening library consisted of the MedChemExpress Epigenetics and Selleck Nordic Oncology sets housed at the Science for Life Laboratory Compound Center, part of Chemical Biology Consortium Sweden (CBCS). The compounds are kept at -20 °C as 10 mM solutions in DMSO (some compound stocks were lower concentrations – see Supplementary Data 1 for more details) under low humidity using a REMP Small-Size Store system. Stocks were transferred to

LabCyte 384 LDV plates (LP-0200) to enable dispensing into assay plates with an Echo 550™ acoustic liquid handler (LabCyte). 100 nL of compound stock solutions were dispensed into white CELLSTAR® 96-well plates with tissue culture-treated surface (Greiner, 655083). Similarly, 100 nL DMSO (negative control) was dispensed into the first column of each assay plate, while 100 nL 10 mM veliparib (positive control) was dispensed into the second column of each plate. After addition of 100 μL growth medium to each well, the final DMSO concentration was 0.1% (v/v) and compound concentrations were up to 10 μM.

**Screen execution and data acquisition.** HCT116 pCW57.1-PARP1 L713F-nLuc/pLenti CMV Blast-akaLuc cells were initially pre-treated with 1 μg/mL DOX for 24 hours to induce expression of the L713F transgene. Cells were then trypsinized and replated into drug-containing assay plates at a concentration of $2.5 \times 10^4$ cells/100 μL and 1 μg/mL DOX using a Multidrop Combi liquid dispenser (Thermo Scientific). Cells were then incubated with drugs for 24 hours at 37 °C with 5% $CO_2$ in a humidified incubator. To minimize edge effects, the plates were placed in self-made humidity chambers that limited evaporation in the outer ring of wells. Luciferase signals were acquired sequentially on a CLARIOstar microplate reader as above using an open filter setting (akaLuc) followed by an nLuc-specific filter (470 ± 40 nm) with live cells.

**Data analysis and confirmation.** To eliminate high variations in nLuc/akaLuc ratios, an akaLuc cut-off was set at >4 standard deviations from the mean of control wells per plate, which improved stringency of the screening campaign and left 840 compounds eligible for analysis (Supplementary Fig. 18d and e). L713F-nLuc signals were first normalized to the akaLuc intensity, then set relative to the mean of the DMSO (negative) control to give relative fold change. Normality of the dataset was then improved by transformation to give $\log_2$(fold change [FC]) (Supplementary Fig. 18f). Hits were defined as ±2 SDs from the mean ($\log_2$[FC] for all valid test compounds), while ±3 SDs was added as an additional cut-off of significance. Complete screening data are supplied in Supplementary Data 1. Hit confirmations were performed with the same experimental set-up as the screen but with two independent sets of triplicate data points at 10 μM compound concentration.

**In vivo target engagement in tumor xenografts**
**Animal husbandry and ethical statement.** BALB/cAnNCrl nude mice (6–8-week-old; strain code: 194 [homozygous], Charles River Labs) were used for tumor xenograft experiments. The mice were housed in individually ventilated cages (type IVC, four per cage) under conditions of a 12-hour light/dark cycle and ambient temperature of 21 ± 4°C with 40–70% humidity. All work was performed in accordance with EU (European Union) and Swedish Ethical Review Authority regulations for animal experimentation under Karolinska Institutet permit no. 5718-2019. Tumors did not exceed 1000 mm3 in size in this study, as mandated by the Swedish Central Animal Research Ethics Committee (Centrala Djurförsöksetiska Nämnden).

**Establishment of PARP1 L713F-GFP/mCherry tumor xenografts and treatment with PARPi.** HCT116 pLenti CMV Blast-PARP1 L713F-GFP/ pLenti CMV Blast-mCherry cells were injected subcutaneously into the flanks of BALB/cAnNCrl nude mice ($2 \times 10^6$ cells in 100 μL PBS). Once the tumors reached approximately 200 mm3 in size, mice were systemically administered vehicle, 15 mg/kg, or 60 mg/kg niraparib formulated in 0.5% methylcellulose (Sigma) by oral gavage once daily (qd) for two days. On day three, the tumors were harvested and sub-divided for downstream analysis as follows: one half was formaldehyde-fixed and paraffin-embedded for sectioning and immunofluorescence analysis, one quarter was snap frozen for western blot analysis, and one quarter was immediately dissociated for live cell flow cytometry analysis.

**Tumor dissociation for flow cytometry.** Dissociation of excised subcutaneous HCT116 L713F-GFP/mCherry tumors was performed similarly to previously reported methods[92]. Briefly, approximately 60-80 mm3 of tumor tissue was finely minced with a sterile scalpel and dissociated for 1 hour at 37 °C with shaking in 9 mL of an enzyme solution containing 1 mg/mL collagenase D (Sigma) and 100 ng/mL DNAse I (≥ 40 U/mL final; Sigma) in McCoy's 5a GlutaMAX without additives. For the final 5 minutes of the incubation period, 1 mL TrypLE Express was added to each tube. Digestive enzymes were deactivated by the addition of 3 mL McCoy's 5a medium containing 10% FBS and the cell suspension was sieved through a 40 μm strainer (Corning), followed by an additional rinse with 2 mL McCoy's/10% FBS. The cells were pelleted by centrifugation at 400 x g for 5 minutes and washed by resuspension in sterile PBS/10% FBS. After centrifugation, the cell pellet was resuspended in 1 mL sterile PBS/5% FBS and transferred to a 5 mL flow cytometry tube via a 40 μm strainer cap (BD Falcon). Viability of the final samples was between 30-35% by trypan blue exclusion. Tumor suspensions were assayed on a BD Accuri C6 (Bectin Dickenson, software version 1.0.264.21) with identical gating strategy for cultured HCT116 GFP/mCherry cells (Supplementary Fig. 20b). Relative L713F-GFP was determined by normalizing median GFP intensity to median mCherry intensity.

**Western blotting of tumor cells.** Excised tumor samples were immediately snap frozen on dry ice and stored at −80 °C until use. In preparation for lysis, tumor masses were then pulverized with a mortar and pestle in liquid nitrogen prior to lysis in RIPA buffer and clarification of proteins by centrifugation. Following protein concentration measurements by BCA assay, equal concentrations of tumor lysate were prepared for western blotting, as above. Primary antibodies used were anti-GFP (mouse; 1:500), anti-mCherry (rabbit; 1:2000), and anti-β-actin (mouse; 1:5000).

**Sectioning, mounting, and immunohistochemistry.** Following excision, tumors were fixed in 4% PFA/PBS for one day and then submerged in 70% ethanol. Paraffinization, sectioning, and mounting of tumors was performed in the Pathology Core Facility at Karolinska Institutet Huddinge campus.

In preparation for immunofluorescence, the sections were deparaffinized and rehydrated with the following steps: 1. Xylene: 2 ×3 minutes, 2. Xylene/100% ethanol (1:1): 3 minutes, 3. 100% ethanol: 2 ×3 minutes, 4. 95% ethanol: 3 minutes, 5. 70% ethanol: 3 minutes, 6. 50% ethanol: 3 minutes, 7. Rinse with cold tap water. Following deparaffinization, antigen retrieval was carried out with citric acid buffer in a standard pressure cooker made for slides. The slides were then washed 2 ×5 minutes with TBS/0.025% Triton X-100 under gentle agitation, followed by blocking in 2% BSA/TBS for 2 hours at room temperature under high humidity. Anti-GFP primary antibody (1:100, mouse monoclonal, Santa Cruz) was then incubated overnight at 4 °C in 2% BSA/TBS. The next day, the slides were washed 3 ×5 minutes in TBS/0.025% Triton X-100 with gentle agitation. Following washing, donkey anti-mouse Alexa Fluor 555 secondary antibody (1:1000 in 2% BSA/TBS) was applied on the slides and incubated at room temperature for 1 hour. The slides were again washed under gentle agitation 3 ×5 minutes in TBS, followed by rinsing with distilled water. The sections were counterstained with DAPI (1:1000), rinsed once with distilled water, and mounted with ProLong Gold (Thermo Fisher). Images were taken using a 40x objective on a Zeiss LSM780 (ZEN software, version 2.1) confocal laser scanning microscope and processed in Fiji (ImageJ, version 2.1.0/1.53c).

**Establishment of PARP1 L713F-nLuc/akaLuc tumor xenografts and treatment with PARPi.** HCT116 pLenti CMV Blast-PARP1 L713F-nLuc/ pLenti CMV Blast-akaLuc cells were injected subcutaneously and treated with vehicle control or 60 mg/kg niraparib as above.

**Bioluminescence imaging of PARP1 L713F-nLuc/akaLuc tumors in live animals.** akaLuc and nLuc bioluminescence signals were sequentially imaged on an IVIS Spectrum (Perkin Elmer). On day two, the animals were anesthetized by isoflurane gas and 1 mg (3 μmol) akaLumine HCl in 100 μL sterile saline was administered IP prior to immediately imaging to determine tumor akaLuc signals[65]. akaLuc bioluminescence was measured every minute for 20 minutes with the instrument settings of open filter, medium binning, 60 second exposure time, and f-stop of 1. To ensure metabolic clearance of akaLumine HCl, the following day, 120 μL of fluorofurimazine/P-407 solution (1 μmol) was administered IP for detection of L713F-nLuc[65], and the mice were again immediately imaged on an IVIS Spectrum (Living Image software, version 4.7.2) with measurements every minute over 20 minutes. Instrument settings were the same as above. Results are presented as radiance (p/sec/cm2/sr) or total flux (photons/sec). Relative L713F-nLuc luminescence was calculated by normalization to akaLuc signal.

**Ex vivo bioluminescence measurements.** Following clearance of fluorofurimazine on day three of niraparib treatment, tumors were excised from the mice prior to sectioning with a sterile scalpel, and a quarter of each tumor was transferred to a well in a white, 96-well plate (Greiner). Tumor masses were gently flattened with sterile Dounce homogenizer plungers to spread tissue evenly in the well and submerged in 200 μL FluoroBrite DMEM containing 200 μM AkaLumine HCl. Immediately after addition of Akalumine, tumor akaLuc luminescence was measured with a CLARIOStar microplate reader (BMG LABTECH, software version 5.40 R2) on an open filter with the spiral averaging feature. The AkaLumine was then removed, and tumors were briefly washed with FluoroBrite DMEM prior to submerging in 200 μL furimazine-FluoroBrite (diluted 1:200 from the Promega Nano-Glo kit). L713F-nLuc signals were then immediately measured using nLuc-specific filter settings (470 ± 40 nm). To determine relative L713F-nLuc luminescence, signals were normalized to akaLuc readings and set relative to the vehicle control tumors.

### Statistical analyses and data transformation

All graphing and statistical analyses were performed using GraphPad Prism, version 10. Saturation curve fitting was performed using the [agonist] vs response four parameter variable slope model in Prism. Specific post-hoc tests, variations, and statistical significances for relevant experiments are described within individual figure legends. In some instances, raw data were transformed to simplify graphical visualization. Specifically, fluorescence values in Fig. 2i, Fig. 7j and k, and Supplementary Fig. 14j were multiplied by a factor of 10. Data in Fig. 4h (γH2A.X) and j (γH2A.X); Supplementary Fig. 14a, b, e, and h; Fig. 5d, f (GFP), g (GFP); Supplementary Fig. 10d and g; Supplementary Fig. 11c; and Supplementary Fig. 15b, c, e (GFP, pCHK1) and 15g, h, and j (GFP, γH2A.X) were multiplied by a factor of 100. Data in Fig. 4j (HA); Supplementary Fig. 10c and f; and Supplementary Fig. 11k were multiplied by a factor of 1000. Data in Supplementary Fig. 13f were multiplied by a factor of 10,000.

### Reporting summary

Further information on research design is available in the Nature Portfolio Reporting Summary linked to this article.

## Data availability

The data generated in this study is available in the main text, Supplementary Information, or Source Data file (provided with this paper), as well as from the corresponding author. The NSC56456-NUDT15 co-crystallization data generated in this study has been deposited in the RCSB Protein Data Bank database under accession code 7NR6. PDB accession codes 7KK2 and 3KCZ were previously published. Materials

are available from the corresponding author upon request. Source data are provided with this paper.

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

## Acknowledgements

The authors thank Promega for providing fluorofurimazine (Thomas Kirkland and Joel Walker); the SciLifeLab Compound Center for assisting with screening compound handling; Sean Rudd, Kristoffer Valerie, Thomas Gustafsson, as well as many other colleagues for critical feedback and discussion of the work; the beamline scientists at BESSY II (Berlin, Germany) for help with structural data collection; and the Protein Science Facility (PSF) at Karolinska Institutet and SciLifeLab (http://ki.se/psf). This research was supported by the Swedish Childhood Cancer Society (TJ2019-0036 – NCKV; PR2016-0101 – TH), Cancer Research KI (Karolinska Institutet) Blue Sky Grant (NCKV), Felix Mindus Contribution to Leukemia Research (2019-01992 – NCKV), Loo and Hans Osterman Foundation (2020-01208 – NCKV), Karolinska Institutet Research Foundation (2020-01685, 2022-01749 – NCKV; 2018-01655 – BDGP), Swedish Society for Medical Research (S16-0152 – BDGP), Michael Smith Foundation for Health Research (18292 – BDGP), Canadian Institutes of Health Research (BMA342854 – JMP), European Research Council (TAROX-695376 – TH; 2015-00162 – TH; 2018-03406 – PS), Swedish Pain Relief Foundation (SSF/01-05 – TH), Swedish Cancer Society (21 0352 PT – NV; CAN2018/0658 – TH; 201287 – PS), Torsten Söderberg Foundation (TH), Alfred Österlund Foundation (PS), Hållsten Foundation (MA), SciLifeLab Technology Development Project Grant (MA), Novo Nordisk Pioneer Innovator Grant 1 (NNF22OC0076798 – MA, NV), and MJP was supported by the European Union's Horizon 2020 research and innovation programme under the Marie Skłodowska-Curie grant agreement No 859860.

## Author contributions

K.S., O.M., and S.M.Z. contributed equally to this work. Conceptualization (N.C.K.V., B.D.G.P., and M.A.), Methodology (N.C.K.V.), Investigation (N.C.K.V. [lead], K.S., O.M., S.M.Z., S.A., M.J.P., H.S., A.R., M.F.L., D.R., A.T., O.P.S., M.D., J.O., P.W., I.A., J.B., L.B., G.B., B.D.G.P.), Resources (P.S., J.M.P., T.H., B.D.G.P., M.A.), Writing – original draft (N.C.K.V. [lead], B.D.G.P., M.A.), Writing – review & editing: (N.C.K.V. [lead], K.S., O.M., S.M.Z., S.A., M.J.P., H.S., A.R., M.F.L., D.R., A.T., O.P.S., M.D., J.O., P.W., I.A., J.B., L.B., G.B., P.S., J.M.P., T.H., B.D.G.P., M.A.), Visualization: (N.C.K.V. [lead], K.S., O.M., S.A., M.J.P., D.R.), Supervision (N.C.K.V. [lead], P.S., J.M.P., T.H., B.D.G.P., M.A.), Project administration (N.C.K.V.), Funding acquisition (N.C.K.V., P.S., J.M.P., T.H., B.D.G.P., and M.A.).

## Funding

## Competing interests

The authors declare the following competing interests: N.C.K.V., B.D.G.P., and M.A. are inventors on a patent application describing CeTEAM and its uses (PCT/EP2019/073769). The remaining authors declare no competing interests.
