## [Transparent Peer Review file · Nature Communications]

Coupling cellular drug-target engagement to downstream pharmacology with CeTEAM

Corresponding Author: Dr Nicholas Valerie

Version 0:

Reviewer comments:

Reviewer #1

(Remarks to the Author)

Valerie et al. provide a clever approach to coupling drug-target engagement with pharmacological outcomes using CeTEAM. By expressing a proteolytically unstable mutant of their target protein, they can readily track drug-target engagement (TE) as the mutant protein is stabilized by the drug binding event. When coupled with a suitable handle for the readout (i.e., V5 or HA tags, GFP) this gives researchers a biosensor for TE to compare and correlate to other pharmacological readouts. In drug discovery, building these TE-pharmacology correlations is a fundamental challenge, and so new tools to address this challenge such as CeTEAM are certainly welcome.

The strength of the paper lies in the PARP1 inhibitor work where the authors nicely build correlations between PARP1 TE and events such as DNA trapping. They follow this up with some in vivo and ex vivo xenograft work to track TE. The work with the Nluc biosensor is particularly welcome, as this gives good signal in the in vivo and ex vivo settings. I would have liked to see rationale for the dose choices (efficacious for xenograft regression?). A comparison of the 15 mg/kg dose in the Nluc vs GFP biosensor would have also strengthened this section, since I'm curious if drug- tumor permeation data in Figure 5F is just a limitation of the sensor choice or actually poor permeation.

Despite the strength of the PARP1 work, there are some concerns around the limitations of the CeTEAM platform. The authors do mention some of these limitations in the Discussion section, which is laudable, but the limitations still stand. 1) First, there isn't much generality to CeTEAM platform that would make it readily deployable. The three chosen target mutants (MTH1 G48E, NUDT15 R139C, and PARP1 L713F, had been discovered in previously published work and are likely the exception rather than the rule for finding these biosensor-ready mutants. Doing de novo discovery and validation of a new CeTEAM biosensor one's target of interest is probably going to represent a large amount of work. Additionally, whole proteome profiling can often detect drug-target upregulation upon binding and can parse complex pharmacology at scale (Ruprecht et al. Nature Chemical Biology 2020, 16(10), 1111). So, any techniques would have to compete with these approaches.

2) Overexpression presents challenges in that it introduces large amounts of the exogenous biosensor, which can overwhelm your endogenous target. The authors do address this to some degree by using dox-inducible constructs but getting the right expression level is not trivial.

3) Confirming equipotent binding between WT and mutant is also a potential challenge and is not addressed by the authors for their targets. This is critical when parsing detailed pharmacology, as one could easily under- or overestimate TE if there is a disconnect.

4) For many of the TE experiments, Western Blotting is used, which provides a semi-quantitative readout at best. This is very a key limitation, as building quantitative correlations to other readouts cannot be done unless multiple replicates are run and the Western assay has been validated to be in the quantitative range through a linear standard curve. Most Western data is presented in the manuscript as single replicate for any given measurement, and for each mutant biosensor, the authors should show a full dose curve and build EC50 target occupancy/stabilization (see point 3 above). This is less a limitation for the GFP biosensor.

The main critique I have of the manuscript is that in the first two examples with MTH1 and NUDT15, the experimental comparisons are made with very high concentrations of drug that are well above their efficacious doses.

1) The MTH1 inhibitors show low or sub-nanomolar IC50s and stabilization of the MTH1 biosensor at low concentration; yet, the pharmacology CeTEAM experiments are run at 1 and 2.5 micromolar drug for TH588 and AZ-19, respectively. If one

runs an experiment at a sufficiently high concentration, one can almost always find off-target effects. At the same time, comparisons are made between TH588 and AZ19, but no comparisons to the other inhibitors are made.

i) Why 2.5 micromolar for TH588 and 1 micromolar for AZ19

ii) For Figure 2F and G why is there a difference in G1/G0 vs G2/M in the total cell population between TH588 and AZ19 treated, but in the V5 high in Figure 2I, there doesn't appear to be a large difference?

2) For NUD15, again significant stabilization of the mutant biosensor is seen at 1 micromolar concentrations, but most experiments were run at 33 micromolar. Also, the choice of 72 hour timepoint is confusing as well since this seems very long to drive a pharmacological effect, and the Western blots show stabilization at a much earlier time (Extended Figure 1c)

i) The authors do build a dose-dependent correlation in Figure 3J and show some differential pharmacological effects between TH8234 and NSC56456, which is the strongest data in this section. However, the use of potentiator is perhaps not appropriate descriptor for TH8234 if it is acting as a pro-drug.

3) The PARP1 experimental work is done in the relevant concentration ranges with the tested inhibitors, and so there is less concern for the work from that part of the manuscript.

Some figures may benefit from some re-working to improve clarity.

1) For some of the statistical comparisons, for example in Extended Figure 5, the precise comparisons are not completely clear. At the same time there are significant results for some comparisons which should be non-significant? (Extended Figures 3A and B). Additionally, the TE dose response in Extended Figures 3D and F are somewhat weak and not (though they are significant)

2) Also, the authors use dox-inducible biosensors for many experiments, but don't always designate when they are using inducible vs non-inducible. They should just make sure they say so consistently throughout the paper in their figure captions.

Overall, the technical execution of the experiments in this work is strong and spans diverse techniques. However, some of the design and conclusions will need additional work before publishing. The impact and significance also have limitations for some of the reasons outlined above and mentioned by the authors in the Discussion section.

Reviewer #2

(Remarks to the Author)

Manuscript by Valerie et al describes novel drug target engagement methodology CeTEAM. Based on the destabilizing mutations authors provide evidence that this methodology can be applied to various targets such as MTH1, NUDT15, PARP and also correlate the target engagement with biological activity identifying off target actions of some compounds.

Furthermore the methodology is expanded to in vivo and non invasive in vivo imaging. It's a remarkable work, still poor writing and data distribution amongst the figures really hamper its potential.

1. The authors have 6 main figures, 10 extended figures and 9 supplementary figures that indicates poor judgement in data presentation. Many of the supplementary figures such as 1, 5,6 etc have GFP intensity plots of dose responses for various compounds that the reader is expected to eyeball. They should be quantified, dose response curves provided and representative plots provided while the rest should go into extended data. Majority of us only have 2-3 screens so having 4 sources plus the suppl discussion really will not help authors to get the value of the work appreciated.

2. Fig 1b is not explained. Why is it needed and what are the compounds there. No assumption should be made that the reader is familiar with TH8234 if wider audience is intended for the paper. The point of the Fig 1 is not clear as similar data for MTH is provided in Fig 2b

3. Fig 2e indicates H2 Ser10 signal is going down while the text is stating the opposite

4. Most of the figures have limited statements on the replicates performed for western blotting or other techniques.

5. The FRAP experiments presented in Suppl Fig 2 are not convincing to support the conclusion that L713F is less mobile than WT PARP1

6. Images in extended Fig 5 are critical for the conclusions but they are extremely pixelated, please provide representative higher quality images. Ext Fig 5a controls for various groups look very different? Is equating trapping with PARP1 concentration fair? Fig 4d is a summary of entire Ext Fig 5 as referred in the text? Some illustration in the main figure would be helpful.

7. In the discussion it would be helpful to circle back on how applicable this approach is beyond 3 targets

Reviewer #3

(Remarks to the Author)

Valerie et al. report the development of a new method for measuring drug target engagement and phenotypic responses in a single assay. Their method, called cellular target engagement by accumulation of mutant (CeTEAM), involves overexpressing the target protein of interest with a destabilizing missense mutation and simultaneously measuring its stabilization by drug binding along with downstream phenotypic effects. The authors demonstrate CeTEAM's utility by investigating MTH1, NUDT15, and PARP1 inhibitors in cells. They further use CeTEAM to investigate PARP1 inhibitor engagement in a mouse model, using ex vivo measurements to show heterogeneity in drug distribution across tumors and in vivo measurements to measure target binding in live mice.

The use of drug-binding-induced stabilization of destabilized mutants to measure target engagement is conceptually interesting and novel. This was even achieved in vivo. Demonstrating on-target engagement and characterizing phenotypic

effects are vitally important in drug discovery and development. However, Valerie et. al. do not adequately show (1) how the benefits of measuring target engagement and phenotypic effect in a single multiparametric experiment outweighs the downsides of overexpressing a mutant protein in their system and (2) how CeTEAM is superior to measuring target engagement and phenotypic effects in separate experiments and correlating the results. To more convincingly demonstrate the utility of their assay, Valerie et al. should present novel results or biology that could not be readily discovered with existing methods or show that their method is applicable across a broader range of drug target classes (e.g., kinases). With these major revisions, I believe this manuscript will be of broad interest to the drug discovery and chemical biology communities.

Major comments:

1. This assay is framed by the authors as an widely applicable platform that will facilitate drug discovery and advance understanding of drug pharmacology. However, it was not shown within the scope of the paper how this platform has the potential to promote discovery of novel biology and/or outperform currently available platforms that utilize multiple, separate assays to achieve conclusions about drug pharmacology or off-target effects (e.g. drug-resistant alleles). In this regard, we suggest the author consider one of the two following directions to bolster their claims:

A. Demonstrate novel results that can only be found in this multiplexed assay and/or substantiated analysis that shows that this outperforms existing methods

B. Demonstrate broader applicability and scalability of platform, e.g. across protein families like kinases and/or larger small molecule libraries

2. In this paper, the authors over-express unstable variants of enzymes that are stabilized by drug binding as a biosensor for cellular target engagement. However, systems that utilize over-expression are known to be susceptible to epiphenomenons that result from non-physiological levels of proteins. To eliminate any doubts about veracity of conclusions established by the results of the platform, the following controls experiments should be added:

A. Repeat one of the three systems with more variants with destabilizing missense mutations to verify stabilization effect for all the mutants regardless of mutation position and show the dynamicity of the systems beyond a single variant.

B. Quantitative experiments that over-express the wild-type variants instead of the missense variant should be added to complement results found with the destabilized variant for MTH1 and NUDT15 to show that effects seen are specific to protein activity and not caused by the specific mutant used. Alternatively, specificity of downstream cellular effects can also be verified by introducing a mutation that results in a catalytically-dead enzyme on top of the destabilizing mutation.

Minor comments:

1. For all figure captions, please provide complete descriptions of the experiments/analysis performed, the cell type used, as well as relevant concentrations and time points.

2. Because the focus of this manuscript is a new concept/method in chemical biology, readers may not be familiar with MTH1, NUDT15, and PARP1 biology specifically. Please provide more background on the biology of these enzymes, why they were chosen for this method, and how their activity is typically quantified. For example, provide a more complete description of DNA trapping assay using Hoechst dye.

3. Several of the figures could benefit from rearranging subpanels to ensure clarity. In general readers expect panels to progress left to right and top to bottom.

Figure by Figure comments:

Figure 1:

Please simplify panel D by removing or moving the cells representing downstream phenotype. The current arrangement is unclear.

Figure 2:

- The claim that TH588 triggers the mitotic surveillance pathway independent of MTH1 binding is only partially supported in this study. Please provide additional background on previous work that came to the same conclusion.
- For panels H and I, I am concerned about the robustness of an analysis based on a gate that includes only 2% of the DMSO treated cells. Please provide a head to head analysis with and without the initial "V5 high" gate.
- For panels D-I, please provide additional controls: Overexpressed WT MTH1, enzymatically inactive MTH1, and MTH1 with other destabilizing missense mutations. (See major comments 2B)
- Consider supplementing cell cycle data in parts F-I with proliferation data to show the overall effect of inhibitor treatments.

Figure 3:

• While the conceptual framework of "non-responder, stabilizer and potentiator" is helpful, TH8234 does not fit cleanly into this model. The authors suggest that TH8234 acts both as a potentiator and a thiopurine prodrug. While this is a possibility, there is insufficient evidence to rule out unintended off-target effects that TH8234 may have in cells, especially at high (33uM) concentrations. To make this claim, I suggest performing further experiments to detect TH8234 to thiopurine

conversion. Additional indirect evidence could be provided by combining 200nM 6-TG with other thiopurines to show a similar effect on DNA damage markers.

- Panel H: The authors do not discuss the clear differences in cells treated with NSC56456 and cells treated with TH8234. TH8234 appears to induce an accumulation in S phase as opposed to G2/M. Visually, this accumulation is very different (and potentially more prone to noise) than the accumulation seen in NSC56456 treated cells. This merits further investigation.

Figure 4:

- Panel D, which is all text, could be included in the body of the manuscript instead of in a figure.
- Panel E could be split into two panels.
- Panel E-G. State exactly how Hoechst dye intensity relates to DNA trapping.

Figure 5:

- Provide a panel that directly shows that the PARP1 sensor detects the spatial distribution of niraparib. Current panels clearly show the PARP sensor is stabilized in vivo, but do not show a clear gradient of signal across the tissue/tumor. Claims of spatial resolution depend on more specific spatial data and these should be highlighted.
- In the text, reserve claims of temporal resolution for the discussion of Figure 6. Figure 5 only provides potential evidence for spatial resolution, not spatiotemporal.

Version 1:

Reviewer comments:

Reviewer #1

(Remarks to the Author)

In the revised manuscript, Valerie et al. provide significant improvement to the overall study and provide a significant amount of additional requested data. The conclusions are valid and experiments well executed.

The primary critique of the translatability of the CeTEAM platform to different target proteins has been addressed in part with the inclusion of additional examples and paralog analysis. While broad applicability across target classes will still be a challenge for the CeTEAM platform, it is less concerning with the next data. Still, this reviewer would be interested to see applicability to non-enzyme targets in the future.

Secondly, the authors address well the choices of compound dose and pharmacological read out to build more quantitative relationships. Increasing the focus on the more quantitative readouts as opposed to the Western Blots is helpful and lends to stronger conclusions overall. This reviewer appreciates the improved and nuanced language the authors use as they decipher complex pharmacology. This strengthens their conclusions as they also appropriately denote limitations and caveats.

Thirdly, the inclusion of the drug screening portion of the paper adds a broader applicability piece that was perhaps missing in the original manuscript. If a biosensor mutant is available for a given target, it could be a useful tool for drug hunters. I'd encourage the authors to think about demonstrating the CeTEAM platform on potential drug targets where this could have outsized utility in parsing pharmacology.

Finally, the work is quite data heavy, and so this reviewer appreciates the improvements to clarity of the visual and text portions of the figures, as well as within the body of the text.

This resubmission is appropriate for publication in Nature Communications and will be of interest to a broad audience across disciplines.

Reviewer #2

(Remarks to the Author)

The concerns and suggestions were fully addressed

Reviewer #3

(Remarks to the Author)

The updates to this manuscript substantially bolster our confidence that CeTEAM is a valid and generalizable technology. The authors have sufficiently addressed our concerns about the usefulness of CeTeam by including additional proteins and performing a small molecule screen. They have also allayed our concerns about overexpression by including additional wildtype controls and demonstrating a similar effect endogenously with OGG1. While they did not attempt to design an inert, catalytically dead sensor, we consider the many other additions to their study to be sufficient. We recommend this paper for acceptance and publication in Nature Communications.

The following are additional suggestions to improve the manuscript for readability and accuracy:

1. Claims that their PARP1 experiments enable “rational” design of sensors based on “biophysical logic” should be scaled back. While successful in this case, the manuscript does not detail specific rules about how to design these destabilizing point mutations for any protein.
2. Supplementary Fig. 5 E-G could be further improved. The gating strategy and sequence is unclear, and it is unclear how each subgate relates to each other. It would be best to clarify: at what point are singlets gated? The way panel G is written, it seems that singlets are a subgate of V5 high; is this correct? Is the histogram in e already gated for singlets? What is the “parental population” in g?
3. Fig. 4 G-H could be simplified greatly using bar graphs for data quantitation while the images could be moved to the supplement.
4. Fig. 4 seems to use NSC56456 and TH7410 interchangeably. If this is indeed the same molecule, please use one name consistently in the figure and text.

REVIEWER COMMENTS

Reviewer #1 (Remarks to the Author):

Valerie et al. provide a clever approach to coupling drug-target engagement with pharmacological outcomes using CeTEAM. By expressing a proteolytically unstable mutant of their target protein, they can readily track drug-target engagement (TE) as the mutant protein is stabilized by the drug binding event. When coupled with a suitable handle for the readout (i.e., V5 or HA tags, GFP) this gives researchers a biosensor for TE to compare and correlate to other pharmacological readouts. In drug discovery, building these TE-pharmacology correlations is a fundamental challenge, and so new tools to address this challenge such as CeTEAM are certainly welcome.

The strength of the paper lies in the PARP1 inhibitor work where the authors nicely build correlations between PARP1 TE and events such as DNA trapping. They follow this up with some in vivo and ex vivo xenograft work to track TE. The work with the Nluc biosensor is particularly welcome, as this gives good signal in the in vivo and ex vivo settings. I would have liked to see rationale for the dose choices (efficacious for xenograft regression?). A comparison of the 15 mg/kg dose in the Nluc vs GFP biosensor would have also strengthened this section, since I'm curious if drug- tumor permeation data in Figure 5F is just a limitation of the sensor choice or actually poor permeation.

- *We thank the Reviewer for the nice comments on the in vivo work. As to the choice of niraparib dosing, we did not consider anti-tumor efficacy at this stage, rather reported pharmacokinetics data. We refer in the text to a recent study highlighting the high bioavailability of niraparib in preclinical models (Sun et al., 2018)¹. They show that 50 mg/kg delivered orally each day for two days routinely gave micromolar plasma concentrations after 24 hours, which theoretically should be more than sufficient to detect with the biosensor based on the in vitro data. This was the basis of our selection of the 60 mg/kg dose and is now further explained in the relevant text on page 16.*

*The 15 mg/kg dosing was chosen as a lower value to get an idea of the sensitivity in the tumor model. Ideally, we would have included the 15 mg/kg dosing with the nLuc biosensor, but we were limited by the number of animals available for the experiment and instead opted for higher numbers of animals per group to ensure robustness of the data. For the 60 mg/kg dose, we see that the abundance fold change is very similar for both the GFP (**Figure 7f**) and nLuc biosensor (**Figure 8c**). We certainly agree that a comparison of 15 mg/kg between the luminescent and fluorescent biosensor would have improved insights on this lower dose, as the nLuc readout is very sensitive.*

Despite the strength of the PARP1 work, there are some concerns around the limitations of the CeTEAM platform. The authors do mention some of these limitations in the Discussion section, which is laudable, but the limitations still stand.

1) First, there isn't much generality to CeTEAM platform that would make it readily deployable. The three chosen target mutants (MTH1 G48E, NUDT15 R139C, and PARP1 L713F, had been discovered in previously published work and are likely the exception rather than the rule for

finding these biosensor-ready mutants. Doing de novo discovery and validation of a new CeTEAM biosensor one's target of interest is probably going to represent a large amount of work. Additionally, whole proteome profiling can often detect drug-target upregulation upon binding and can parse complex pharmacology at scale (Ruprecht et al. Nature Chemical Biology 2020, 16(10), 1111). So, any techniques would have to compete with these approaches.

- *We certainly agree with the reviewer that de novo discovery and validation of CeTEAM biosensors would be a major concern in terms of establishing generality for the technique. While we generally find that reporting of destabilizing or potentially destabilizing missense mutants is relatively widespread and that restricting the choice of destabilizing mutation to those residues predicted or known to locally destabilize the protein structure are highly successful with the method, we have added additional data suggesting that amenable biosensors can be rationally identified. In addition to the three examples initially reported, we now include further examples to illustrate this point: 1) DHFR (**Supplementary Figure 2**) and 2) OGG1 (with both an exogenous and endogenous format; **Supplementary Figure 3**). More importantly, we demonstrate that the concept is not limited to any one missense mutant by extending the PARP1 example to multiple other mutations in the HD domain, including some that additively destabilize (**Figure 2a-e, Supplementary Figure 4a**). The amenability of these mutants followed biophysical logic, arguing that they can be rationally identified. To illustrate this in practice, we show that the destabilization of PARP1 by L713F can be transferred to its close paralog, PARP2, via the analogous L269 residue (**Figure 2f-i, Supplementary Figure 4b**). Thus, it is conceivable that the pool of compatible biosensors can be readily expanded with structural insights. We hope that the expanded data sections ameliorate some of these concerns.*

We also thank the reviewer for highlighting the benefits of analogous techniques, for example the proteomic profiling technique described by Ruprecht et al. We believe that CeTEAM is a great complement to techniques such as these or CETSA, that can employ proteome-wide analyses to characterize the downstream effects of a drug treatment. We envision a workflow that could employ CeTEAM for initial characterization of target binding and phenotypic responses, then expanding to a proteome-wide scale that could be useful for identifying the specific mechanism-of-action of the drug in question. Such an approach could rapidly triage molecules for more detailed mechanistic analyses. We have added a citation to the Ruprecht et al. study in the Introduction to acknowledge more classical proteome-wide analysis of drug mechanism-of-action (page 3).

2) Overexpression presents challenges in that it introduces large amounts of the exogenous biosensor, which can overwhelm your endogenous target. The authors do address this to some degree by using dox-inducible constructs but getting the right expression level is not trivial.

- *Thank you for your comment. We agree with the Reviewer that overexpression of an exogenous protein can introduce artifacts and complicate the interpretation of drug binding data vs observed phenotypes. In addition to doxycycline-regulable promoters, we employed lentiviral transduction to further control expression levels of our biosensors. We consistently see that they are expressed at comparable (or even below) endogenous protein levels (except for possibly MTH1). We also note that upon titrating doxycycline with our*

nLuc PARP1 biosensor, we did not see appreciable changes to the observed stabilization EC_{50} following PARPi treatment (only magnitude of signal plateau intensity), suggesting the approach can be robust even with reasonably variable biosensor expression (**Figure 6c, Supplementary Figure 13n**).

While we primarily utilize exogenous expression constructs in the study, the approach is also applicable to naturally occurring mutants. New data comparing OGG1 R229Q by exogenous and endogenous expression (a biallelic mutation in KG-1 leukemia cells) and demonstrate that they behave essentially the same (although the dynamic range afforded by the higher affinity GFP antibody was overall better; **Supplementary Figure 3**). Further, we have redacted the text to emphasize that exogenous expression or the use of a fusion tag is not necessary for a functional CeTEAM assay, merely a mutant capable of increasing the dynamic range of protein abundance after drug binding (see pages 3, 5, and 20, as well as **Figure 1d**/it's legend). Nonetheless, this is an important consideration for this application or any other experimental set-ups reliant on exogenous expression systems.

3) Confirming equipotent binding between WT and mutant is also a potential challenge and is not addressed by the authors for their targets. This is critical when parsing detailed pharmacology, as one could easily under- or overestimate TE if there is a disconnect.

- This is a great point to keep in mind. In general, we have seen high correlation of compound-induced thermal shifts between WT and mutant forms by DSF for targets tested (**Figure 2e, Supplementary Figure 8a-c**; or previously reported). These results demonstrated to us that binding potency is still highly similar in many cases, but it would be overclaiming to say that this is generally true. Available structural information has also enabled the selection of mutations that are not expected to impact the binding site. From our CeTEAM experimental results, we see that many drugs have a similar stabilization EC_{50} as reported biochemical values. Specifically, for the in-depth examples:
 - **MTH1 G48E**: Biochemical inhibition of MTH1 G48E is also notably similar to WT, which further strengthens the claim that potency is similar for this mutant (Mur et al., 2019)². Related to this, observed EC_{50} values of TH588 and AZ19 with MTH1 G48E were similar to those from CETSA with WT MTH1 (15 nM and 2 nM vs. 10 nM and 7 nM, respectively; Kettle et al., 2016)³, again suggesting that drug binding modes are highly similar.
 - **NUDT15 R139C**: NUDT15i have highly similar binding modes in WT and R139C protein co-crystal structures (Rehling et al., 2021)⁴.
 - **PARP1 L713F**: PARPi biochemical IC_{50} values for PARP1 L713F are similar to WT (Chen et al., 2019)⁵.

Accordingly, the text on pages 7, 9, and 11 has been amended to emphasize the supporting data regarding the similar potency between WT and mutant proteoforms.

An exception regarding mutant versus WT drug binding was the DHFR example (**Supplementary Figure 2**), where the P67L mutation is expected to discourage methotrexate binding (Lévy, Johnston, and Varshavsky, 1999)⁶. Methotrexate is a picomolar hDHFR inhibitor in cells (Jolivet and Chabner, 1983)⁷, and this difference in

*binding is reflected in the rightward shift in apparent stabilization EC_{50} towards the high nanomolar range (~325 nM; **Supplementary Figure 2c**).*

We also wish to stress that CeTEAM can most appropriately facilitate interpretation of relative drug potency and how this relates to phenotypic outcomes. While not absolutely quantitative, these insights will be invaluable to understanding the relationship of target binding and related pharmacology of small molecules. We have highlighted this challenge in our Discussion section and believe that comparisons to in vitro analyses and to other cellular target engagement techniques are beneficial to characterizing observed drug binding events using the CeTEAM method.

4) For many of the TE experiments, Western Blotting is used, which provides a semi-quantitative readout at best. This is very a key limitation, as building quantitative correlations to other readouts cannot be done unless multiple replicates are run and the Western assay has been validated to be in the quantitative range through a linear standard curve. Most Western data is presented in the manuscript as single replicate for any given measurement, and for each mutant biosensor, the authors should show a full dose curve and build EC_{50} target occupancy/stabilization (see point 3 above). This is less a limitation for the GFP biosensor.

- *Thank you for your comment. This could have been something that was clearer in the text. We completely agree on the semi-quantitative limitations of Western blotting. Nonetheless, we included blots in the initial submission to demonstrate qualitative changes in mutant abundance in the presence of a binding ligand and, thus, they were not quantified. We reserved quantitation to more sensitive and reliable methods, such as fluorescence (PARP1) or immunofluorescence (IF) microscopy (MTH1 and NUDT15), or bioluminescence (PARP1). Quantification by these methods is generally preferred and, further, can provide more information about i.e., the uniformity of drug binding at the single-cell level, as well as more absolute quantification (in the case of luminescence).*

*For the resubmission, we have included Western blot dose curves for DHFR P67L (**Supplementary Figure 2**), OGG1 R229Q (**Supplementary Figure 3**), MTH1 G48E (**Figure 3d, Supplementary Figure 5b and c**), NUDT15 R139C (**Supplementary Figure 7a and b**), while the WT proteins are compared at a single high dose of drugs. We show representative blots from at least three independent experiments, as well as the summary quantitation. Our rationale for this was to 1) simply demonstrate the amenability of DHFR P67L and OGG1 R229Q to CeTEAM; and 2) justify appropriate drug doses used for microscopy (NUDT15) and flow cytometry (MTH1) experiments. In the case of MTH1 G48E, saturation profiles by WB were informative for choosing appropriate drug concentrations to use for multiplexed flow cytometry analysis. The PARP1 and PARP2 mutants were primarily analyzed by live-cell fluorescent microscopy, which as the Reviewer mentions, is more accurately quantifiable. Thus, while WBs are still informative for CeTEAM, the use of direct fluorescence or bioluminescent readouts is more reliable and should be preferred.*

The main critique I have of the manuscript is that in the first two examples with MTH1 and NUDT15, the experimental comparisons are made with very high concentrations of drug that are

well above their efficacious doses.

1) The MTH1 inhibitors show low or sub-nanomolar IC50s and stabilization of the MTH1 biosensor at low concentration; yet, the pharmacology CeTEAM experiments are run at 1 and 2.5 micromolar drug for TH588 and AZ-19, respectively. If one runs an experiment at a sufficiently high concentration, one can almost always find off-target effects. At the same time, comparisons are made between TH588 and AZ19, but no comparisons to the other inhibitors are made.

- *This is absolutely a fair point. We agree that this was the only example without a relevant dose curve and needed to be more systematically addressed. We have now included a dose-response comparing TH588 and AZ19 with the G48E mutant (Figure 3d, Supplementary Figure 5b and c). We utilized the subsequent saturation profiles to identify relevant drug concentrations corresponding to G48E occupancy/stabilization states. This enabled us to make relevant comparisons of TH588 and AZ19 based on their apparent binding potency to the biosensor and, thus, facilitated comparison of differential phenotypic changes. Comparison of TH588 and AZ19 was then done by multiplexed flow cytometry read-out at sub-saturated, saturated, and literature (supersaturated) drug concentrations (Figure 3d-f, Supplementary Figure 5d-g). This rational approach can more appropriately show 1) that TH588 uniquely induces a mitotic delay phenotype but only well beyond G48E saturation and 2) that the concentrations of these molecules typically used in cell-based assays are likely much higher than is appropriate to decipher relevant pharmacology. Similar results were obtained with exogenous WT MTH1 to support this point (Supplementary Figure 5d). Thanks for this comment, it really strengthened these data.*

As for the TH588 comparisons, we rationalized that a more in-depth comparison with one of the other molecules would be sufficient (we simply chose AZ19). Previous work with each of these molecules showed that only TH588 (or its related analog, TH287) elicited a mitotic proliferation defect (probably through disruption of tubulin dynamics) while the others do not, and our preliminary were in line with these observations (Figure 3b)^{3, 8-10}.

i) Why 2.5 micromolar for TH588 and 1 micromolar for AZ19

- *Please refer the comment above, which addresses this point in detail.*

ii) For Figure 2F and G why is there a difference in G1/G0 vs G2/M in the total cell population between TH588 and AZ19 treated, but in the V5 high in Figure 2I, there doesn't appear to be a large difference?

- *Thanks for this point. This is likely having to do with the number of cells qualifying as V5 high –particularly evident for the DMSO control, where < 1000 cells were captured – which may not give so accurate representations of the cell cycle. Your point also highlights the complexity of correlating biosensor readouts to the cell cycle, as the cells are asynchronous and, in most cases, only delayed through the cell cycle rather than stopped. One way to address this would be through cell synchronization, which would certainly improve uniformity and correlations but at the expense of realistic scenarios.*

*This data has now been updated with newer experiments (detailed description above) where more cells were collected. The summary of the overall cell populations for V5 and pHH3 Ser10, as well as DNA content (cell cycle) are shown in **Figure 3e-f**. A representative experiment of the V5 high gating and statistics is now shown in **Supplementary Figure 5e-g**. In the overall cell populations, only cells treated with high-dose TH588 (“literature”, 10 μ M) had a meaningful change in mitotic cells. For V5 high cells, those treated with 10 μ M TH588 were even further enriched in mitotic cells (4n DNA/pHH3 Ser10⁺). As before, AZ19 did not appreciably affect the cell cycle or pHH3 Ser10 levels, even in the V5 high fraction.*

*We did notice a discrepancy in overall G0/G1 content between TH588 treatment experiments in the initial submission (2.5 μ M TH588 and 1 μ M AZ19) versus new results we have included as part of the revision (10 μ M TH588 and 10 μ M AZ19). Specifically, 2.5 μ M TH588 yielded a clear G0/G1 enrichment at the expense of S-phase cells, while mitotic cells (pHH3 Ser10⁺/4n DNA) were also clearly enriched. In contrast, 10 μ M TH588 overwhelmingly enriched cells in mitosis with modest changes to G0/G1- and S-phase proportions at the same time point of 24 hours. After consulting the literature, this discrepancy falls in line with data reported by Gul et al., 2019¹¹, where cells treated with \sim <8 μ M TH588 were able to proceed through mitosis and arrest in the next G1-phase while higher doses permanently arrested most cells in metaphase. We have added a cross-comparison of this data in **Supplementary Figure 5h and i** to illustrate that TH588-treated cells that do exit mitosis then arrest in G1, in line with the mitotic surveillance checkpoint hypothesis put forth by Gul et al. The text on page 8 has also been updated to reflect this.*

2) For NUD15, again significant stabilization of the mutant biosensor is seen at 1 micromolar concentrations, but most experiments were run at 33 micromolar. Also, the choice of 72 hour timepoint is confusing as well since this seems very long to drive a pharmacological effect, and the Western blots show stabilization at a much earlier time (Extended Figure 1c)

- *Thank you for your comment. Some of the data presented in **Figure 4** (previously **Figure 3**; specifically, panels **g** and **h**) are to show representations of multiparametric analysis – now updated for comparing data across a single dose of NUDT15i. These data are picked from larger dose-response experiments shown in greater detail in **Supplementary Figure 7**. Meanwhile, the data in **Figure 4i** and **j** is a summary of the dose response work in **Supplementary Figure 7c-i**. To clarify this, we have amended the text on page 10.*

*We absolutely agree with the Reviewer that the 72-hr time point is much later than would be necessary to detect stabilization of NUDT15 R139C (already prominent by 24 hours, as seen in **Supplementary Figure 1c** for 6TG [formerly Extended Figure 1c]). The problem relates to the mechanism-of-action of thiopurines and the requirement of several rounds of DNA replication (\sim 2-3 days’ time) for DNA damage to manifest (Valerie et al., 2016)¹². To ensure sufficient time for evaluating 6TG potentiation, we then used 72 hours post-6TG as an endpoint. Although not ideal, this has been a sufficient readout of intracellular NUDT15 activity in lieu of an alternative because the endogenous function has not yet been found^{4, 12-14}. We have clarified our rationale on this point throughout the text on pages 9-11.*

i) The authors do build a dose-dependent correlation in Figure 3J and show some differential pharmacological effects between TH8234 and NSC56456, which is the strongest data in this section. However, the use of potentiator is perhaps not appropriate descriptor for TH8234 if it is acting as a pro-drug.

- *Thank you for pointing this out. In the original submission, the description of NUDT15i triaging from **Figure 3j** was inaccurate regarding potential thiopurine prodrugs (e.g., TH8234). The updated panel (**Figure 4j**) now reflects the potential for R139C stabilizers to either be classified as “potentiator” (in the presence of low-dose thioguanine) or “6TG mimetics” if they elicit DNA damage without supplemental thioguanine. These changes are also specified in the text on page 11.*

3) The PARP1 experimental work is done in the relevant concentration ranges with the tested inhibitors, and so there is less concern for the work from that part of the manuscript.

- *Thank you for your comment.*

Some figures may benefit from some re-working to improve clarity.

1) For some of the statistical comparisons, for example in Extended Figure 5, the precise comparisons are not completely clear. At the same time there are significant results for some comparisons which should be non-significant? (Extended Figures 3A and B). Additionally, the TE dose response in Extended Figures 3D and F are somewhat weak and not (though they are significant)

“For some of the statistical comparisons, for example in Extended Figure 5, the precise comparisons are not completely clear.”

- *In **Supplementary Figure 9** (formerly **Extended Figure 5**), the statistical comparisons are made with 1) the unirradiated, DMSO controls for each variant and 2) between the unirradiated, DMSO controls (indicated by the brackets). We agree that this should be clearer in the figure legend. Accordingly, the legend has been updated to include this information.*

“At the same time there are significant results for some comparisons which should be non-significant? (Extended Figures 3A and B).”

- *Thanks for this comment. Regarding the datasets in question from **Supplementary Figure 7** (formerly **Extended Figure 3**), the original data presented were of single cells from a representative experiment – which really says more about the spread of the individual cells in the experiment rather than estimations of true differences from multiple experiments. The data in **Supplementary Figure 7** has been updated to represent median values from multiple experiments ($n \geq 2$) for better estimation of true changes in the different treatment groups. This is also denoted in relevant areas of the text for clarity.*

“Additionally, the TE dose response in Extended Figures 3D and F are somewhat weak and not (though they are significant)”

- *The target engagement responses in these datasets are likely weaker due to the cells being a heterogeneous population (i.e., without clonal selection), which gives a much greater*

*spread for individual cell TE measurements. The uniformity from clonal cell populations helps condense this spread for better assessment of TE dynamics (as evidenced with R139C in **Supplementary Figure 7j-m** and with other examples in the manuscript).*

2) Also, the authors use dox-inducible biosensors for many experiments, but don't always designate when they are using inducible vs non-inducible. They should just make sure they say so consistently throughout the paper in their figure captions.

- *Thank you for pointing this out to us. We have updated the figure legends so that this is clearer to the reader.*

Overall, the technical execution of the experiments in this work is strong and spans diverse techniques. However, some of the design and conclusions will need additional work before publishing. The impact and significance also have limitations for some of the reasons outlined above and mentioned by the authors in the Discussion section.

- *Thank you for the constructive criticism. We have hopefully addressed many of your concerns in the revised manuscript.*

Reviewer #2 (Remarks to the Author):

Manuscript by Valerie et al describes novel drug target engagement methodology CeTEAM. Based on the destabilizing mutations authors provide evidence that this methodology can be applied to various targets such as MTH1, NUDT15, PARP and also correlate the target engagement with biological activity identifying off target actions of some compounds. Furthermore the methodology is expanded to in vivo and non invasive in vivo imaging. It's a remarkable work, still poor writing and data distribution amongst the figures really hamper its potential.

- *Thank you for your comments. As part of the revision, there has been extensive reorganization of the text, which we hope improves the logical flow, readability, and interpretation. These are more thoroughly addressed below, but here is a brief overview of major changes:*
 - *Replaced single-cell intensity scatterplots with summary dose-response curves, where appropriate*
 - *Removed superfluous data irrelevant to findings and conclusions*
 - *Improved introduction to NUDT15 R139C and relevant ligands – **Figure 1** now begins with only thioguanine then expands to NSC56456 and derivatives later in the phenotype multiplexing section of NUDT15 R139C results*
 - *Clarified descriptions of biological and technical replicates in all figure legends*
 - *Reworked “Discussion” section with dedicated section related to expanding CeTEAM-amenable targets*

1. The authors have 6 main figures, 10 extended figures and 9 supplementary figures that indicates poor judgement in data presentation. Many of the supplementary figures such as 1, 5,6 etc have GFP intensity plots of dose responses for various compounds that the reader is expected to eyeball. They should be quantified, dose response curved provided and representative plots provided while the rest should go into extended data. Majority of us only have 2-3 screens so having 4 sources plus the suppl discussion really will not help authors to get the value of the work appreciated.

- *Thank you for this point. We agree that a lot of data was presented in the original submission, including raw datasets that could have more appropriately been included in summary form (e.g., represented as curves, etc.). We have made a concerted effort to streamline presentation of these data. Specifically:*
 - *Relevant NUDT15 R139C supporting data from the original Extended Figure 3 and Supplementary Figure 1 are now combined as summary datasets in **Supplementary Figure 7**.*
 - *Relevant PARP1 L713F supporting data from the original Supplementary Figures 5 are now represented as summaries in **Supplementary Figure 10e-i** in combination with clonal selection data. Similarly, the raw datasets for the pCHK1 and γ H2A.X immunofluorescence from Supplementary Figures 6 and 7 have been removed but the summary data remains as **Supplementary Figure 11**.*
 - *The FRAP data originally in Supplementary Figure 2 has been trimmed and relevant data has been combined with former Extended Figure 4 (now*

Supplementary Figure 8) to give a more concise overview of PARP1 WT and L713F biophysical characterizations.

- The PARP1 laser microirradiation IF data in former Extended Figure 4 and 5 has been reformed as **Supplementary Figure 9** and combined with L713F toxicity data that was originally presented in Supplementary Figure 3.
- Doxycycline-dependent basal PARP1 L713F-nLuc expression originally shown as Supplementary Figure 9 has been combined with akaLuc characterization data in **Supplementary Figure 13**.
- Relevant data for the PARP1 L713F in vivo work originally described in **Extended Figures 7 and 10** have been combined as one in vivo supplement in **Supplementary Figure 15**.
- The uncropped western blots originally from **Supplementary Figure 10** have now been moved to “Source data” files in accordance with the journal policy.
- Furthermore, some data that was not applicable in shaping the conclusions were removed to de-clutter the paper:
 - **Supplementary Figure 8**, detailing and testing PARP1 L713F complemented with mCherry, was removed as it did not add anything more than already described in current **Figure 7a-c**.

Despite adding significant experimental work related to the rational discovery of CeTEAM-amenable mutants and a small molecule screen for PARP1 L713F stabilizers, the manuscript now contains 8 main figures, 16 supplementary figures, and 4 supplementary tables/data tables. We hope this a more streamlined experience for the reader.

2. Fig 1b is not explained. Why is it needed and what are the compounds there. No assumption should be made that the reader is familiar with TH8234 if wider audience is intended for the paper. The point of the Fig 1 is not clear as similar data for MTH is provided in Fig 2b

- Sorry for this confusion. The intention was to introduce the general concept that these mutants could accumulate by ligand binding in **Figure 1**, then go into more detail related to the target biology/pharmacology in subsequent figures – but this was perhaps unclear. The western blot data from later panels are indeed like **Figure 1** (for both MTH1 G48E [now **Figure 3b**] and PARP1 L713F [now **Figure 5b**]). This was intentional, as we then included relevant phenotypic readouts in this simpler format before transitioning to more quantitative, multiparametric measurements. We have now extensively updated the text to, hopefully, better make this point.

The biggest changes relate to the NUDT15 R139C introduction and relevant ligands. As you mention, TH8228 and TH8234 have not been previously described and are not properly introduced as part of **Figure 1**. We were concerned that doing so at this point would take away from the main message of the figure (mutants being stabilized by ligands). To address this, we now only show the 6TG data in **Figure 1** to make the point that the mutant is amenable to stabilization by a known NUDT15 binder. This also better aligns with the dose-response and time-course described in **Supplementary Figure 1c-d** (which is also only with 6TG). After establishing this, we introduce NSC56465 (TH7410), a

published thiopurine-derived NUDT15i¹³, and describe further functionalization of both an inactive (TH8228) and potential 6TG prodrug analog (TH8234) to systematically explore thiopurine binding in cells. This hopefully improves the logical flow in the R139C section and is easier to follow by the reader.

3. Fig 2e indicates H2 Ser10 signal is going down while the text is stating the opposite

- *Thanks for this observation. While in fact the overall signal of pHH3 Ser10⁺ cells decreased following TH588 treatment, they are still considered positive by threshold gating. A probable explanation for this apparent decrease is that the number of pHH3 Ser10⁺ cells is much higher following TH588 treatment, so there is some saturation of the antibody. In the new experiments for this section (**Supplementary Figure 5f**), we see the same phenomenon. While this may have been prevented by increasing antibody concentration, the trade-off would have been higher background for cells with a lower proportion of pHH3 Ser10⁺ cells.*

4. Most of the figures have limited statements on the replicates performed for western blotting or other techniques.

- *Thank you for pointing this out. As part of the revision, we have updated all the figure legends so that biological/technical replicates are explicitly stated.*

5. The FRAP experiments presented in Suppl Fig 2 are not convincing to support the conclusion that L713F is less mobile than WT PARP1

- *We agree that differences in the FRAP curve plateaus are insignificant, as shown in **Supplementary Figure 8h** (previously, Supplementary Figure 2), indicating more permanent immobilization is essentially the same for both WT and L713F – with or without PARPi. Looking more closely at fast vs slow diffusion components of the curve ($t_{1/2fast}$ vs $t_{1/2slow}$), however, L713F has less transient mobility (reflected by a higher $t_{1/2slow}$) than WT, as seen in the shallower arc approaching the plateau (**Supplementary Figure 8j**). Pre-treatment with olaparib or talazoparib had no additional effect on this basal difference, however.*

To clarify this in the text, we have now updated the sentence on page 12 describing the FRAP results to say: “Fluorescence recovery after photobleaching (FRAP) experiments also suggested that PARP1 L713F has slightly less transient mobility than WT...”

6. Images in extended Fig 5 are critical for the conclusions but they are extremely pixelated, please provide representative higher quality images. Ext Fig 5a controls for various groups look very different? Is equating trapping with PARP1 concentration fair? Fig 4d is a summary of entire Ext Fig 5 as referred in the text? Some illustration in the main figure would be helpful.

“Images in extended Fig 5 are critical for the conclusions but they are extremely pixelated, please provide representative higher quality images. Ext Fig 5a controls for various groups look very different?”

- *We apologize for the image quality. As the data in Extended Figure 5 is now incorporated with the supplementary data (**Supplementary Figure 9**), we have increased the resolution reasonably so that details in the representative images are hopefully clearer.*

*The simplest explanation for the differences in WT PARP1 in **Supplementary Figure 9b** (formerly Extended Figure 5a) is that, unlike the L713F cells, the WT cells are not a clonal population. By chance there are fields that have worse GFP-WT expression than others, and this variation is even clearer in the quantification of GFP-WT signals seen in panel **c** and **i** where there is a pronounced spread of intensities compared to L713F. This also illustrates the advantage of clonal selection for measuring biosensor activity in single cells.*

“Fig 4d is a summary of entire Ext Fig 5 as referred in the text? Some illustration in the main figure would be helpful.”

- *Regarding the PARPi trapping rankings that were originally in Figure 4d, these were referencing the rankings described in the literature, but they also happened to match well with our datasets. The original panel has now been removed at the recommendation of Reviewer 3 and included in the main text (page 13).*

“Is equating trapping with PARP1 concentration fair?”

- *Thanks for the interesting point about equating trapping with PARPi concentration. Given the nature of our CeTEAM assay, we have revised our designation of PARP trapping to require both L713F-GFP stabilization and S-G2 shifts, which follows the logic that PARPi must bind PARP to induce its DNA retention (**Figure 5c**). While this relationship is not absolutely quantitative in our assay, this should facilitate relative comparisons and ranking of PARPi based on the selected readouts.*

7. In the discussion it would be helpful to circle back on how applicable this approach is beyond 3 targets

- *Absolutely. In addition to the new data exploring multiple mutants in the PARP1 HD domain and paralogous expansion to PARP2, we have added significant discussion of this point to the text. To summarize, our data show that it is feasible to identify amenable mutants by applying biophysical logic (with help from structural insights) and that expansion is possible to paralogs with high structural similarity within the mutation region. So, in addition to serendipitous discovery, CeTEAM biosensors can be rationally designed. Theoretically, this can even be extended to proteins without resolved structures using AI-assisted tools, such as AlphaFold. We hope this provides further optimism for applying CeTEAM to diverse drug targets in the future.*

Thank you for your feedback on our manuscript. We hope the revision has adequately addressed your concerns.

Reviewer #3 (Remarks to the Author):

Valerie et al. report the development of a new method for measuring drug target engagement and phenotypic responses in a single assay. Their method, called cellular target engagement by accumulation of mutant (CeTEAM), involves overexpressing the target protein of interest with a destabilizing missense mutation and simultaneously measuring its stabilization by drug binding along with downstream phenotypic effects. The authors demonstrate CeTEAM's utility by investigating MTH1, NUDT15, and PARP1 inhibitors in cells. They further use CeTEAM to investigate PARP1 inhibitor engagement in a mouse model, using ex vivo measurements to show heterogeneity in drug distribution across tumors and in vivo measurements to measure target binding in live mice.

The use of drug-binding-induced stabilization of destabilized mutants to measure target engagement is conceptually interesting and novel. This was even achieved in vivo. Demonstrating on-target engagement and characterizing phenotypic effects are vitally important in drug discovery and development. However, Valerie et. al. do not adequately show (1) how the benefits of measuring target engagement and phenotypic effect in a single multiparametric experiment outweighs the downsides of overexpressing a mutant protein in their system and (2) how CeTEAM is superior to measuring target engagement and phenotypic effects in separate experiments and correlating the results. To more convincingly demonstrate the utility of their assay, Valerie et al. should present novel results or biology that could not be readily discovered with existing methods or show that their method is applicable across a broader range of drug target classes (e.g., kinases). With these major revisions, I believe this manuscript will be of broad interest to the drug discovery and chemical biology communities.

- *Thank you for your comments on the manuscript. Following your suggestions, we have incorporated substantial new data that help bolster our main conclusions, and these have definitively improved the quality of the paper. Please see below for more detailed responses to each of your points.*

Major comments:

1. This assay is framed by the authors as an widely applicable platform that will facilitate drug discovery and advance understanding of drug pharmacology. However, it was not shown within the scope of the paper how this platform has the potential to promote discovery of novel biology and/or outperform currently available platforms that utilize multiple, separate assays to achieve conclusions about drug pharmacology or off-target effects (e.g. drug-resistant alleles). In this regard, we suggest the author consider one of the two following directions to bolster their claims:

A. Demonstrate novel results that can only be found in this multiplexed assay and/or substantiated analysis that shows that this outperforms existing methods

- *To the best of our knowledge, detection of drug binding to a specific target has not been shown by non-invasive imaging in a living organism – which is a key advantage available to this platform. However, we would agree this could be better demonstrated with*

longitudinal studies employing the dual luciferase platform. While perhaps better as a follow-up study, it would be theoretically possible to simultaneously track PARP1 binding and tumor burden to determine if, e.g. the development of drug resistance is due to failed drug-target engagement. We did however address point B as part of our revised manuscript – please read on below.

B. Demonstrate broader applicability and scalability of platform, e.g. across protein families like kinases and/or larger small molecule libraries

- *Thank you for this suggestion. Although we had alluded to this possibility in the original submission, we did not include empirical data to support this claim. As part of the enclosed revision, we have addressed this point from two arms.*

*1) We demonstrate that the CeTEAM-amenable destabilization conferred by PARP1 L713F can be transferred to the analogous residue on PARP2, L269 (**Figure 2f-i, Supplementary Figure 2, Supplementary Figure 10**). Results with the GFP-tethered PARP2 biosensor were essentially the same as with PARP1 L713F-GFP, supporting that most PARPi are equipotent between these paralogs but also that the biosensors are similar in terms of sensitivity (**Supplementary Figure 10j**). These new results argue that amenable mutants could be readily expanded within protein families to close paralogs with similar structural features.*

*2) We also carried out a larger screen of drug-like molecules after 24 hours at 10 μ M with the PARP1 L713F dual luciferase system (**Figure 6, Supplementary Figure 14**). These molecules were sourced from the MedChemExpress Epigenetics and Selleck Nordic Oncology libraries (~1200 compounds total) and contained several PARPi, which made it an intriguing proof-of-concept set. Inclusion of the akaLuc normalization proved critical for triaging acutely toxic molecules and several epigenetic modulators that upregulated signals from both readouts, which may have otherwise been false positives, as nLuc/akaLuc linearity suffered at these extremes (**Supplementary Figure 14d, e**). The screen identified >90% of annotated PARPi with reported biochemical $IC_{50} < 1 \mu$ M (verified by hit confirmation) but also 14 other compounds eliciting a specific increase in L713F-nLuc abundance, which appeared to be related to the specific chemotype, rather than the annotated target for the compound (**Supplementary Figure 6e-h**). Only 4 of these non-PARPi significantly increased L713F-nLuc upon retesting and may be legitimate off-target binders of PARP1; however, several others reproduced small but statistically insignificant stabilization of the biosensor. Notably, the DNMT1/3-trapping inhibitors, azacitidine and decitabine, (but not other non-covalent DNMT inhibitors) were enriched in the screen, which likely reflects the recruitment of PARP1 for repair of DNMT1/3-DNA crosslinks. Thus, a CeTEAM screening format is useful for discovery of direct but also indirect biophysical perturbagens.*

We hope these key additions better support our claims of generalizability and scalability of CeTEAM platforms.

2. In this paper, the authors over-express unstable variants of enzymes that are stabilized by drug binding as a biosensor for cellular target engagement. However, systems that utilize over-expression are known to be susceptible to epiphenomenons that result from non-physiological levels of proteins. To eliminate any doubts about veracity of conclusions established by the results of the platform, the following controls experiments should be added:

- *This is absolutely a valid point. We utilize tagged, exogenous mutants to demonstrate a proof-of-concept that one can seamlessly integrate phenotypic readouts using stability-dependent drug biosensors. But as now shown with OGG1 R229Q, this is also applicable to endogenously expressed variants (Supplementary Figure 3). While the fusion tag is not strictly necessary, it can facilitate specific and robust detection.*

*To better reflect this, we have updated the graphical overview of the assay in **Figure 1** and discuss this issue as it pertains to interpretation of phenotypic results and potential epiphenomena in the Discussion. Please see below for detailed responses regarding the inclusion of other controls.*

A. Repeat one of the three systems with more variants with destabilizing missense mutations to verify stabilization effect for all the mutants regardless of mutation position and show the dynamicity of the systems beyond a single variant.

- *Thank you for this suggestion. We had planned to do this as part of a follow-up study but understand that it would be more effective at supporting the initial description of the assay. We have now included new data comparing multiple leucine mutations within the PARP1 HD domain and relating them to available biophysical measures of stability change (i.e., differential scanning fluorimetry; **Figure 2a-e, Supplementary Figure 4a**). Briefly, we see that the biosensing ability of the mutation follows biophysical logic and is dependent on surrounding structural features. While several HD leucine mutants do recapitulate the drug biosensing ability of L713F, the most interesting case was the additive destabilization conferred by combining two mutations distal to the hydrophobic core (L698A and L701A). Either mutant alone was comparable to WT PARP1, while their combination imparted a stabilization dynamic range like L713F (**Figure 2, Supplementary Figure 4**). Thus, while certain mutations could be globally destabilizing (and not rescuable by ligand binding), these results underscore that CeTEAM-amenable destabilization can be rationally designed, especially with available structural information.*

Further discussion of these points with relation to the generalizability of CeTEAM are also covered in more detail in the Discussion section.

B. Quantitative experiments that over-express the wild-type variants instead of the missense variant should be added to complement results found with the destabilized variant for MTH1 and NUDT15 to show that effects seen are specific to protein activity and not caused by the specific mutant used. Alternatively, specificity of downstream cellular effects can also be verified by introducing a mutation that results in a catalytically-dead enzyme on top of the destabilizing mutation.

- *Thanks for this suggestion. In addition to the WT protein data given before with MTH1 (Figure 3b), we have now supplemented with the WT protein for flow cytometry experiments to make a more compelling point regarding MTH1i-related phenotypes (Figure 3e and f, Supplementary Figure 5b-d). Further, WT NUDT15 has been added for comparison to R139C (Supplementary Figure 7a and b). We agree that these new data should help strengthen the biological conclusions related to these mutants.*

The idea of also removing catalytic activity from the biosensor is interesting and could eliminate some potential confounding factors related to epiphenomena. We have raised this possibility in the Discussion section as one alternative to get around this issue.

Minor comments:

1. For all figure captions, please provide complete descriptions of the experiments/analysis performed, the cell type used, as well as relevant concentrations and time points.

- *Thank you for this point. We have done our best to include complete descriptions of the figures (within the limitations set by the journal on figure legend length). Hopefully this has improved clarity, but additional details are given in the Methods section, where necessary.*

2. Because the focus of this manuscript is a new concept/method in chemical biology, readers may not be familiar with MTH1, NUDT15, and PARP1 biology specifically. Please provide more background on the biology of these enzymes, why they were chosen for this method, and how their activity is typically quantified. For example, provide a more complete description of DNA trapping assay using Hoechst dye.

- *Thank you for pointing this out. We have reworked the backgrounds on MTH1, NUDT15, and PARP1 for their relevant Results sections (pages 7, 9, and 11) so that it is clearer what their functions are and why they are of interest for use with CeTEAM.*

For some newer examples demonstrating the existence of CeTEAM-amenable mutants (DHFR and OGG1), we do not give much background due to space constraints, as we only wish to show they are stabilized by cognate ligands. However, they are of general interest as cancer therapeutic targets, and, more importantly, have reported destabilizing mutants and small molecule inhibitors. PARP2 is introduced as a PARP1 paralog for the purposes of expanding the biosensor pool via structural similarity. We give context here regarding the selectivity of PARPi towards PARP1/2.

Regarding Hoechst intensity as a surrogate readout for PARP trapping, we have provided more details in the text but also in the graphical depiction of the described CeTEAM assay (pages 12-13; Figure 5c). Briefly, we describe previous literature about “trapping”-dependent replication stress and how measuring S-G2 cell cycle shifts (with e.g, DAPI or Hoechst) accurately reflects this phenomenon. We then applied this readout of replication stress and add the dimension of PARP1 binding (via L713F abundance) to define trapping

as requiring both aspects, which more completely enables screening for trapping PARPi with this CeTEAM assay.

3. Several of the figures could benefit from rearranging subpanels to ensure clarity. In general readers expect panels to progress left to right and top to bottom.

- *Thank you for this suggestion. We have rearranged some of the figures to improve clarity (e.g., **Figure 4**, **Figure 5**, **Figure 7**, and **Supplementary Figure 13**, among others). Newer figures also adhere to these expectations.*

Figure by Figure comments:

Figure 1:

Please simplify panel D by removing or moving the cells representing downstream phenotype. The current arrangement is unclear.

- *We apologize for the confusion. We agree that the previous graphic was a bit complicated and have now replaced it with an updated, simplified version (**Figure 1d**). We also emphasize in the legend (as well as the Discussion) that the presence of a fusion tag is not necessary but can aid detection of changes in abundance of the mutant proteins.*

Figure 2:

• The claim that TH588 triggers the mitotic surveillance pathway independent of MTH1 binding is only partially supported in this study. Please provide additional background on previous work that came to the same conclusion.

- *We agree that this specific mechanism was not adequately supported in the original submission. To do this, a demonstration TH588-dependent mitotic delay transitioning to a G1 arrest would be more convincing. Gul et al. proposed this mechanism for TH588 in 2019, whereby cells escaping TH588-dependent prolonged mitoses are prevented from re-entering the next cell cycle via a USP28- and p53-dependent G1 arrest¹¹. They also demonstrate that higher TH588 doses (~>8 μ M) yield a more permanent mitotic arrest, while lower doses can eventually progress through mitosis and instead get stuck in the subsequent G1-phase.*

*In the revision, we have predominantly focused on using the G48E biosensor as a surrogate measure of MTH1 occupancy to demonstrate that TH588-related mitotic problems arise independently of MTH1 binding (**Figure 3c-f**, **Supplementary Figure 5**), which we felt made a more compelling example for employing CeTEAM. However, as further support for the mitotic surveillance hypothesis, we have included a comparison of cell cycle profiles following two different doses that elicit anti-cancer effects: 2.5 μ M TH588 (from the original submission) and 10 μ M from the revised dataset. Indeed, in agreement with Gul et al., we see that cells treated with 2.5 μ M TH588 have elevated levels of mitotic cells but also a significant G1-phase delay, whereas 10 μ M were strictly enriched in mitosis (**Supplementary Figure 5h and i**). Nonetheless, we have modified the language on pages*

8-9 to provide more details on this mechanism and merely that our data support this possible explanation for TH588-mediated toxicity.

- For panels H and I, I am concerned about the robustness of an analysis based on a gate that includes only 2% of the DMSO treated cells. Please provide a head to head analysis with and without the initial “V5 high” gate.

- *This is absolutely a concern, especially when measuring relatively rare events such as mitotic cells, which may give an inadequate representation of the population. In the original submission, we did include a comparison of pHH3 Ser10⁺ cells in the total population alongside the designated “V5 high” population (former Figure 2j). In these experiments we collected ~20,000 total events per condition, which was perhaps on the low side for studying events that typically represent <2% of the population.*

*For the newer experiments included in the revision, we demonstrate this analysis again comparing 10 μ M MTH1i but after having collected ~100,000 events per sample, which helped improve the robustness of these measurements to some extent (**Supplementary Figure 5e-g**), although the high stringency of this gating will always generate more noise in “V5 high” DMSO populations. As before, we include the comparison of “V5 high” and total cell populations and we see a strong enrichment of pHH3 Ser10 in cells with high TH588 exposure (high V5 signal, **Supplementary Figure 5g**).*

- For panels D-I, please provide additional controls: Overexpressed WT MTH1, enzymatically inactive MTH1, and MTH1 with other destabilizing missense mutations. (See major comments 2B)

- *As mentioned above, we have now included WT MTH1 as a control in the flow cytometry analyses for comparison (**Figure 3, Supplementary Figure 5**). Following the prompt in the major comments above, we included WT MTH1 in lieu of a catalytic dead variant combined with G48E to help make this point. For the surveying of other destabilizing mutations within a given target, please refer to the new data with the PARP1 HD domain mutants (mentioned above; **Figure 2a-e**).*

- Consider supplementing cell cycle data in parts F-I with proliferation data to show the overall effect of inhibitor treatments.

- *We had considered including proliferation data to complement our analyses, but we reasoned that these measurements would require several days to manifest and, therefore, did not include them. However, an earlier study demonstrated that proliferation defects were more apparent with TH588 compared to AZ19 after 5 or 7 days (we refer to this in the summary of anti-cancer effects described for TH588 in the text)³. At 24 hours, our analyses indicated that TH588 elicited small increases in sub-G1 cells at 10 μ M, suggesting early stages of apoptosis (**Figure 3f**).*

Figure 3:

- While the conceptual framework of “non-responder, stabilizer and potentiator” is helpful, TH8234 does not fit cleanly into this model. The authors suggest that TH8234 acts both as a potentiator and a thiopurine prodrug. While this is a possibility, there is insufficient evidence to rule out unintended off-target effects that TH8234 may have in cells, especially at high (33uM) concentrations. To make this claim, I suggest performing further experiments to detect TH8234 to thiopurine conversion. Additional indirect evidence could be provided by combining 200nM 6-TG with other thiopurines to show a similar effect on DNA damage markers.

- *Thank you for this comment. Unfortunately, we did not have the capacity to measure the metabolic conversion of TH8234 to thioguanine by LC-MS, which would have been an ideal approach to confirm this possibility. Instead, we have made some revisions to clarify this point and better support our claims.*
 - *We have updated the model in **Figure 4i** to include the designation of “6TG mimetic” to describe compounds that both bind NUDT15 R139C and elicit DNA damage without supplemental 6TG – i.e., what we see with TH8234.*
 - *We have clarified in the text that TH8234 alone exhibited both markers at multiple doses (from ~500 nM upward). Further, **Figure 4h** has been updated to compare 3.67 μ M of each NUDT15i \pm 200 nM 6TG with the added cell cycle dimension, facilitating the identification of 6TG-dependent DNA damage and G2 arrest. We see that TH8234 alone resembles a similar dose of 6TG (3.33 μ M), and the addition of low-dose 6TG appears to exaggerate this phenotype further.*
 - *We additionally emphasize in the text that low-dose 6TG enhanced TH8234-dependent G2-phase arrest and DNA damage at lower doses (**Supplementary Figure 7g-i**).*

While not definitive, these changes to the presented data and the text implicate the conversion of TH8234 to a 6TG-like metabolite, although we are now explicit in the text on page 11 that we did not confirm this possibility by other means.

- Panel H: The authors do not discuss the clear differences in cells treated with NSC56456 and cells treated with TH8234. TH8234 appears to induce an accumulation in S phase as opposed to G2/M. Visually, this accumulation is very different (and potentially more prone to noise) than the accumulation seen in NSC56456 treated cells. This merits further investigation.

- *Thanks for this point. The data shown in **Figure 4h** (formerly Figure 3h) were representative from the dose-response experiments described in **Figure 4f**. At the 33 μ M dose originally shown, TH8234 toxicity is quite high resulting in even fewer (dying) cells, therefore, a likely poorer indication of cell cycle (see cell numbers in Source Data for **Supplementary Figure 7i and m**). To better illustrate this point, we have revamped this panel to demonstrate the changes elicited by the NUDT15i alone and in combination with low-dose 6TG. This helps to delineate 6TG-dependent DNA damage and G2-phase arrest more clearly and demonstrates that TH8234 elicits a similar response on its own, especially when compared to a higher dose of 6TG (3.33 μ M) from a separate experiment.*

Figure 4:

- Panel D, which is all text, could be included in the body of the manuscript instead of in a figure.

- *Thank you for this suggestion. The original panel **d** has been removed from the figure and incorporated in the main text on page 13.*

- Panel E could be split into two panels.

- *The original panel **e** has now been split into two distinct panels, **d** and **e**, which we hope improves clarity.*

- Panel E-G. State exactly how Hoechst dye intensity relates to DNA trapping.

- *Absolutely. We discussed this point in greater detail above, but to summarize, we describe this relationship in better detail in the main text (page 13) but also update the graphical procedure in **Figure 5c**. Trapping PARPi elicit DNA replication stress (S-phase delay) and DNA content measurement has been an effective readout of this effect (Michelena et al., 2018)¹⁵. With this CeTEAM assay, we take this a step further and can define PARP trapping as binding to PARP (L713F biosensor) and the presence of DNA replication stress (S/G2-phase enrichment).*

Figure 5:

- Provide a panel that directly shows that the PARP1 sensor detects the spatial distribution of niraparib. Current panels clearly show the PARP sensor is stabilized in vivo, but do not show a clear gradient of signal across the tissue/tumor. Claims of spatial resolution depend on more specific spatial data and these should be highlighted.

- *Thanks for this distinction. We agree that the data do not demonstrate a clear gradient of signal across the tumor by immunofluorescence, but more accurately, that we are able to detect niraparib binding at the level of single cells within the tumor mass. The text has been updated on page 16 to reflect this.*

- In the text, reserve claims of temporal resolution for the discussion of Figure 6. Figure 5 only provides potential evidence for spatial resolution, not spatiotemporal.

- *Absolutely a fair point. We have updated the text in the ex vivo measurement section to only consider potential spatial insights.*

Thank you very much for your constructive critique. Many of your points significantly improved the quality of our study.

References

1. Sun, K. et al. A comparative pharmacokinetic study of PARP inhibitors demonstrates favorable properties for niraparib efficacy in preclinical tumor models. *Oncotarget* **9**, 37080-37096 (2018).
2. Mur, P. et al. Germline variation in the oxidative DNA repair genes NUDT1 and OGG1 is not associated with hereditary colorectal cancer or polyposis. *Hum Mutat* **39**, 1214-1225 (2018).
3. Kettle, J.G. et al. Potent and Selective Inhibitors of MTH1 Probe Its Role in Cancer Cell Survival. *J Med Chem* **59**, 2346-2361 (2016).
4. Rehling, D. et al. Crystal structures of NUDT15 variants enabled by a potent inhibitor reveal the structural basis for thiopurine sensitivity. *J Biol Chem*, 100568 (2021).
5. Chen, H.D. et al. Increased PARP1-DNA binding due to autoPARylation inhibition of PARP1 on DNA rather than PARP1-DNA trapping is correlated with PARP1 inhibitor's cytotoxicity. *Int J Cancer* **145**, 714-727 (2019).
6. Lévy, F., Johnston, J.A. & Varshavsky, A. Analysis of a conditional degradation signal in yeast and mammalian cells. *Eur J Biochem* **259**, 244-252 (1999).
7. Jolivet, J. & Chabner, B.A. Intracellular pharmacokinetics of methotrexate polyglutamates in human breast cancer cells. Selective retention and less dissociable binding of 4-NH₂-10-CH₃-pteroylglutamate⁴ and 4-NH₂-10-CH₃-pteroylglutamate⁵ to dihydrofolate reductase. *J Clin Invest* **72**, 773-778 (1983).
8. Ellermann, M. et al. Novel Class of Potent and Cellularly Active Inhibitors Devalidates MTH1 as Broad-Spectrum Cancer Target. *ACS Chem Biol* **12**, 1986-1992 (2017).
9. Petrocchi, A. et al. Identification of potent and selective MTH1 inhibitors. *Bioorg Med Chem Lett* **26**, 1503-1507 (2016).
10. Kawamura, T. et al. Proteomic profiling of small-molecule inhibitors reveals dispensability of MTH1 for cancer cell survival. *Sci Rep* **6**, 26521 (2016).
11. Gul, N. et al. The MTH1 inhibitor TH588 is a microtubule-modulating agent that eliminates cancer cells by activating the mitotic surveillance pathway. *Sci Rep* **9**, 14667 (2019).
12. Valerie, N.C.K. et al. NUDT15 Hydrolyzes 6-Thio-DeoxyGTP to Mediate the Anticancer Efficacy of 6-Thioguanine. *Cancer Research* **76**, 5501-5511 (2016).
13. Zhang, S.M. et al. NUDT15-mediated hydrolysis limits the efficacy of anti-HCMV drug ganciclovir. *Cell Chem Biol* **28**, 1693-1702.e1696 (2021).
14. Zhang, S.M. et al. Development of a chemical probe against NUDT15. *Nat Chem Biol* **16**, 1120-1128 (2020).
15. Michelena, J. et al. Analysis of PARP inhibitor toxicity by multidimensional fluorescence microscopy reveals mechanisms of sensitivity and resistance. *Nat Commun* **9**, 2678 (2018).

REVIEWERS' COMMENTS

Reviewer #1 (Remarks to the Author):

In the revised manuscript, Valerie et al. provide significant improvement to the overall study and provide a significant amount of additional requested data. The conclusions are valid and experiments well executed.

The primary critique of the translatability of the CeTEAM platform to different target proteins has been addressed in part with the inclusion of additional examples and paralog analysis. While broad applicability across target classes will still be a challenge for the CeTEAM platform, it is less concerning with the next data. Still, this reviewer would be interested to see applicability to non-enzyme targets in the future.

Secondly, the authors address well the choices of compound dose and pharmacological read out to build more quantitative relationships. Increasing the focus on the more quantitative readouts as opposed to the Western Blots is helpful and lends to stronger conclusions overall. This reviewer appreciates the improved and nuanced language the authors use as they decipher complex pharmacology. This strengthens their conclusions as they also appropriately denote limitations and caveats.

Thirdly, the inclusion of the drug screening portion of the paper adds a broader applicability piece that was perhaps missing in the original manuscript. If a biosensor mutant is available for a given target, it could be a useful tool for drug hunters. I'd encourage the authors to think about demonstrating the CeTEAM platform on potential drug targets where this could have outsized utility in parsing pharmacology.

Finally, the work is quite data heavy, and so this reviewer appreciates the improvements to clarity of the visual and text portions of the figures, as well as within the body of the text.

This resubmission is appropriate for publication in Nature Communications and will be of interest to a broad audience across disciplines.

- Thank you for your constructive comments! We are pleased that the resubmitted manuscript has addressed most of your concerns.

Reviewer #2 (Remarks to the Author):

The concerns and suggestions were fully addressed

- Thank you. We are happy to read that your comments were adequately addressed.

Reviewer #3 (Remarks to the Author):

The updates to this manuscript substantially bolster our confidence that CeTEAM is a valid and generalizable technology. The authors have sufficiently addressed our concerns about the

usefulness of CeTeam by including additional proteins and performing a small molecule screen. They have also allayed our concerns about overexpression by including additional wildtype controls and demonstrating a similar effect endogenously with OGG1. While they did not attempt to design an inert, catalytically dead sensor, we consider the many other additions to their study to be sufficient. We recommend this paper for acceptance and publication in Nature Communications.

- We are thrilled that the revisions have sufficiently addressed your concerns. Thank you for the thoughtful criticism and recommended improvements.

The following are additional suggestions to improve the manuscript for readability and accuracy:

1. Claims that their PARP1 experiments enable “rational” design of sensors based on “biophysical logic” should be scaled back. While successful in this case, the manuscript does not detail specific rules about how to design these destabilizing point mutations for any protein.

- This is a fair criticism given the extent of supporting data thus far. We have redacted these statements throughout the text to soften this stance. In particular, the following has been added to the Discussion on page 20: “While these results suggest that suitable CeTEAM mutations can be defined and applied to structural paralogs to expand the target pool, further examples are needed to reinforce this possibility. The advent of modern computational, artificial intelligence, and protein engineering technologies should be helpful in this endeavor and may even extend to proteins without empirically resolved structures.”

2. Supplementary Fig. 5 E-G could be further improved. The gating strategy and sequence is unclear, and it is unclear how each subgate relates to each other. It would be best to clarify: at what point are singlets gated? The way panel G is written, it seems that singlets are a subgate of V5 high; is this correct? Is the histogram in e already gated for singlets? What is the “parental population” in g?

- We agree that the clarity of Supplementary Fig. 5e-g could be further improved. To this end, we have added a concise overview of the gating structure above the V5 intensity histograms in panel e. Furthermore, we’ve updated the summary table in panel g to more clearly define which cell populations we are referring to. Hopefully, this is now easier to follow for the reader.

3. Fig. 4 G-H could be simplified greatly using bar graphs for data quantitation while the images could be moved to the supplement.

- Thank you for this suggestion – we certainly understand where the reviewer is coming from, as Figure 4 is certainly quite data-rich. After careful consideration, we have decided to move the images to the supplement but leave the data from Figure 4h. We left 4h (now 4g) as-is because we believe that this is the clearest data demonstrating that TH8234 is a likely 6TG prodrug. This layered representation, which adequately shows the link between G2-phase enrichment and DNA damage, a hallmark of thiopurine responses, shows the

high similarity between TH8234 and an equivalent amount of 6TG alone. In this sense, we felt it was important to provide this in the main figure.

4. Fig. 4 seems to use NSC56456 and TH7410 interchangeably. If this is indeed the same molecule, please use one name consistently in the figure and text.

- Thank you for pointing this out. They are indeed referring to the same molecule, but TH7410 was included by mistake. This is now replaced with NSC56456, in line with the rest of the paper.